# LOG-BIT DISTRIBUTED LEARNING WITH HARMONIC MODULATION

## ABSTRACT

We consider distributed learning over a communication graph where decentralized clients, as local data owners, exchange information only with their neighbors to train a system-level model, making communication complexity a critical factor. To mitigate this complexity, we introduce a communication quantization scheme based on Harmonic Modulation, in which high-dimensional vectors are compressed and quantized prior to transmission, thereby substantially reducing communication overhead. Building on this idea, we propose Log-Bit Gradient Descent with Harmonic Modulation, where each sender compresses a $d$-dimensional vector into a single scalar, quantizes it into an $m$-bit binary codeand transmits it to the receivers for decoding. Under a sufficient condition, our method achieves an $\mathcal{O}(1/t)$ convergence rate, where $t$ denotes the number of iterations. Moreover, we establish a conservative lower bound showing that only $\log_2(\mathcal{O}(d))$ bits per communication are required, with $d$ representing the vector dimension. Experimental results on synthetic quadratic optimization, logistic regression, and neural network training validate our approach. In logistic regression, LBGD-HarMo matches baseline accuracy while using $800\times$ fewer bits per iteration and nearly two orders of magnitude less communication. In neural network training, each client transmits only 0.0001 MB per iteration while maintaining accuracy.

## 1 INTRODUCTION

In recent years, the vast amount of data generated by physically decentralized systems has sparked significant interest in federated and distributed learning (DL), where multiple devices, servers, or organizations collaboratively train a shared model without directly sharing their raw data (Konečný et al., 2016; Mcmahan et al., 2017; Mohri et al., 2019; Pillutla et al., 2022). The objective of federated learning is to solve the following system-level optimization problem,

$$\min F(\boldsymbol{x}) = \sum_{i=1}^{n} f_i(\boldsymbol{x}) \tag{1}$$

where $\boldsymbol{x} \in \mathbb{R}^d$ represents parameters of a global modeland $f_i(\boldsymbol{x}) : \mathbb{R}^d \to \mathbb{R}$ is the local loss function from the data owned privately by $i \in \mathcal{V} = \{1, 2, \ldots, n\}$. The agents share model updates (such as gradients or parameters) with a central server, e.g., (Fallah et al., 2020; T. Dinh et al., 2020; Li et al., 2020; Kairouz et al., 2021), which then aggregates these updates to improve the global model. The strength of federated learning lies in its capacity to preserve data privacy, improve scalabilityand reduce communication overhead, as opposed to methods that rely on centralizing all data for training.

In the standard federated learning setup, the role of the central server may be replaced by fully distributed information aggregation mechanisms. The cost function $F(\mathbf{x})$ in (1) is inherently separable, a feature long studied in distributed optimization (Tsitsiklis, 1984; Nedić & Ozdaglar, 2009; Duchi et al., 2012). In such schemes, the agents in $\mathcal{V}$ are connected via wired or wireless links that define a communication graph. Each agent exchanges updates only with its immediate neighbors, aggregates the received information through distributed averagingand refines its local model using its private data, for example via distributed gradient descent. These algorithms provide excellent convergence guarantees and scalability for convex problems. Moreover, in the machine learning setting, distributed learning enhances security and privacy by eliminating the need for a central server, which may otherwise be malicious or vulnerable to attack (Li et al., 2020).

One of the central challenges in distributed learning is the high communication complexity. In both federated and decentralized settings, every update requires agents to exchange real-valued vectors whose dimension equals that of the model parameters. This quickly becomes a scalability bottleneck, particularly for modern large-scale models (Seide et al., 2014). To alleviate this, two common strategies are employed: compression and quantization. Compression methods, such as Top-$\alpha$ sparsification (Alistarh et al., 2018), reduce communication load by sending only a fraction of the vector entries, while quantization (Alistarh et al., 2017) lowers the bit-width of each transmitted entry by mapping continuous values onto a discrete set. While both are effective in practice, applying them naively, either alone or in combination, may result in instability or divergence in decentralized optimization (Arjevani et al., 2023). To counteract this, error-feedback mechanisms (Stich et al., 2018) are widely adopted, as they compensate for the bias induced by compression and quantization, thereby supporting higher compression ratios and the use of low-precision representations. Furthermore, most distributed algorithms suffer from slower convergence with sparse communication links, whereas single-scalar communication can outperform compression and quantization, especially in wireless sensor networks where bandwidth and energy constraints make high-dimensional communication inefficient (Zhang et al., 2024; Joseph et al., 2025).

In this paper, we introduce **L**og-**B**it **G**radient **D**escent with **Har**monic **Mo**dulation (**LBGD-HarMo**), a fully digital and distributed framework for learning over graphs. The method integrates three key components: (i) a harmonic modulation scheme that compresses high-dimensional updates into single real-valued statistics, (ii) a quantizer that converts the compressed updates into binary representationsand (iii) a distributed primal–dual algorithm that enables local updates with quantized information. This design provides a principled solution to communication-efficient distributed learning over digital channels, while preserving strong theoretical convergence guarantees under convexity assumptions. The main contributions of this work are summarized as follows:

- We prove that under standard connectivity (for the communication graph) and convexity (for the cost functions) assumptions, the LBGD-HarMo achieves the optimal $\mathcal{O}(1/t)$ convergence rate, while requiring only $\log_2(\mathcal{O}(d))$ bits of communication per iteration, where $t$ denotes the number of iterations and $d$ is the dimension of the decision variable.

- We conduct experiments on synthetic quadratic optimization, logistic regression and neural network training tasks. The results demonstrate that LBGD-HarMo achieves comparable convergence to representative decentralized baselines, including DSGD (Lian et al., 2017), CHOCO with Top-$\alpha$ compression (Koloskova et al., 2020a), MoTEF with Top-$\alpha$ (Islamov et al., 2025)and LBGD with Sign quantization, while requiring up to two orders of magnitude fewer transmitted bits to reach the same target accuracy.

To the best of our knowledge, LBGD-HarMo is the first distributed optimization and learning framework that operates under logarithmic bit rates, thereby opening new avenues for both theoretical investigation and practical deployment.

**Large Language Models**. The authors used large language models solely for polishing the writing. They were not employed for retrieval, discovery, or research ideation.

## 2 PROBLEM DEFINITION

### 2.1 DISTRIBUTED LEARNING ON GRAPHS

We consider a system with $n$ clients. Each agent $i \in \mathcal{V}$ possesses a private local dataset $\mathbb{D}_i$, a loss function $f_i : \mathbb{R}^d \to \mathbb{R}$and a learning model $\boldsymbol{x}_i \in \mathbb{R}^d$. The agents are interconnected via a connected and undirected communication graph $\mathcal{G} = (\mathcal{V}, \mathcal{E})$. The system-level goal is described by the following optimization problem:

$$\min_{\boldsymbol{x}} F(\boldsymbol{x}) = \frac{1}{n} \sum_{i=1}^{n} f_i(\boldsymbol{x}_i; \mathbb{D}_i) \tag{2}$$
$$\text{s.t. } \boldsymbol{x}_i = \boldsymbol{x}_j, \quad \forall i, j \in \mathcal{V}.$$

Any optimal solution to (2) implies a learning model that is trained on the collection of all datasets $\mathbb{D}_i, i = 1, \dots, n$. We are interested in distributed algorithms that solve (2) with digital communications, i.e., agents only share digital messages with neighbors on the graph $\mathcal{G}$.

## 2.2 QUANTIZED COMMUNICATION

Clearly, all communication taking place over the graph $\mathcal{G}$ must be digital. An $m$-bit quantization function (Kajiyama et al., 2021) for some $m \in \mathbb{N}_+$ is a mapping $q_m : \mathbb{R} \to \mathbb{R}$ which maps a real value $a \in \mathbb{R}$ to a quantized value with finite levels. Given integer parameters $m_1, m_2 \in \mathbb{N}_+$ satisfying $m_1 + m_2 = m$, we define $K := 2^{m_1-1}$ and $l := 2^{-m_2}$ as the quantization boundary and the quantization error, respectively. Then, $q_m(\cdot)$ is defined component-wise by

$$
q_m(a) = \begin{cases} K - \dfrac{l}{2}, & a > K; \\ I, & a \in (I - \dfrac{l}{2},\ I + \dfrac{l}{2}]; \\ -K + \dfrac{l}{2}, & a \le -K. \end{cases} \tag{3}
$$

where $I = \pm\dfrac{1}{2}l,\ \pm\dfrac{3}{2}l,\ \ldots,\ \pm(K - \dfrac{1}{2}l)$. The quantization error always satisfies

$$
|q_m(a) - a|_\infty \le \frac{l}{2}, \quad \forall |a| \le K.
$$

Next, the function $\tilde{q}_m : \mathbb{R} \to \{0, 1\}^m$ is a binary encoder that transforms the output of $q_m$ into an $m$-bit digital representation suitable for communication. That is,

$$
\tilde{q}_m(a) := \texttt{BinEncode}(q_m(a)), \tag{4}
$$

where `BinEncode` maps each quantized value of $q_m(a)$ to one of $2^m$ pre-defined binary codes shared among all clients.

## 2.3 RELATED WORK

**Decentralized Optimization.** Research on decentralized optimization began with the seminal work of Tsitsiklis (1984), which analyzed distributed decision-making and optimization over networks. Subsequent progress was achieved through gossip protocols, where clients iteratively average information with neighbors, including randomized gossip (Kempe et al., 2003), fastest mixing gossip (Xiao et al., 2004)and randomized analysis (Xiao & Boyd, 2004). These protocols highlighted that local information exchanges along graph edges are sufficient for reaching global agreement. Building on these insights, distributed (sub)gradient methods were developed to solve convex programs. Nedić & Ozdaglar (2009) proved convergence under diminishing stepsizesand Johansson et al. (2010) extended the analysis to randomized and asynchronous updates. At the same time, distributed ADMM formulations were proposed for consensus and constrained optimization (Wei & Ozdaglar, 2012; Iutzeler et al., 2013), while decentralized dual averaging schemes provided topology-dependent convergence guarantees (Duchi et al., 2012; Nedić et al., 2015). In recent years, these algorithmic foundations have been extended to machine learning applications. He et al. (2018) investigated decentralized training for generalized linear models. Gao et al. (2024) introduced compressed decentralized SGD for large-scale nonconvex learning.

**Communication Compression and Quantization.** Reducing communication overhead is a major challenge in decentralized optimization. Two main directions have been explored: compression and quantization. Compression-based methods aim to reduce the dimensionality of transmitted information. For instance, Beznosikov et al. (2023) analyzed biased operators such as Top-$\alpha$ sparsification and established convergence with error compensation. In addition, Wang et al. (2024) proposed scalarized communication schemes and proved linear convergence for distributed linear equations. Quantization-based methods, in contrast, focus on reducing bit precision. Thanou et al. (2012) examined consensus under uniform quantization and introduced refinement strategies to improve accuracy. Reisizadeh et al. (2019) proposed an encoding/decoding mechanism ensuring vanishing consensus errorand Doan et al. (2020a;b) developed unbiased random and adaptive quantization rules with linear convergence guarantees. Kajiyama et al. (2020) further established linear convergence via time-varying quantizers. In stochastic optimization, Bernstein et al. (2018) introduced the SignSGD algorithm that communicates only gradient signsand Karimireddy et al. (2019) incorporated error-feedback to show that compressed updates can attain convergence rates comparable to full-precision methods.

# 3 HARMONIC MODULATION

## 3.1 HARMONIC MODULATION

In this section, we propose the Harmonic Modulation (HarMo), which reduces each $d$-dimensional vector to a single scalar, then quantifies it into an $m$-bit digital representation. This drastic reduction in message size enables efficient decentralized communication while preserving convergence. In the following, we present a detailed formulation and analysis of the proposed HarMo.

We define some functions in our compression process: the harmonic compression sequence $\boldsymbol{\psi}_{\text{HarMo}}(t)$, the HarMo encoder $\mathcal{C}_{\text{E}}$ and its decoder counterpart $\mathcal{C}_{\text{D}}$.

**Harmonic Modulation Sequence $\boldsymbol{\psi}_{\text{HarMo}}(t)$.** The HarMo sequence $\boldsymbol{\psi}_{\text{HarMo}}(t) \in \mathbb{R}^d$ is defined as:

$$\boldsymbol{\psi}_{\text{HarMo}}(t) = \left[\sin\left(\frac{\pi}{d+1}t\right),\ \sin\left(\frac{2\pi}{d+1}t\right),\ \ldots,\ \sin\left(\frac{d\pi}{d+1}t\right)\right]^\top, \tag{5}$$

where $t \in \mathbb{N}$ denotes the communication round or iteration indexand $d$ is the dimensionality of the original vector. This harmonic structure introduces periodicity and diversity across time steps, allowing the compressor to project high-dimensional information along varying directions with minimal computational and memory cost. Notably, since $\boldsymbol{\psi}_{\text{HarMo}}(t)$ is deterministically constructed and shared among all clients, it requires no additional communication, making it highly efficient in decentralized, bandwidth-constrained settings. Importantly, the harmonic structure of $\boldsymbol{\psi}_{\text{HarMo}}(t)$ is reminiscent of the basis functions used in the Fourier transform, enabling the system to approximate frequency-aware projections of the original signal. This analogy allows the compressor to implicitly exploit the spectral structure of the input, which is particularly beneficial for preserving informative components under aggressive quantization.

**HarMo Encoder $\mathcal{C}_{\text{E}}$.** The function $\mathcal{C}_{\text{E}} : \mathbb{R}^d \times \mathbb{N}_+ \to \{0,1\}^m$ projects a $d$-dimensional real-valued vector $\boldsymbol{b} \in \mathbb{R}^d$ onto a scalar using $\boldsymbol{\psi}_{\text{HarMo}}(t)$, then applies the quantization function $\tilde{q}_m$ into an $m$-bits binary representation suitable for transmission. That is,

$$\mathcal{C}_{\text{E}}(\boldsymbol{b}, t) := \tilde{q}_m(\boldsymbol{\psi}_{\text{HarMo}}(t)^\top \cdot \boldsymbol{b}), \tag{6}$$

**HarMo Decoder $\mathcal{C}_{\text{D}}$.** The decoder function $\mathcal{C}_D : \{0,1\}^m \times \mathbb{N}_+ \to \mathbb{R}^d$ first decodes the received $m$-bit binary message (e.g., $\mathcal{C}_{\text{E}}(\boldsymbol{b},t)$) into a quantized scalar value. It then reconstructs a $d$-dimensional vector by expanding this scalar along $\boldsymbol{\psi}_{\text{HarMo}}(t)$,

$$\mathcal{C}_{\text{D}}(\{0,1\}^m, t) := \boldsymbol{\psi}_{\text{HarMo}}(t) \cdot \texttt{BinDecode}(\{0,1\}^m), \tag{7}$$

where $\texttt{BinDecode}$ recovers the quantized real-valued scalar from the corresponding $m$-bit binary representation.

**Definition 3.1.** The Harmonic Modulation Channel $\mathcal{C}_{\text{HarMo}}: \mathbb{R}^d \times \mathbb{N}_+ \to \mathbb{R}^d$ satisfies

$$\mathcal{C}_{\text{HarMo}}(\boldsymbol{b}, t) = \mathcal{C}_{\text{D}}(\mathcal{C}_{\text{E}}(\boldsymbol{b}, t), t) = \boldsymbol{\psi}_{\text{HarMo}}(t) \cdot (q_m(\boldsymbol{\psi}_{\text{HarMo}}(t)^\top \cdot \boldsymbol{b})) \tag{8}$$

for some $m \in \mathbb{N}_+$.

The resulting vector $\hat{\boldsymbol{b}} = \mathcal{C}_{\text{HarMo}}(\boldsymbol{b}, t)$ is subsequently used in downstream computations such as consensus updates or local gradient steps. This separation of analog quantization and digital encoding enables both precision control and bandwidth efficiency, allowing each component of $\boldsymbol{a}$ to be transmitted using exactly $m$ bits.

## 3.2 PERSISTENT EXCITATION CONDITION

The Persistent Excitation (PE) condition captures the idea that even when only a scalar projection of a high-dimensional vector is transmitted at each step, the sequence of projection directions must vary over time to ensure that all dimensions are sufficiently explored. Without such variation, certain components of the vector may be neglected, leading to biased or incomplete information. This concept has a natural connection with the Discrete Fourier Transform (DFT), which also represents signals through structured oscillatory components. The DFT, however, is defined on a finite time

window and transforms the entire signal within this limited horizon into complex-valued frequency components based on orthogonal bases. In contrast, PE relies on real-valued projections that evolve over time and can extend over an unbounded horizon. Whereas the DFT captures all information within its fixed window, PE ensures that the accumulated effect of projections over time, whether in a finite sliding window or over an infinite sequence, spans the full space even if individual directions are not orthogonal. This temporal coverage makes PE particularly suitable for sequential communication scenarios with compressed updates, enabling effective reconstruction of the original signal under communication constraints.

**Lemma 3.1.** *The HarMo sequence $\boldsymbol{\psi}_{HarMo}(t)$ is uniformly bounded and persistently excited, i.e.,*

$$\alpha_2 \mathbf{I}_d \geq \sum_{t=k}^{k+N-1} \boldsymbol{\psi}_{HarMo}(t) \cdot \boldsymbol{\psi}_{HarMo}(t)^\top \geq \alpha_1 \mathbf{I}_d, \quad \forall k \geq 0 \qquad (9)$$

*for $\alpha_1 = \alpha_2 = \frac{(2d-1)!}{2}$ and $N = (2d-1)!$.*

This result provides the theoretical foundation for using the HarMo sequence in compressed communication settings, ensuring that directional diversity is preserved over time despite transmitting only scalar information at each step. The detailed proof is provided in Appendix B.

## 4 LOG-BIT GRADIENT DESCENT WITH HARMONIC MODULATION

### 4.1 THE ALGORITHM

In this subsection, we propose a novel algorithm to address the communication bottleneck in fully decentralized federated learning. Specifically, we incorporate the HarMo into a distributed optimization framework. The resulting method, named **Log-Bit Gradient Descent with Har**monic **Mo**dulation (**LBGD-HarMo**), is summarized in Algorithm 1. In lines 6-7, inspired by the work in Kajiyama et al. (2021), the algorithm quantizes the error state and introduces a decaying coefficient $g_t$ to scale the transmitted value before and after quantization, reducing quantization error. In lines 8-10, The compression process involves encoding $\boldsymbol{\theta}$ into a single real number using the **HarMo Encoder** $\mathcal{C}_{\mathrm{E}}$, followed by quantizing and encoding this value into an $m$-bits binary number for communication. In lines 13-14, the **HarMo Decoder** $\mathcal{C}_{\mathrm{D}}$ reconstructs the transmitted value back to a real vector, allowing the local updates to be performed. Lines 16-17 introduce a distributed filter and a distributed integrator. The filter $\boldsymbol{\sigma}$ tracks the local state $\boldsymbol{x}$, while the integrator $\boldsymbol{z}$ tracks the term $(\kappa_0 g_t \hat{\boldsymbol{\theta}}_i - \kappa_0 g_t \sum_{j \in \mathcal{N}_i} a_{ij} \hat{\boldsymbol{\theta}}_j)$, which is used to balance local and global information. Our proposed algorithm shares a similar compression approach to other methods (Koloskova et al., 2020a; Liu et al., 2021; Yi et al., 2023; Islamov et al., 2025), in the sense of compressing and transmitting error states.

We introduce a weight matrix $[a_{ij}] \in \mathbb{R}^{n \times n}$ on $\mathcal{G} = (\mathcal{V}, \mathcal{E})$ that satisfies $a_{ij} > 0$ if $(j, i) \in \mathcal{E}$ and $a_{ij} = 0$ otherwise. The Laplacian matrix $\boldsymbol{L}$ is given by $[\boldsymbol{L}]_{ij} = -a_{ij}$ for $i \neq j$ and $[\boldsymbol{L}]_{ii} = \sum_{j=1}^{n} a_{ij}$. The neighbor set of node $i$ is $\mathcal{N}_i = \{j \in \mathcal{V} \mid [\boldsymbol{L}]_{ij} \neq 0\}$.

In Algorithm 1, the vector $\boldsymbol{x}_i := [x_i^1, \ldots, x_i^d]^\top \in \mathbb{R}^d$ represents model parameters of agent $i$. The global vector $\boldsymbol{x} := [\boldsymbol{x}_1; \ldots; \boldsymbol{x}_n] \in \mathbb{R}^{nd}$ collects the model parameters of all clients across the network. Let the parameters $\kappa, \kappa_0, \alpha, \eta > 0$ be step size and tuning constants. We denote the local loss function of agent $i$ as $f_i(\cdot)$ and its gradient by $\nabla f_i(\boldsymbol{x}_i) := \left[\frac{\partial f_i}{\partial x_i^1}, \ldots, \frac{\partial f_i}{\partial x_i^d}\right]^\top \in \mathbb{R}^d$. Only the $m$-bits binary messages $\mathcal{C}_{\mathrm{E}}\left(\frac{\boldsymbol{x}_{i,t} - \boldsymbol{\sigma}_{i,t}}{g_t}, t\right)$ are transmitted over the communication network.

To simplify the analysis, we define the extended gradient mapping as:

$$\mathcal{H}(\boldsymbol{x}) := [\nabla f_1(\boldsymbol{x}_1); \ldots; \nabla f_n(\boldsymbol{x}_n)] \in \mathbb{R}^{nd}, \quad \forall \boldsymbol{x} \in \mathbb{R}^{nd}. \qquad (10)$$

### 4.2 CONVERGENCE RESULT

We analyze the convergence behavior of Algorithm 1 (LBGD-HarMo) and establish the following theoretical guarantee. First, we impose the following assumptions for the later analysis.

---

**Algorithm 1** Log-Bit Gradient Descent with Harmonic Modulation (LBGD-HarMo)

---

1: **Input:** $T, \mathcal{G} = (\mathcal{V}, \mathcal{E}), \kappa, \kappa_0, \eta, \alpha, g_t, \boldsymbol{\sigma}_{i,t}, \boldsymbol{z}_{i,t}, \boldsymbol{x}_{i,t}, \forall i \in \mathcal{V}, 0 \leq t \leq T$

2: **Output:** $\boldsymbol{x}_{i,T+1}, \forall i \in \mathcal{V}$

3: **Initialize** $t \leftarrow 0, \kappa, \kappa_0, \eta, \alpha, g_0, \gamma > 0, \boldsymbol{\sigma}_{i,0} = \boldsymbol{z}_{i,0} = \boldsymbol{x}_{i,0} = \boldsymbol{0}_d$

4: **while** $t \leq T$ **do**

5:     **for all** clients $i \in \mathcal{V}$ **do**

6:         $g_t = g_0 \gamma^t$

7:         $\boldsymbol{\theta}_{i,t} = \dfrac{\boldsymbol{x}_{i,t} - \boldsymbol{\sigma}_{i,t}}{g_t}$

8:         $y_{i,t} = \boldsymbol{\psi}_{\text{HarMo}}(t)^\top \cdot \boldsymbol{\theta}_{i,t}$                   $\triangleleft$ compress $\boldsymbol{\theta}_{i,t}$ to a single real number

9:         $y_{i,t} = q_m(y_{i,t})$                   $\triangleleft$ quantify $y_{i,t}$ to a specific quantization error

10:        $y_{i,t} = \tilde{q}_m(y_{i,t})$                   $\triangleleft$ encode $y_{i,t}$ to $m$-bits binary numbers

11:        **for** neighbors $j \in \mathcal{N}_i$ (including $i$) **do**

12:            Transmit $y_{i,t}$ and receive $y_{j,t}$

13:            $y_{j,t} = \texttt{BinDecode}(y_{j,t})$         $\triangleleft$ decode $y_{i,t}$ to the original quantified number

14:            $\hat{\boldsymbol{\theta}}_{j,t} = \boldsymbol{\psi}_{\text{HarMo}}(t) \cdot y_{j,t}$           $\triangleleft$ reconstruct $y_{i,t}$ to a real vector

15:        **end for**

16:        $\boldsymbol{\sigma}_{i,t+1} = \boldsymbol{\sigma}_{i,t} + \kappa_0 g_t \hat{\boldsymbol{\theta}}_{i,t}$

17:        $\boldsymbol{z}_{i,t+1} = \boldsymbol{z}_{i,t} + \kappa_0 g_t \hat{\boldsymbol{\theta}}_{i,t} - \kappa_0 g_t \sum_{j \in \mathcal{N}_i} a_{ij} \hat{\boldsymbol{\theta}}_{j,t}$

18:        $\boldsymbol{x}_{i,t+1} = \boldsymbol{x}_{i,t} - \kappa \left[ \beta \left( \boldsymbol{\sigma}_{i,t} - \boldsymbol{z}_{i,t} \right) + \frac{\eta}{t+1} \nabla f_i(\boldsymbol{x}_{i,t}) \right]$    $\triangleleft$ update the local variable

19:     **end for**

20:     $t \leftarrow t + 1$

21: **end while**

---

**Assumption 4.1.** *The global cost function $F(\boldsymbol{x}) = \frac{1}{n} \sum_{i=1}^n f_i(\boldsymbol{x})$ is strongly convex, i.e., $F(\boldsymbol{x})$ satisfies*

$$F(\boldsymbol{y}) \geq F(\boldsymbol{x}) + \nabla F(\boldsymbol{x})^\top (\boldsymbol{y} - \boldsymbol{x}) + \frac{\mu}{2} \|\boldsymbol{y} - \boldsymbol{x}\|^2, \forall \boldsymbol{x}, \boldsymbol{y} \in \mathbb{R}^d,$$

*for some constant $\mu > 0$.*

**Assumption 4.2.** *The extended gradient mapping $\mathcal{H}(\cdot)$ in (10) is Lipschitz continuous, i.e.,*

$$\|\mathcal{H}(\boldsymbol{x}) - \mathcal{H}(\boldsymbol{x}')\| \leq L_\mathcal{H} \|\boldsymbol{x} - \boldsymbol{x}'\|, \quad \forall \boldsymbol{x}, \boldsymbol{x}' \in \mathbb{R}^{nd},$$

*for some constant $L_\mathcal{H} > 0$.*

**Assumption 4.3.** *The graph $\mathcal{G}$ is undirected, connectedand time-invariant.*

Note that if Assumption 4.3 holds, the Laplacian matrix $\boldsymbol{L}$ is symmetric positive semi-definite with eigenvalues $0 = \lambda_1 < \lambda_2 \leq \cdots \leq \lambda_n$ and $\boldsymbol{1}_n^\top \boldsymbol{L} = \boldsymbol{0}$ by Mesbahi & Egerstedt (2010).

**Theorem 4.1.** *Consider the DL problem (2) over a communication graph $\mathcal{G}$and suppose Assumptions 4.1-4.3 hold. Then, for some $\kappa, \kappa_0, \eta, \beta, g_0, \gamma > 0$ and sufficiently large bit-length*

$$m \geq \widetilde{m} = \mathcal{O}(\log_2(d)),$$

*the model parameter $\boldsymbol{x}_{i,t}$ of each client $i$ produced by Algorithm 1 (LBGD-HarMo) converges to a common model $\boldsymbol{x}^\star$ at a rate of $\mathcal{O}(1/t)$, i.e.,*

$$\|\boldsymbol{x}_{i,t} - \boldsymbol{x}^\star\| = \mathcal{O}(1/t),$$

*where $t$ denotes the iteration index.*

This theorem establishes the sublinear convergence rate of LBGD-HarMo under standard assumptions, providing a rigorous guarantee for its effectiveness. Notably, it highlights the fundamental role of logarithmic bit complexity in ensuring convergence of LBGD-HarMo, showing that only $\mathcal{O}(\log_2(d))$ bits are sufficient for reliable optimization. The complete proof is provided in Appendix C.

# 5 NUMERICAL RESULTS

## 5.1 EXPERIMENTAL SETUP

For all experiments, we evaluate each scheme in terms of convergence rate and communication complexity, reporting number of iterations and communication cost.

**Topologies.** The communication topologies considered in our experiments include the ring, torus, fully-connected networkand the complex network Erdős–Rényi (ER) graph.

**Compressors and quantizers.** In addition to our proposed HarMo, we also compare against the Top-$\alpha$ (Alistarh et al., 2018) and the Sign quantizer (Bernstein et al., 2018). The details of these two methods are provided in the Appendix.

**Algorithms.** We compare our proposed **LBGD-HarMo** with several representative baselines, including **DSGD** (Lian et al., 2017), **CHOCO** (Koloskova et al., 2020a)and **MoTEF** (Islamov et al., 2025). We also evaluate **LBGD** combined with **Sign** quantizer (**LBGD-Sign**), where the local model $\boldsymbol{\theta}_{i,t}$ is directly quantized using a standard Sign quantizer (Kajiyama et al., 2021) without additional compression. For all methods, the step sizes are carefully tuned to ensure fair convergenceand detailed hyperparameter configurations are provided in Appendix D. Additionally, we include **FedAvg** (McMahan et al., 2017) as a centralized baseline for reference in neural network training.

## 5.2 SYNTHETIC QUADRATIC OPTIMIZATION PROBLEM

We consider a synthetic quadratic optimization problem to demonstrate the validity of the theoretical results for Algorithm 1. This problem follows the classical setups in strongly convex optimization (Gao et al., 2024). For each client $i$, the local objective is defined as $f_i(\boldsymbol{x}_i) := \frac{1}{2}\|\boldsymbol{Q}_i\boldsymbol{x}_i - \boldsymbol{s}_i\|^2$, where $\boldsymbol{Q}_i = \frac{i^2}{n}\boldsymbol{I}_d \in \mathbb{R}^{d \times d}$ is positive definite to ensure strong convexityand $\boldsymbol{s}_i \sim \mathcal{N}\left(0, \frac{\rho^2}{i^2}\boldsymbol{I}_d\right)$ introduces heterogeneity through the linear component (Koloskova et al., 2020b). To be consistent with prior works, Gaussian noise with variance $\sigma^2$ is added to the gradients. The quadratic form guarantees a unique minimizerand the heterogeneity across clients arises from differences in $\boldsymbol{Q}_i$ and $\boldsymbol{s}_i$. We evaluate performance using two standard metrics: the *Optimality Error*, defined as $\frac{1}{n}\sum_{i=1}^{n}\|\boldsymbol{x}_{i,t} - \boldsymbol{x}^\star\|^2$ measuring the deviation from the optimal solution $\boldsymbol{x}^\star$and the *Consensus Error*, defined as $\frac{1}{n}\sum_{i=1}^{n}\|\boldsymbol{x}_{i,t} - \bar{\boldsymbol{x}}_t\|^2$ quantifying disagreement across clients, where $\bar{\boldsymbol{x}}_t = \frac{1}{n}\sum_{i=1}^{n}\boldsymbol{x}_{i,t}$ denotes the network average.

**Effect of number of clients and communication topologies.** Figure 1a and Figure 1b show that the number of clients has little effect on the performance of LBGD-HarMo. For network structures, the fully-connected topology achieves the best overall performance, as its dense connectivity minimizes consensus error and ensures more accurate results. These results confirm the robustness of LBGD-HarMo to both network size and topology.

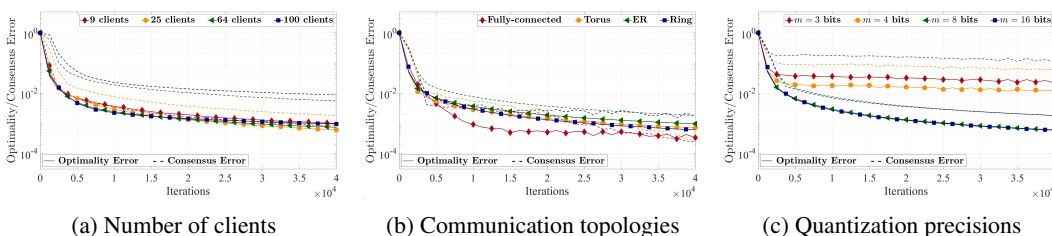

| (a) Number of clients | (b) Communication topologies | (c) Quantization precisions |

Figure 1: Convergence performance of synthetic quadratic optimization problem under different settings: (a) varying the number of clients $n$; (b) varying the communication topologies; (c) varying the quantization precision $m$. In these experiments, we fix the quantization precision to $m = 8$ in panels (a) and (b), use a ring topology in panels (a) and (c), and set the number of clients to 25 in panels (b) and (c), with dimension $d = 8$.

**Effect of quantization precisions.** In Figure 1c, we study the impact of different quantization precisions $m$ on the convergence of our proposed LBGD-HarMo algorithm. As the quantization

precision $m$ decreases, the number of communication cost is significantly reduced, thereby alleviating the communication burden across clients. However, in our experiments we found that $m = 3$ bits is the minimum precision that still ensures convergence, as lower precisions with higher quantization noise may lead to divergence. Moreover, the results with $m = 8$ and $m = 16$ bits are almost identical, indicating diminishing returns from further increasing precision. These findings are consistent with Theorem 4.1, which establishes convergence under finite but sufficiently large quantization levels.

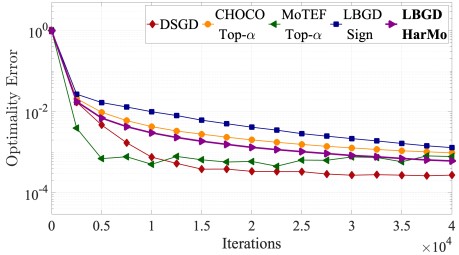 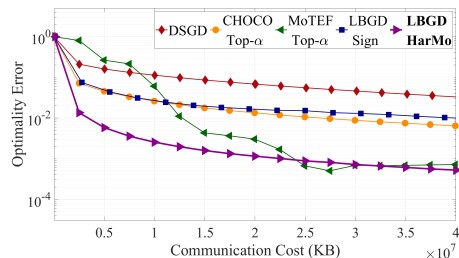

(a) Convergence of different algorithms  (b) Communication cost of different algorithms

Figure 2: Comparison of DSGD, CHOCO, MoTEF with Top-$\alpha$ ($\alpha = 0.125$), LBGD-Sign and LBGD-HarMo ($m = 8$ bits) on the synthetic quadratic optimization problem. The experiment is conducted with 25 clients connected over a ring topology, with detailed hyperparameter values provided in the appendix.

**Comparison against other algorithms.** As illustrated in Figure 2, LBGD-HarMo attains comparable convergence while clearly outperforming all baselines in terms of communication cost, achieving the same accuracy with far fewer transmitted bits. Although MoTEF exhibits linear speedup in the early stage, it still requires substantially more communication to reach higher-precision accuracy, highlighting the superior efficiency of LBGD-HarMo.

### 5.3 LOGISTIC REGRESSION WITH STRONGLY CONVEX REGULARIZER

We further evaluate our proposed Algorithm 1 on a logistic regression task with an $\ell_2$-regularizer. Specifically, the local objective function for each client $i$ is given by $f_i(\boldsymbol{x}_i) = \frac{1}{m_i} \sum_{j=1}^{m_i} \log \left(1 + \exp(-b_{ij} \boldsymbol{a}_{ij}^\top \boldsymbol{x}_i)\right) + \frac{1}{2m_i} \|\boldsymbol{x}_i\|_2^2$, where $\boldsymbol{a}_{ij} \in \mathbb{R}^d$ represents the feature vector of the $j$-th data sample on client $i$, $b_{ij} \in \{-1, 1\}$ is the corresponding labeland $m_i$ denotes the number of samples assigned to client $i$. To examine the effect of data heterogeneity, we adopt two distribution settings: (i) *IID*, where samples are uniformly and randomly assigned, so each client holds a representative subset of the dataset; (ii) *Non-IID*, where samples are unevenly partitioned such that each client mainly contains data from a limited set of classes, inducing statistical heterogeneity. Each experiment is repeated three timesand we evaluate the *Optimality Error* $f(\bar{\boldsymbol{x}}_t) - f(\boldsymbol{x}^\star)$, where $\bar{\boldsymbol{x}}_t = \frac{1}{n} \sum_{i=1}^n \boldsymbol{x}_{i,t}$ denotes the average model across all $n$ clients at iteration $t$and $f(\boldsymbol{x}^\star)$ is computed using the *LogisticRegression* from scikit-learn (Pedregosa et al., 2011). *Furthermore, we record the per-iteration Communication Cost per node and the final Test Accuracy and Runtime after training, providing a comprehensive evaluation of both communication efficiency and model performance.* We first compare our approach against several representative algorithms. In addition, we compare the accuracy across different numbers of clients, various network topologiesand both IID and Non-IID data distributions. The corresponding hyperparameter settings and experimental results are provided in Appendices D.4 and E.

**Datasets.** We conduct experiments on the *epsilon* dataset (Sonnenburg et al., 2008), a large-scale benchmark for binary classification consisting of 400,000 training samples, 100,000 validation samplesand 2,000 features. The dataset's high dimensionality and large sample size provide a rigorous testbed for communication-efficient algorithms.

**Comparison against other algorithms.** As shown in Figure 3a, LBGD with both the proposed HarMo and Sign quantizer achieves comparable convergence behavior to DSGD, CHOCOand MoTEF with Top-$\alpha$, while maintaining stable and consistent performance across runs. More importantly, when the communication cost is measured in terms of communication cost in Figure 3b,

LBGD-HarMo significantly outperforms the baselines. Table 2 indicates that LBGD-HarMo reduces the communication overhead to only $0.07$ KB, compared with $281.25$ KB for DSGD and $56.25$ KB for CHOCO and MoTEF with the Top-$\alpha$ compressor, even the LBGD-Sign still requires $8.93$ KB. These empirical findings are consistent with our theoretical guarantees, showing that the proposed approach substantially reduces communication cost while maintaining accuracy close to the baselines.

Table 1: Per-client communication cost per iteration and the corresponding test accuracies and runtime for different algorithms under various compressors and quantizers in the Logistic Regression experiment. Experiments are conducted using a Top-$\alpha$ compressor with $\alpha = 0.1$, Sign quantizer with 1 bitand HarMo with $m = 16$ bits, evaluated on $n = 9$ clients arranged in a ring topology under IID data distribution.

| Algorithm | Method | Communication cost (KB) | Test accuracy (%) | Runtime (s) |
|-----------|--------|-------------------------|-------------------|-------------|
| DSGD | None | 31.25 | 88.44 | 453.67 |
| CHOCO | TOP-$\alpha$ | 6.25 | 88.23 | 957.34 |
| MoTEF | TOP-$\alpha$ | 6.25 | 87.42 | 1497.01 |
| LBGD | Sign | 0.99 | 86.82 | 608.63 |
| **LBGD** | **HarMo** | **0.008** | **87.84** | **733.49** |

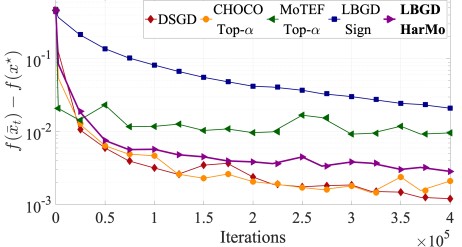

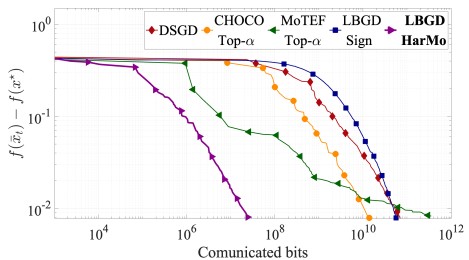

(a) Convergence of different algorithms  (b) Communication cost of different algorithms

Figure 3: Comparison of Algorithm 1 (LBGD-HarMo), CHOCO, MoTEF with the Top-$\alpha$ compressorand LBGD with the Sign quantizer on *epsilon* in terms of iterations and communication cost, which respectively indicate the convergence rates and the total number of communication cost needed to achieve the same accuracy.

## 5.4 NEURAL NETWORK TRAINING

We further evaluate the proposed LBGD algorithm on the standard image classification task. Each client employs a neural network as the local model and trains it using the standard cross-entropy loss. To simulate both IID and Non-IID data distributions across federating clients, we adopt the Dirichlet-based partitioning strategy $\mathrm{Dir}(p)$ (Hsu et al., 2019), where larger values of $p$ correspond to more balanced (IID) data splits, while smaller values produce stronger data heterogeneity (Non-IID). Following the same evaluation metrics as in Section 5.3, we record the *Training Loss*, the *per-iteration Communication Cost per node*, the *final Test Accuracy*and the *Runtime* after training, providing a comprehensive evaluation of both communication efficiency and model performance.

**Dataset.** We evaluate our method on the *CIFAR-10* dataset (Krizhevsky, 2009), which is a widely used benchmark for image classification. CIFAR-10 consists of 60,000 color images of size $32 \times 32$ across 10 classes, with 50,000 images for training and 10,000 for testing.

**Model.** We adopt ResNet-18 (He et al., 2016) as the backbone network for each client. ResNet-18 is a lightweight residual convolutional neural network consisting of 18 layers, including one initial convolutional layer, four residual stages with two BasicBlocks each, and a final fully connected layer.

**Comparison against other algorithms.** To comprehensively evaluate the effectiveness of our proposed method, we compare **LBGD-HarMo** with several representative baselines. For the central-

ized setting, we adopt **FedAvg** (McMahan et al., 2017) as a benchmark. For the decentralized federated learning setting, we employ two widely used algorithms: **DSGD** (Lian et al., 2017), representing the standard decentralized optimization framework, and **CHOCO** (Koloskova et al., 2020a), which integrates communication compression in decentralized learning. Each compression scheme is applied independently to every layer of ResNet-18 to ensure fair comparison. We evaluate the test accuracy on each client and report the averaged performance over all clients.

Table 2: Per-client communication cost per iteration and the corresponding total training accuracies and runtime for different algorithms under various compressors and quantizers in the Neural Network Training experiment. Experiments are conducted using a Top-$\alpha$ compressor with $\alpha = 0.1$, Sign quantizer with 1 bitand HarMo with $m = 24$ bits, evaluated on $n = 9$ clients arranged in a ring topology under IID data distribution.

| Algorithm | Method | Communication cost (MB) | Test accuracy (%) | Runtime (min) |
|---|---|---|---|---|
| FedAvg | None | 804.60 (Central Server) | 89.92 | 99 |
| DSGD | None | 178.80 | 90.09 | 84 |
| CHOCO | TOP-$\alpha$ | 35.76 | 87.03 | 209 |
| LBGD | Sign | 5.33 | 81.94 | 147 |
| **LBGD** | **HarMo** | **0.0001** | **86.69** | **203** |

From Table 2 and Figure 4, we can conclude that LBGD-HarMo requires considerably less communication cost compared to other algorithms and the training time for LBGD-HarMo is comparable to that of other compression algorithms. Furthermore, LBGD-HarMo achieves competitive test accuracy, meeting the performance standards while significantly reducing the communication overhead. These results suggest that LBGD-HarMo provides an effective balance between communication efficiency and model performance in decentralized neural network training.

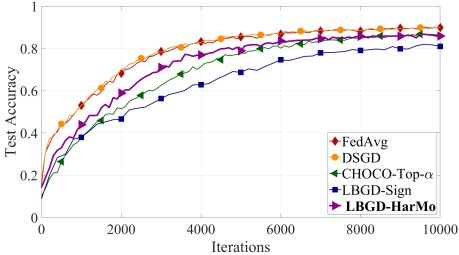 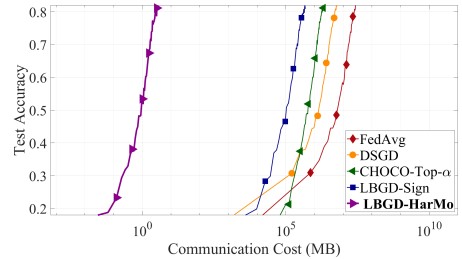

(a) Convergence of different algorithms   (b) Communication cost of different algorithms

Figure 4: Comparison of Algorithm 1 (LBGD-HarMo), FedAvg, DSGD, CHOCO with the Top-$\alpha$ compressorand LBGD with the Sign quantizer on the ResNet-18 training task over the *CIFAR-10* dataset, in terms of test accuracy and total communication cost.

## 6 CONCLUSIONS

In this paper, we proposed LBGD-HarMo, a novel log-bit quantization scheme with harmonic modulation for communication-efficient distributed learning over graphs. We developed provably convergent algorithm that compresses high-dimensional variables into log-bit transmissions while preserving convergence guarantees comparable to those of uncompressed methods. Both theoretical analysis and empirical results demonstrated that LBGD-HarMo substantially reduces communication cost. Furthermore, we showed that the bit-width $m$ can be tuned to trade off communication overhead against learning performance, with even small values of $m$ ensuring stable convergence.

A limitation of this work is that our analysis focuses on strongly convex objectivesand future research will involve deriving explicit expressions for the relationship between convergence rate and parameters in the strongly convex case. Additionally, extending the theoretical analysis of LBGD-HarMo to non-convex objectives remains an important direction for future work.

ETHICS STATEMENT

This work focuses on the development of decentralized optimization algorithms for federated learning. Our study is entirely theoretical and experimentaland does not involve human subjects, personally identifiable information, or sensitive data. The datasets used in our experiments are standard public benchmarks that are widely adopted in the machine learning community, ensuring compliance with privacy, fairnessand ethical standards. No harmful applications or misuse of the proposed methodology are foreseenand our code will be released to facilitate transparency, reproducibilityand future research. We confirm that this research adheres to the ICLR Code of Ethics.

REPRODUCIBILITY STATEMENT

All experiments are conducted on a server equipped with an Intel(R) Xeon(R) Platinum 8336C CPU @ 2.30GHz (32 cores, 2 threads per core) and nine NVIDIA GeForce RTX 4090 GPUs. The synthetic quadratic optimization experiments were implemented in MATLAB R2024a, while the logistic regression experiments were implemented in PYTHON 3.8. To ensure reproducibility, we provide an anonymous GitHub repository containing all source codes and scripts necessary to replicate our results. Our implementation is based on open-source code from (Koloskova et al., 2020a) `https://github.com/epfml/ChocoSGD` and is available at `https://anonymous.4open.science/r/LBGD-HarMo`.

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

## A  NOTATION

In this paper, $\|\cdot\|$ denotes the Euclidean norm. The notation $\mathbf{1}_n$ ($\mathbf{0}_n$), $\mathbf{1}_{n \times d}$ ($\mathbf{0}_{n \times d}$), $\boldsymbol{I}_n$ and $\{\boldsymbol{e}_1, \ldots, \boldsymbol{e}_d\}$ denote the one (zero) column, the one (zero) matrix, identity matrixand base vectors in $\mathbb{R}^d$, respectively. The expression $\mathrm{blkdiag}(\boldsymbol{x}_1, \ldots, \boldsymbol{x}_n)$ is a diagonal matrix with the $i$-th diagonal matrix being $\boldsymbol{x}_i$. The symbol $\otimes$ denotes the Kronecker product, $\odot$ denotes the Hadamard productand $\lceil \cdot \rceil$ denotes the ceiling operator. For a differentiable function, $\nabla(\cdot)$ denotes its gradient. For column vectors $\boldsymbol{a}$ and $\boldsymbol{b}$, $[\boldsymbol{a}; \boldsymbol{b}]$ means $\left[\boldsymbol{a}^\top, \boldsymbol{b}^\top\right]^\top$. The notation $\mathcal{O}(\cdot)$ means the magnitude notation.

## B  PROOF OF LEMMA 3.1. —— HARMO SEQUENCE SATISFIES THE PERSISTENT EXCITATION (PE) CONDITION

**Lemma 3.1.** The HarMo sequence $\boldsymbol{\psi}_{\mathrm{HarMo}}(t)$ is uniformly bounded and persistently excited, i.e.,

$$\alpha_2 \mathbf{I}_d \geq \sum_{t=k}^{k+N-1} \boldsymbol{\psi}_{\mathrm{HarMo}}(t) \cdot \boldsymbol{\psi}_{\mathrm{HarMo}}(t)^\top \geq \alpha_1 \mathbf{I}_d, \quad \forall k \geq 0$$

for $\alpha_1 = \alpha_2 = \frac{(2d-1)!}{2}$ and $N = (2d-1)!$.

*Proof.* Let the dimension be $d \in \mathbb{N}$ and define the Harmonic Modulation (HarMo) sequence $\psi_{\text{HarMo}}(t) \in \mathbb{R}^d$ as

$$\psi_{\text{HarMo}}(t) = \left[\sin\left(\frac{\pi t}{d+1}\right), \ \sin\left(\frac{2\pi t}{d+1}\right), \ \ldots, \ \sin\left(\frac{d\pi t}{d+1}\right)\right]^\top. \tag{B.1}$$

We aim to show that $\{\psi_{\text{HarMo}}(t)\}$ satisfies the persistent excitation (PE) condition

$$\alpha_2 \mathbf{I}_d \succeq \sum_{t=k}^{k+N-1} \psi_{\text{HarMo}}(t)\psi_{\text{HarMo}}(t)^\top \succeq \alpha_1 \mathbf{I}_d, \quad \forall k \geq 0 \tag{B.2}$$

for some $0 < \alpha_1 \leq \alpha_2$ and all integers $N \geq N_0$.

Consider the accumulated Gram matrix

$$M_N(k) := \sum_{t=k}^{k+N-1} \psi_{\text{HarMo}}(t)\psi_{\text{HarMo}}(t)^\top.$$

Its $(i,j)$-entry can be expressed as

$$[M_N(k)]_{i,j} = \sum_{t=k}^{k+N-1} \sin\left(\frac{i\pi t}{d+1}\right)\sin\left(\frac{j\pi t}{d+1}\right).$$

Applying the trigonometric identity $\sin(a)\sin(b) = \frac{1}{2}[\cos(a-b) - \cos(a+b)]$, we obtain

$$[M_N(k)]_{i,j} = \frac{1}{2}\sum_{t=k}^{k+N-1}\left[\cos\left(\frac{(i-j)\pi t}{d+1}\right) - \cos\left(\frac{(i+j)\pi t}{d+1}\right)\right].$$

When $i \neq j$, both cosine terms are periodic with integer multiples of $\frac{2(d+1)}{|i-j|}$ and $\frac{2(d+1)}{i+j}$, respectively. By choosing $N$ as a common multiple of these periods, the summation vanishes and hence $[M_N(k)]_{i,j} = 0$. This shows that the Gram matrix is diagonal.

For $i = j$, one has

$$[M_N(k)]_{i,i} = \sum_{t=k}^{k+N-1} \sin^2\left(\frac{i\pi t}{d+1}\right).$$

Since $\sin^2(x)$ has average value $\frac{1}{2}$ over its period, taking $N$ as a multiple of the fundamental period $\frac{d+1}{i}$ yields

$$[M_N(k)]_{i,i} = \frac{N}{2}.$$

Thus, all diagonal entries coincide and the Gram matrix satisfies

$$M_N(k) = \frac{N}{2}\mathbf{I}_d.$$

Consequently, the HarMo sequence satisfies the PE condition (B.2) with $\alpha_1 = \alpha_2 = \frac{N}{2}$. A conservative universal choice of $N$ can be made by taking the least common multiple of all possible periods,

$$N = \text{lcm}\left\{\frac{2(d+1)}{s} : \ s = 1, 2, \ldots, 2d-1\right\} = 2(d+1)\cdot\text{lcm}(1, 2, \ldots, 2d-1),$$

which can be upper bounded by $(2d-1)!$. Therefore, the PE condition holds with constants $\alpha_1 = \alpha_2 = \frac{N}{2} = \frac{(2d-1)!}{2}$. $\qquad\square$

## C  PROOF OF THEOREM 4.1. —— CONVERGENCE RATES OF LBGD-HARMO

**Theorem 4.1.** Consider the DL problem over a communication graph $\mathcal{G}$ and suppose Assumptions 4.1-4.3 hold. Then, for some $\kappa, \eta, g_0, \gamma > 0$ and sufficiently large bit-length

$$m \geq \widetilde{m} = \mathcal{O}(\log_2(d)),$$

the model parameters $\boldsymbol{x}_{i,t}$ of each client $i$ produced by Algorithm 1 (LBGD-HarMo) converge to a common model $\boldsymbol{x}^\star$ at a rate of $\mathcal{O}(1/t)$, i.e.,

$$\|\boldsymbol{x}_{i,t} - \boldsymbol{x}^\star\| = \mathcal{O}(1/t),$$

where $t$ denotes the iteration index.

*Proof.* As illustrated in Algorithm 1, lines 16–18 can be rewritten as follows:

$$\boldsymbol{\sigma}_{i,t+1} = \boldsymbol{\sigma}_{i,t} + \kappa_0 g_t \, \mathcal{C}_{\text{HarMo}}\left(\frac{\boldsymbol{x}_{i,t} - \boldsymbol{\sigma}_{i,t}}{g_t}, t\right),$$

$$\boldsymbol{z}_{i,t+1} = \boldsymbol{z}_{i,t} + \kappa_0 g_t \, \mathcal{C}_{\text{HarMo}}\left(\frac{\boldsymbol{x}_{i,t} - \boldsymbol{\sigma}_{i,t}}{g_t}, t\right) - \kappa_0 g_t \sum_{j \in \mathcal{N}_i} \mathcal{C}_{\text{HarMo}}\left(\frac{\boldsymbol{x}_{j,t} - \boldsymbol{\sigma}_{j,t}}{g_t}, t\right), \quad \text{(C.1)}$$

$$\boldsymbol{x}_{i,t+1} = \boldsymbol{x}_{i,t} - \kappa\left[\beta(\boldsymbol{\sigma}_{i,t} - \boldsymbol{z}_{i,t}) + \tfrac{\eta}{t+1}\nabla f_i(\boldsymbol{x}_{i,t})\right],$$

$$g_t = g_0\gamma^t.$$

where $\mathcal{C}_{\text{HarMo}}(\frac{\boldsymbol{x}_{j,t} - \boldsymbol{\sigma}_{j,t}}{g_t}, t) = \mathcal{C}_{\text{D}}(\mathcal{C}_{\text{E}}(\frac{\boldsymbol{x}_{j,t} - \boldsymbol{\sigma}_{j,t}}{g_t}, t), t) = \boldsymbol{\psi}_{\text{HarMo}}(t) \cdot (q_m(\boldsymbol{\psi}_{\text{HarMo}}(t)^\top \cdot \frac{\boldsymbol{x}_{j,t} - \boldsymbol{\sigma}_{j,t}}{g_t}))$.

By recalling the relation established in Yi et al. (2022), we have

$$\boldsymbol{\sigma}_{i,t} - \boldsymbol{z}_{i,t} = \sum_{j \in \mathcal{N}_i} L_{ij}\,\boldsymbol{\sigma}_{j,t},$$

where $L_{ij}$ denotes the $(i,j)$-th entry of the graph Laplacian matrix.

For completeness, we briefly sketch the derivation. From the update rules in Algorithm 1, we subtract the two updates and obtain

$$\boldsymbol{\sigma}_{i,t+1} - \boldsymbol{z}_{i,t+1} = (\boldsymbol{\sigma}_{i,t} - \boldsymbol{z}_{i,t}) + \kappa_0 g_t \sum_{j \in \mathcal{N}_i} \mathcal{C}_{\text{HarMo}}\left(\frac{\boldsymbol{x}_{j,t} - \boldsymbol{\sigma}_{j,t}}{g_t}, t\right).$$

Since the update of $\boldsymbol{\sigma}_{j,t}$ satisfies

$$\boldsymbol{\sigma}_{j,t+1} = \boldsymbol{\sigma}_{j,t} + \kappa_0 g_t \, \mathcal{C}_{\text{HarMo}}\left(\frac{\boldsymbol{x}_{j,t} - \boldsymbol{\sigma}_{j,t}}{g_t}, t\right),$$

we have

$$\kappa_0 g_t \, \mathcal{C}_{\text{HarMo}}\left(\frac{\boldsymbol{x}_{j,t} - \boldsymbol{\sigma}_{j,t}}{g_t}, t\right) = \boldsymbol{\sigma}_{j,t+1} - \boldsymbol{\sigma}_{j,t}.$$

Substituting this identity gives

$$\boldsymbol{\sigma}_{i,t+1} - \boldsymbol{z}_{i,t+1} = (\boldsymbol{\sigma}_{i,t} - \boldsymbol{z}_{i,t}) + \sum_{j \in \mathcal{N}_i} (\boldsymbol{\sigma}_{j,t+1} - \boldsymbol{\sigma}_{j,t}). \quad \text{(C.2)}$$

By telescoping the recursion equation (C.2) from $s = 0$ to $t - 1$, we obtain

$$\boldsymbol{\sigma}_{i,t} - \boldsymbol{z}_{i,t} = (\boldsymbol{\sigma}_{i,0} - \boldsymbol{z}_{i,0}) + \sum_{s=0}^{t-1}\sum_{j \in \mathcal{N}_i} (\boldsymbol{\sigma}_{j,s+1} - \boldsymbol{\sigma}_{j,s})$$

$$= (\boldsymbol{\sigma}_{i,0} - \boldsymbol{z}_{i,0}) - \sum_{j \in \mathcal{N}_i} \boldsymbol{\sigma}_{j,0} + \sum_{j \in \mathcal{N}_i} \boldsymbol{\sigma}_{j,t}.$$

With the standard initialization $\boldsymbol{\sigma}_{i,0} = \boldsymbol{z}_{i,0} = \boldsymbol{0}$, this simplifies to

$$\boldsymbol{\sigma}_{i,t} - \boldsymbol{z}_{i,t} = \sum_{j \in \mathcal{N}_i} \boldsymbol{\sigma}_{j,t}.$$

Finally, by adopting the Laplacian notation, we can equivalently rewrite this relation as

$$\boldsymbol{\sigma}_{i,t} - \boldsymbol{z}_{i,t} = \sum_{j \in \mathcal{N}_i} L_{ij}\,\boldsymbol{\sigma}_{j,t}. \tag{C.3}$$

Combining the relation in (C.3) with the update rule (C.1), the iteration can be equivalently expressed in the following compact form:

$$\boldsymbol{\sigma}_{t+1} = \boldsymbol{\sigma}_t + \kappa_0 \widetilde{\boldsymbol{C}}_{\text{HarMo}}(\boldsymbol{x}_t - \boldsymbol{\sigma}_t, t) + \kappa_0 g_t \boldsymbol{\delta}_t,$$

$$\boldsymbol{x}_{t+1} = \boldsymbol{x}_t - \kappa \left[ \beta \mathcal{L} \boldsymbol{\sigma}_t + \frac{\eta}{t+1} \mathcal{H}(\boldsymbol{x}_t) \right], \tag{C.4}$$

$$g_t = g_0 \gamma^t,$$

where $\widetilde{\boldsymbol{C}}_{\text{HarMo}}(\boldsymbol{x}_t, t) := \left[ \boldsymbol{\psi}_{\text{HarMo}}(t)\boldsymbol{\psi}_{\text{HarMo}}(t)^\top \boldsymbol{x}_{1,t}; \ldots ; \boldsymbol{\psi}_{\text{HarMo}}(t)\boldsymbol{\psi}_{\text{HarMo}}(t)^\top \boldsymbol{x}_{n,t} \right] \in \mathbb{R}^{nd}, \boldsymbol{\delta}_t :=$ $\boldsymbol{C}_{\text{HarMo}} \left( \frac{\boldsymbol{x}_t - \boldsymbol{\sigma}_t}{g_t}, t \right) - \widetilde{\boldsymbol{C}}_{\text{HarMo}} \left( \frac{\boldsymbol{x}_t - \boldsymbol{\sigma}_t}{g_t}, t \right) \in \mathbb{R}^{nd}, \mathcal{L} := \boldsymbol{L} \otimes \boldsymbol{I}_d \in \mathbb{R}^{nd \times nd}$ and $\mathcal{H}(\boldsymbol{x}) :=$ $[\nabla f_1(\boldsymbol{x}_1); \ldots ; \nabla f_n(\boldsymbol{x}_n)] \in \mathbb{R}^{nd}$. Notably, (C.4) is obtained by noting that

$$g_t \widetilde{\boldsymbol{C}}_{\text{HarMo}} \left( \frac{\boldsymbol{x}_t - \boldsymbol{\sigma}_t}{g_t}, t \right) = \widetilde{\boldsymbol{C}}_{\text{HarMo}} \left( \boldsymbol{x}_t - \boldsymbol{\sigma}_t, t \right).$$

By Scutari et al. (2014) , as Assumptions 4.1–4.3, there exists a unique solution $\boldsymbol{x}^\star \in \mathbb{R}^d$ such that $\mathcal{H}(\boldsymbol{1}_n \otimes \boldsymbol{x}^\star) = \boldsymbol{0}_{nd}$. To facilitate the analysis, we introduce the state error variables $\bar{\boldsymbol{\sigma}}_t :=$ $\boldsymbol{\sigma}_t - \boldsymbol{1}_n \otimes \boldsymbol{x}^\star, \bar{\boldsymbol{x}}_t := \boldsymbol{x}_t - \boldsymbol{1}_n \otimes \boldsymbol{x}^\star$, which represent the deviations of $\boldsymbol{\sigma}_t$ and $\boldsymbol{x}_t$ from the steady-state solution $\boldsymbol{x}^\star$. Substituting these definitions into the update rules, we obtain the following equivalent system:

$$\bar{\boldsymbol{\sigma}}_{t+1} = \bar{\boldsymbol{\sigma}}_t + \kappa_0 \widetilde{\boldsymbol{C}}_{\text{HarMo}}(\bar{\boldsymbol{x}}_t - \bar{\boldsymbol{\sigma}}_t, t) + \kappa_0 g_t \boldsymbol{\delta}_t,$$

$$\bar{\boldsymbol{x}}_{t+1} = \bar{\boldsymbol{x}}_t - \kappa \left[ \beta \mathcal{L} \bar{\boldsymbol{\sigma}}_t + \frac{\eta}{t+1} \overline{\mathcal{H}}(\bar{\boldsymbol{x}}_t) \right], \tag{C.5}$$

$$g_t = g_0 \gamma^t,$$

where $\overline{\mathcal{H}}(\bar{\boldsymbol{x}}_t) := \mathcal{H}(\boldsymbol{x}_t) - \mathcal{H}(\boldsymbol{1}_n \otimes \boldsymbol{x}^\star)$.

To analyze the convergence of the system, we introduce a projection-based decomposition of the state variables. Let $\boldsymbol{S} \in \mathbb{R}^{n \times (n-1)}$ be a matrix whose rows are eigenvectors corresponding to the nonzero eigenvalues of the graph Laplacian $\boldsymbol{L}$, and define the projection operators $\mathcal{S} := \boldsymbol{S} \otimes \boldsymbol{I}_d, \mathcal{I} :=$ $\frac{1}{\sqrt{n}} \boldsymbol{1}_n \otimes \boldsymbol{I}_d$. By construction, these satisfy $\mathcal{S}^\top \mathcal{I} = \boldsymbol{0}_{(n-1)d \times d}$ and $\mathcal{S}\mathcal{S}^\top + \mathcal{I}\mathcal{I}^\top = \boldsymbol{I}_{nd}$. Then, for the state errors $\bar{\boldsymbol{x}}_t$, we introduce the decomposition $\bar{\boldsymbol{x}}_t^\perp := \mathcal{S}^\top \bar{\boldsymbol{x}}_t \in \mathbb{R}^{(n-1)d}, \bar{\boldsymbol{x}}_t^\| := \mathcal{I}^\top \bar{\boldsymbol{x}}_t \in \mathbb{R}^d$, so that

$$\bar{\boldsymbol{x}}_t = \mathcal{S}\bar{\boldsymbol{x}}_t^\perp + \mathcal{I}\bar{\boldsymbol{x}}_t^\|. \tag{C.6}$$

Then it follows that the convergence of $\bar{\boldsymbol{x}}_t$ can be established by showing that both its consensus component $\bar{\boldsymbol{x}}_t^\|$ and disagreement component $\bar{\boldsymbol{x}}_t^\perp$ converge to the zero equilibrium, respectively.

With the decomposition in (C.5) and the fact that $\mathcal{L}\mathcal{I} = \boldsymbol{0}_{nd \times d}$ and $\mathcal{I}^\top \mathcal{L} = \boldsymbol{0}_{d \times nd}$, we obtain the following equivalent dynamics:

$$\bar{\boldsymbol{\sigma}}_{t+1} = \bar{\boldsymbol{\sigma}}_t + \kappa_0 \widetilde{\boldsymbol{C}}_{\text{HarMo}}(\bar{\boldsymbol{x}}_t - \bar{\boldsymbol{\sigma}}_t, t) + \kappa_0 g_t \boldsymbol{\delta}_t,$$

$$\bar{\boldsymbol{x}}_{t+1}^\perp = \bar{\boldsymbol{x}}_t^\perp - \kappa \beta \mathcal{S}^\top \mathcal{L} \bar{\boldsymbol{\sigma}}_t - \frac{\kappa \eta}{t+1} \mathcal{S}^\top \overline{\mathcal{H}}(\bar{\boldsymbol{x}}_t),$$

$$\bar{\boldsymbol{x}}_{t+1}^\| = \bar{\boldsymbol{x}}_t^\| - \frac{\kappa \eta}{t+1} \mathcal{I}^\top \overline{\mathcal{H}}(\bar{\boldsymbol{x}}_t), \tag{C.7}$$

$$g_t = g_0 \gamma^t.$$

Besides, we can also abtain that

$$\|\mathcal{L}\bar{\boldsymbol{\sigma}}_t\|^2 \le 2\lambda_n^2 \|\bar{\boldsymbol{x}}_t - \bar{\boldsymbol{\sigma}}_t\|^2 + 2\lambda_n^2 \|\bar{\boldsymbol{x}}_t^\perp\|^2. \tag{C.8}$$

To separate the consensus and disagreement components, we define the projected error variables as $\bar{\boldsymbol{\sigma}}_t^\perp := \mathcal{S}^\top \bar{\boldsymbol{\sigma}}_t \in \mathbb{R}^{(n-1)d}$. Correspondingly, the variations of these projected variables are defined as

$$\Delta \bar{\boldsymbol{\sigma}}_t^\perp := \bar{\boldsymbol{\sigma}}_{t+1}^\perp - \bar{\boldsymbol{\sigma}}_t^\perp = \kappa_0 \widetilde{C}_{\text{HarMo}}(\bar{\boldsymbol{x}}_t^\perp - \bar{\boldsymbol{\sigma}}_t^\perp, t) + \kappa_0 g_t \mathcal{S}^\top \boldsymbol{\delta}_t,$$

$$\Delta \bar{\boldsymbol{x}}_t^\perp := \bar{\boldsymbol{x}}_{t+1}^\perp - \bar{\boldsymbol{x}}_t^\perp = -\kappa[\beta \mathcal{S}^\top \mathcal{L}\bar{\boldsymbol{\sigma}}_t + \frac{\eta}{t+1}\mathcal{S}^\top \mathcal{H}(\bar{\boldsymbol{x}}_t)],$$

$$\Delta \bar{\boldsymbol{x}}_t^\parallel := \bar{\boldsymbol{x}}_{t+1}^\parallel - \bar{\boldsymbol{x}}_t^\parallel = -\frac{\kappa\eta}{t+1}\mathcal{I}^\top \overline{\mathcal{H}}(\bar{\boldsymbol{x}}_t^\parallel) + \frac{\kappa\eta}{t+1}\mathcal{I}^\top(\overline{\mathcal{H}}(\bar{\boldsymbol{x}}_t^\parallel) - \overline{\mathcal{H}}(\bar{\boldsymbol{x}}_t)) \tag{C.9}$$

$$g_t = g_0 \gamma^t.$$

Now we are ready to propose Lyapunov functions for system (C.7). Define $V_{1,t} = \frac{1}{2}\|\bar{\boldsymbol{x}}_t^\perp\|^2$, then

$$V_{1,t+1} - V_{1,t} = \frac{1}{2}\|\bar{\boldsymbol{x}}_{t+1}^\perp\|^2 - \frac{1}{2}\|\bar{\boldsymbol{x}}_t^\perp\|^2$$

$$= \frac{1}{2}\|\bar{\boldsymbol{x}}_t^\perp - \kappa\beta \mathcal{S}^\top \mathcal{L}\bar{\boldsymbol{\sigma}}_t - \frac{\kappa\eta}{t+1}\mathcal{S}^\top \overline{\mathcal{H}}(\bar{\boldsymbol{x}}_t)\|^2 - \frac{1}{2}\|\bar{\boldsymbol{x}}_t^\perp\|^2$$

$$\le \left( -\frac{1}{2}\kappa\beta\lambda_2\|\bar{\boldsymbol{x}}_t^\perp\|^2 + \frac{1}{2}\kappa\beta\lambda_n\|\bar{\boldsymbol{\sigma}}_t - \bar{\boldsymbol{x}}_t\|^2 \right.$$

$$\left. + \frac{1}{2}\frac{\kappa\eta}{t+1}(1 + L_{\mathcal{H}}^2)\|\bar{\boldsymbol{x}}_t^\perp\|^2 + \frac{1}{2}\frac{\kappa\eta}{t+1}L_{\mathcal{H}}^2\|\bar{\boldsymbol{x}}_t^\parallel\|^2 \right)$$

$$+ \left( \kappa^2\beta^2\|\mathcal{L}\boldsymbol{\sigma}_t\|^2 + (\frac{\kappa\eta}{t+1})^2 L_{\mathcal{H}}^2(\|\bar{\boldsymbol{x}}_t^\perp\|^2 + \|\bar{\boldsymbol{x}}_t^\parallel\|^2) \right) \tag{C.10}$$

$$\le \left( -\frac{1}{2}\kappa\beta\lambda_2\|\bar{\boldsymbol{x}}_t^\perp\|^2 + \frac{1}{2}\kappa\beta\lambda_n\|\bar{\boldsymbol{\sigma}}_t - \bar{\boldsymbol{x}}_t\|^2 \right.$$

$$\left. + \frac{1}{2}\frac{\kappa\eta}{t+1}(1 + L_{\mathcal{H}}^2)\|\bar{\boldsymbol{x}}_t^\perp\|^2 + \frac{1}{2}\frac{\kappa\eta}{t+1}L_{\mathcal{H}}^2\|\bar{\boldsymbol{x}}_t^\parallel\|^2 \right)$$

$$+ \left( 2\kappa^2\beta^2\lambda_n\|\bar{\boldsymbol{x}}_t - \bar{\boldsymbol{\sigma}}_t\|^2 + 2\kappa^2\beta^2\lambda_n\|\bar{\boldsymbol{x}}_t^\perp\|^2 \right.$$

$$\left. + (\frac{\kappa\eta}{t+1})^2 L_{\mathcal{H}}^2(\|\bar{\boldsymbol{x}}_t^\perp\|^2 + \|\bar{\boldsymbol{x}}_t^\parallel\|^2) \right),$$

where the first inequality is obtained by

$$\mathcal{L}\bar{\boldsymbol{x}}_t = \mathcal{L}(\mathcal{S}\mathcal{S}^\top + \mathcal{I}\mathcal{I}^\top)\bar{\boldsymbol{x}}_t = \mathcal{L}\mathcal{S}\bar{\boldsymbol{x}}_t^\perp,$$

$$\|\overline{\mathcal{H}}(\bar{\boldsymbol{x}}_t)\|^2 \le L_{\mathcal{H}}^2\|\bar{\boldsymbol{x}}_t\|^2 = L_{\mathcal{H}}^2\left( \|\bar{\boldsymbol{x}}_t^\perp\|^2 + \|\bar{\boldsymbol{x}}_t^\parallel\|^2 \right), \tag{C.11}$$

derived from Assumption 4.2 and the last inequality is obtained by (C.8).

Define $V_{2,t} := \frac{1}{2}\|\bar{\boldsymbol{x}}_t^\parallel\|^2$, then

$$V_{2,t+1} - V_{2,t} = \frac{1}{2}\|\bar{\boldsymbol{x}}_{t+1}^\parallel\|^2 - \frac{1}{2}\|\bar{\boldsymbol{x}}_t^\parallel\|^2$$

$$= \frac{1}{2}\|\bar{\boldsymbol{x}}_t^\parallel - \frac{\kappa\eta}{t+1}\mathcal{I}^\top \overline{\mathcal{H}}(\bar{\boldsymbol{x}}_t)\|^2 - \frac{1}{2}\|\bar{\boldsymbol{x}}_t^\parallel\|^2$$

$$\le \left( -\frac{\kappa\eta}{t+1}(\mathcal{I}\bar{\boldsymbol{x}}_t^\parallel)^\top[\mathcal{H}(\bar{\boldsymbol{x}}_t + \mathbf{1}_n \otimes \boldsymbol{x}^\star) \right.$$

$$- \mathcal{H}(\mathcal{I}\bar{\boldsymbol{x}}_t^\parallel + \mathbf{1}_n \otimes \boldsymbol{x}^\star) + \mathcal{H}(\mathcal{I}\bar{\boldsymbol{x}}_t^\parallel + \mathbf{1}_n \otimes \boldsymbol{x}^\star) \tag{C.12}$$

$$\left. - \mathcal{H}(\mathbf{1}_n \otimes \boldsymbol{x}^\star)]) + (\frac{\kappa\eta}{t+1})^2\|\overline{\mathcal{H}}(\bar{\boldsymbol{x}}_t)\|^2 \right.$$

$$\le \left( -\frac{\kappa\eta}{t+1}\|\bar{\boldsymbol{x}}_t^\parallel\|^2 + \frac{\kappa\eta}{t+1}\|\bar{\boldsymbol{x}}_t^\perp\|^2 \right)$$

$$+ (\frac{\kappa\eta L_{\mathcal{H}}}{t+1})^2(\|\bar{\boldsymbol{x}}_t^\perp\|^2 + \|\bar{\boldsymbol{x}}_t^\parallel\|^2),$$

where the second inequality is obtained by (C.6) and (C.11).

By recalling Anderson (1977), we know $\bar{\boldsymbol{x}}_{e,t+1} = \bar{\boldsymbol{x}}_{e,t} - \kappa_0 \widetilde{C}_{\text{HarMo}}(\bar{\boldsymbol{x}}_{e,t}, t)$ is uniformly globally linearly stable for some $\kappa_0 > 0$, then there exist positive constants $C$, $\gamma_D < 1$ such that for any $t$ and $N \in \mathbb{N}_+$, the solution satisfies

$$\left( \| \boldsymbol{x}_e \left( t + N \right) \|^2 \right) \leq C \left( \| \boldsymbol{x}_e(t) \|^2 \right) \gamma_D^N.$$

We assume $\phi_t^{t+T} \left( \boldsymbol{x}_e(t) \right)$ is the state of the system $\boldsymbol{x}_e \left( t + 1 \right) = \boldsymbol{x}_e(t) - \kappa_0 \Lambda \widetilde{C}_{\text{HarMo}} \left( \boldsymbol{x}_e(t), t \right)$ in $t + T$ moment for any $0 \leq T \leq N$ with the state in $t$ moment is $\boldsymbol{x}_e(t)$. It is easy to verify that there exists some $L_\phi > 0$ that $\| \phi_t^{t+T} \left( \boldsymbol{x} \right) \|^2 \leq L_\phi \| \boldsymbol{x} \|^2$ holds for any $\boldsymbol{x} \in \mathbb{R}^{(n-1)d}$ and $0 \leq T \leq N$.

We define a Lyapunov function $V_{e,t} \left( \boldsymbol{x}_e, t \right) := \sum_{j=0}^{N-1} \| \phi_t^{t+j} \left( \boldsymbol{x}_e \right) \|^2$ satisfying

$$c_1 \| \boldsymbol{x}_e \|^2 \leq V_{e,t} \leq c_2 \| \boldsymbol{x}_e \|^2 \tag{C.13}$$

for $c_1 = 1, c_2 = N L_\phi$.

In addition, we have

$$
\begin{aligned}
\Delta V_{e,t} &= \sum_{j=1}^{N} \| \phi_{t+1}^{t+j} \left( \boldsymbol{x}_e \left( t + 1 \right) \right) \|^2 - \sum_{j=0}^{N-1} \| \phi_t^{t+j} \left( \boldsymbol{x}_e(t) \right) \|^2 \\
&= \| \boldsymbol{x}_e \left( t + N \right) \|^2 - \| \boldsymbol{x}_e(t) \|^2 \\
&\leq - \left( 1 - C \gamma_D^N \right) \| \boldsymbol{x}_e(t) \|^2 \leq -c_3 \| \boldsymbol{x}_e(t) \|^2
\end{aligned}
\tag{C.14}
$$

We choose a $N \in \mathbb{N}_+$ large enough and then $c_3 := 1 - C \gamma_D^N > 0$, i.e.,

$$
\begin{aligned}
&\sum_{j=1}^{N} \| \phi_{t+1}^{t+j} \left( \boldsymbol{x}_e - \hat{\kappa} \Lambda \widetilde{C}_{\text{HarMo}} \left( \boldsymbol{x}_e, t \right) \right) \|^2 - \sum_{j=0}^{N-1} \| \phi_t^{t+j} \left( \boldsymbol{x}_e \right) \|^2 \\
&\leq -c_3 \| \boldsymbol{x}_e \|^2.
\end{aligned}
\tag{C.15}
$$

In addition, we have

$$\| \boldsymbol{x}_e - \kappa_0 \Lambda \widetilde{C}_{\text{HarMo}} \left( \boldsymbol{x}_e, t \right) \|^2 \leq \theta \| \boldsymbol{x}_e \|^2, \tag{C.16}$$

for $\theta := 2 + 2 L_c^2 \kappa_0^2 \lambda_n^2 > 0$.

For the update rule C.14, letting $V_{3,t} := V_{e,t}(\bar{\boldsymbol{x}}_t - \bar{\boldsymbol{\sigma}}_t, t)$, we obtain

$$
\begin{aligned}
V_{3,t+1} - V_{3,t} &= \sum_{j=1}^{N} \| \phi_{t+1}^{t+j}(\boldsymbol{x}_{t+1} - \bar{\boldsymbol{\sigma}}_{t+1}) \|^2 - \sum_{j=0}^{N-1} \| \phi_t^{t+j}(\boldsymbol{x}_t - \bar{\boldsymbol{\sigma}}_t) \|^2 \\
&\leq -c_3 \| \boldsymbol{x}_t - \bar{\boldsymbol{\sigma}}_t \|^2 + c_4 \sqrt{n} \kappa \| \boldsymbol{x}_t - \bar{\boldsymbol{\sigma}}_t \| \| \beta \mathcal{L} \bar{\boldsymbol{\sigma}}_t \| \\
&\quad + \eta \frac{1}{t+1} \overline{\mathcal{H}}(\boldsymbol{x}_t) \| + \kappa_0 c_4 \sqrt{n} \| \boldsymbol{x}_t - \bar{\boldsymbol{\sigma}}_t \| \| g_t \delta \| \\
&\quad + 3 N L_\phi (\kappa^2 \| \beta \mathcal{L} \bar{\boldsymbol{\sigma}}_t \|^2 + \kappa^2 \| \eta \frac{1}{t+1} \overline{\mathcal{H}}(\boldsymbol{x}_t) \|^2 + \kappa_0^2 \| g_t \delta \|^2) \\
&\leq -[\frac{c_3}{2} - \kappa \left( - c_4 \sqrt{n} \beta / r - c_4^2 \sqrt{n} \eta / r \right. \\
&\quad - 2 c_4 \sqrt{n} \beta \lambda_n^2 r )] \| \boldsymbol{x}_t - \bar{\boldsymbol{\sigma}}_t \|^2 + \kappa \big( (2 c_4 \sqrt{n} \lambda_n^2 \beta r \\
&\quad + \frac{1}{t+1} \sqrt{n} \eta r L_\mathcal{H}^2) \| \boldsymbol{x}_t^\perp \|^2 + \frac{1}{t+1} \sqrt{n} \eta r L_\mathcal{H}^2 \| \boldsymbol{x}_t^\| \|^2 \big) \\
&\quad + \kappa^2 N L_\phi \big( 6 \beta^2 \lambda_n^2 (\| \boldsymbol{x}_t - \bar{\boldsymbol{\sigma}}_t \|^2 + \| \boldsymbol{x}_t^\perp \|^2) \\
&\quad + \frac{3}{t+1} \eta^2 L_\mathcal{H}^2 (\| \boldsymbol{x}_t^\perp \|^2 + \| \boldsymbol{x}_t^\| \|^2) \big) + C_1 g_t^2 \| \delta_t \|^2,
\end{aligned}
\tag{C.17}
$$

for $c_4 := N L_\varphi \theta$ and $C_1 := \kappa_0^2 \left( \frac{2 c_4^2}{c_3} + 3 N L_\varphi \right)$, where the first inequality is obtained by (C.13)-(C.14) and the fact $\| \boldsymbol{\delta}_t \|_\infty \leq \delta$ and the last inequality is obtained by (C.8) and Young's Inequality, with $r > 0$ being an undetermined parameter to be chosen later.

Now we introduce some parameters $\xi_1, \xi_2, \cdots > 0$ independent of $\beta, \eta$ and $r$ and some parameters $\zeta_1, \zeta_2, \cdots > 0$ independent of $\kappa$ as follows

$$\xi_1 = \frac{\lambda_2}{2}, \xi_2 = \frac{1 + L_{\mathcal{H}}^2}{2} + \frac{n^2 L_{\mathcal{H}}^4}{\mu^2},$$

$$\xi_3 = 2c_4\sqrt{n}\lambda_n^2 + \sqrt{n}L_{\mathcal{H}}^2 + \frac{4n^{\frac{5}{2}}L_{\mathcal{H}}^4}{\mu^2}, \xi_4 = c_4\sqrt{n},$$

$$\xi_4' = c_4^2\sqrt{n},$$

$$\xi_5 = \frac{\lambda_n}{2} + 2c_4\sqrt{n}\lambda_n^2,$$

$$\zeta_1 = 2\beta^2\lambda_n^2 + \eta^2(4+p)L_{\mathcal{H}}^2 + 6\beta^2\lambda_n^2 NL_\phi,$$

$$\zeta_2 = \eta^2(4+p)L_{\mathcal{H}}^2, \zeta_3 = 2\beta^2\lambda_n^2 + 6\beta^2\lambda_n^2 NL_\phi\theta.$$

where

$$p = \frac{4L_{\mathcal{H}}^2 n}{\mu} + \frac{8n^{\frac{3}{2}}rL_{\mathcal{H}}^2}{\mu} > 0.$$

Then we define the total Lyapunov functions of system (C.7) as

$$V_t := V_{1,t} + pV_{2,t} + V_{3,t}.$$

By (C.13), it is bounded as

$$V_t \leq \tfrac{1}{2}\|\bar{\boldsymbol{x}}_t^\perp\|^2 + \tfrac{p}{2}\|\bar{\boldsymbol{x}}_t^\|\|^2 + c_2\|\bar{\boldsymbol{x}}_t - \bar{\boldsymbol{\sigma}}_t\|^2. \tag{C.18}$$

We let $r \leq 1, \eta \leq \beta$ to simplify the following process, then by (C.10), (C.12) and (C.17), we have

$$V_{t+1} - V_t \leq -\kappa(\xi_1\beta - \frac{\xi_2\eta}{t+1} - \xi_3\beta r)\|\bar{\boldsymbol{x}}_t^\perp\|^2$$

$$-\frac{1}{t+1}\kappa\left(p\eta\frac{\mu}{4n}\right)\|\bar{\boldsymbol{x}}_t^\|\|^2 - \kappa\left(\frac{c_3}{2} - \xi_4\beta/r - \xi_4'\eta/r(t+1) - \xi_5\beta\right)\|\bar{\boldsymbol{x}}_t - \bar{\boldsymbol{\sigma}}_t\|^2$$

$$+ \kappa^2\left(\zeta_1\|\bar{\boldsymbol{x}}_t^\perp\|^2 + \frac{1}{(t+1)^2}\zeta_2\|\bar{\boldsymbol{x}}_t^\|\|^2 + \zeta_3\|\bar{\boldsymbol{x}}_t - \bar{\boldsymbol{\sigma}}_t\|^2\right)$$

$$+ C_1 g_t^2\|\delta_t\|^2. \tag{C.19}$$

Letting $r = \min\{\frac{\xi_1}{3\xi_3}, 1\}, \beta \leq \frac{c_3}{6(\xi_4/r+\xi_5)}, \eta \geq \frac{8n}{\mu\kappa}$ and $\kappa \leq \frac{1}{2}\min\{\frac{\xi_1\beta}{3\zeta_1}, \sqrt{\frac{2p}{\zeta_2}}, \frac{c_3}{6\zeta_3}\}$, with C.18, we can conclude that for $t \geq t_0 = \max\{\frac{3\xi_2\eta}{\xi_1\beta}, \frac{6\xi_4'\eta}{rc_3}, \frac{6p}{\kappa\xi_1\beta}, \frac{12pc_2}{\kappa c_3}\} - 1$, there holds

$$\Delta V_t \leq -\frac{2V_t}{t+1} + C_1 g_t^2\|\delta_t\|^2,$$

which yields

$$V_t \leq -\frac{2V_{t_0}}{(t+1)^2} + \sum_{\tau=t_0}^{t}\frac{2C_1}{(t+1-\tau)^2(\tau+1)^2}g_0^2\|\delta_t\|^2$$

$$\leq \frac{2V_{t_0} + 4C_1 g_0^2\|\delta_t\|^2}{(t+1)^2} + o\left(\frac{1}{t^2}\right). \tag{C.20}$$

Now, assuming that

$$\left\|\frac{\bar{\boldsymbol{x}}_t - \bar{\boldsymbol{\sigma}}_t}{g_t}\right\|_\infty \leq \frac{K}{\sqrt{d\bar{\psi}}} \tag{C.21}$$

holds for $t \geq t_0$, where $\overline{\psi}$ is the uniform upper bound of $\|\psi(t)\|$. By the definition of $\widetilde{C}_{\text{HarMo}}$, we know that there holds

$$\|\boldsymbol{\delta}_t\| \leq \frac{l\sqrt{d}}{2}\overline{\psi}.$$

Substituting it into equation C.20, by the definition of $V_t$, we have

$$\left\|\frac{\bar{\boldsymbol{x}}_t - \bar{\boldsymbol{\sigma}}_t}{g_t}\right\|_\infty^2 \leq \frac{V_t}{g_t^2} \leq \frac{2V_{t_0} + 4C_1 g_0^2 \|\delta_t\|^2}{g_0^2} \leq \frac{2V_{t_0} + C_1 g_0^2 l^2 d\overline{\boldsymbol{\psi}}^2}{g_0^2}.$$

Then, letting $g_0^2 = \frac{2V_{t_0}}{C_1 d\overline{\boldsymbol{\psi}}^2 l^2}$, then it can be directly obtained that equation C.21 holds if

$$\frac{K}{l} \geq 2C_1 \overline{\boldsymbol{\psi}}^2 d,$$
$$\Rightarrow m \geq \log_2(2C_1 \overline{\boldsymbol{\psi}}^2 d). \tag{C.22}$$

In one word, when the condition above is satisfied, we can obtain $V(t) = \mathcal{O}\left(\frac{1}{t^2}\right)$ by equation C.20. With the definition of $V_t$, Theorem 4.1 is proved.

$\square$

## D EXPERIMENT DETAILS

### D.1 COMMUNICATION TOPOLOGY

We consider four representative communication topologies in our experiments: the ring, torus, fully-connected networkand the Erdős–Rényi (ER) random graph, as illustrated in Figure 5. To further evaluate the robustness of our algorithm under different network structures, we report in Table 3 the average node degree of each topology.

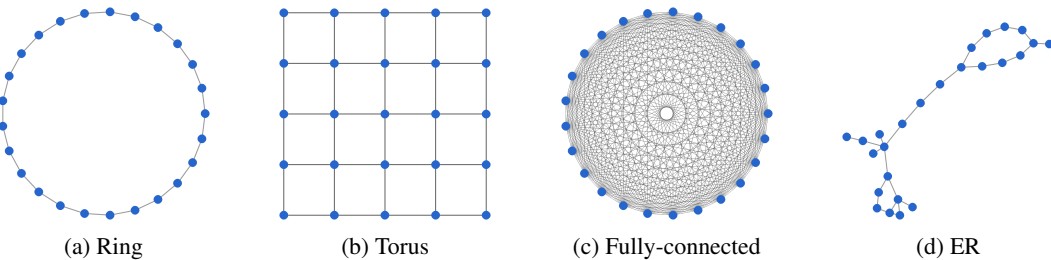

|        (a) Ring        |        (b) Torus        |        (c) Fully-connected        |        (d) ER        |

Figure 5: Illustration of different communication topologies with $n = 25$ clients: (a) Ring topology, where each node connects to two immediate neighbors; (b) Torus topology, represented as a $5 \times 5$ periodic grid; (c) Fully-connected topology, where each node connects to all others; (d) ER graph, modeling a complex network with probabilistic connectivity $p_{er}$.

Table 3: Average node degree of different topologies.

| Topology | Ring | Torus | Fully-connected | ER |
|---|---|---|---|---|
| **Average node degree** | 2 | 4 | $n-1$ | $p_{er} * (n-1)$ |

### D.2 COMPRESSORS AND QUANTIZERS IN COMPARATIVE EXPERIMENTS

**Top-$\alpha$ Compressor.** Following Alistarh et al. (2018); Stich et al. (2018), the biased Top-$\alpha$ operator $\mathrm{top}_\alpha : \mathbb{R}^d \to \mathbb{R}^d$ is defined as

$$\mathrm{Top}_\alpha(\boldsymbol{x}) := \boldsymbol{x} \odot \boldsymbol{u}(\boldsymbol{x}), \tag{11}$$

where $\boldsymbol{u}(\boldsymbol{x}) \in \{0,1\}^d$ is a binary masking vector. The mask selects the $\lceil \alpha d \rceil$ entries of $\boldsymbol{x}$ with the largest absolute values, i.e., $\|\boldsymbol{u}(\boldsymbol{x})\|_1 = \lceil \alpha d \rceil$ and $(\boldsymbol{u})_i = 1$ if index $i$ corresponds to one of these largest coordinates. Formally, let $\pi$ be a permutation such that

$$|\boldsymbol{x}_{\pi(1)}| \geq |\boldsymbol{x}_{\pi(2)}| \geq \cdots \geq |\boldsymbol{x}_{\pi(d)}|,$$

then $(\boldsymbol{u})_i = 1$ if $i \in \{\pi(1), \ldots, \pi(\lceil \alpha d \rceil)\}$ and $(\boldsymbol{u})_i = 0$ otherwise.

The Top-$\alpha$ operator therefore keeps only the top $\alpha$ fraction of coordinates and zeros out the rest, which corresponds to a compression level of $\delta = \alpha$ (Stich et al., 2018). To transmit the compressed vector, we need to send both the selected values and their indices, leading to a communication cost of $2 \cdot 32 \lceil \alpha d \rceil$ bits (assuming 32-bit floating-point representation). Thus, Top-$\alpha$ reduces the communication cost from $32d$ bits to $64\lceil \alpha d \rceil$ bits per iteration.

**Sign Quantizer** (Bernstein et al., 2018; Karimireddy et al., 2019). The biased (scaled) sign operator Sign : $\mathbb{R}^d \to \mathbb{R}^d$ is defined as

$$\text{Sign}(\boldsymbol{x}) := \frac{\|\boldsymbol{x}\|_1}{d} \cdot \text{sgn}(\boldsymbol{x}), \tag{12}$$

where $\text{sgn}(\boldsymbol{x})$ denotes the element-wise sign function. This operator replaces each entry of $\boldsymbol{x}$ with only its sign (i.e., $+1$ or $-1$), and rescales the whole vector by the average magnitude $\frac{\|\boldsymbol{x}\|_1}{d}$. The sign operator is a $\delta = \frac{\|\boldsymbol{x}\|_1^2}{d\|\boldsymbol{x}\|_2^2}$ compression operator (Karimireddy et al., 2019). Regarding communication cost, we only need to transmit $d + 32$ bits in total: $d$ bits to indicate the sign of each coordinate and 32 bits to transmit the scaling factor $\|\boldsymbol{x}\|_1$. In comparison, transmitting the full-precision vector requires $32d$ bits. Thus, the sign compressor reduces communication from $32d$ bits to $(d + 32)$ bits per iteration.

**Sketch** (Alon et al., 1996). Sketching compresses a high-dimensional vector by projecting it into a much lower-dimensional space using a sketching matrix $R \in \mathbb{R}^{b \times d}$ with $b \ll d$. For any vector $\boldsymbol{x} \in \mathbb{R}^d$, the sketch operator sk : $\mathbb{R}^d \to \mathbb{R}^b$ is defined as

$$\text{sk}(\boldsymbol{x}) := R\boldsymbol{x} \in \mathbb{R}^b, \tag{13}$$

which corresponds to multiplying $\boldsymbol{x}$ by the sketching matrix $R$.

To recover an unbiased estimate of $\boldsymbol{x}$ from the compressed vector $\text{sk}(\boldsymbol{x})$, a desketching operator desk : $\mathbb{R}^b \to \mathbb{R}^d$ is applied using the transpose of the same sketching matrix:

$$\text{desk}(\boldsymbol{s}) := R^\top \boldsymbol{s} \in \mathbb{R}^d, \tag{14}$$

where $\boldsymbol{s} = \text{sk}(\boldsymbol{x})$.

Regarding communication efficiency, the client transmits the $b$-dimensional sketched vector $\text{sk}(\boldsymbol{x})$, requiring $32b$ bits per iteration (each entry stored in 32-bit floating-point format), instead of the original $32d$ bits. Since $b \ll d$, the sketch operator significantly reduces communication while still preserving an unbiased estimate of the update direction.

**Low-Rank** (Wang et al., 2018). Low-Rank method approximates a high-dimensional matrix-shaped update by factorizing it into two much smaller matrices of rank $r$. Let $\boldsymbol{G} \in \mathbb{R}^{p \times n}$ denote a gradient matrix reshaped from a $d$-dimensional updateand assume $r \ll \min\{p, n\}$. The Low-Rank method defines a mapping $\mathbb{R}^{p \times n} \to \mathbb{R}^{p \times r} \times \mathbb{R}^{r \times n}$, which projects $\boldsymbol{G}$ into two low-dimensional factors.

To construct a rank-$r$ approximation, a random matrix $\boldsymbol{Q} \in \mathbb{R}^{n \times r}$ is sampledand the first projection is computed as

$$\boldsymbol{P} := \boldsymbol{G}\boldsymbol{Q} \in \mathbb{R}^{p \times r}. \tag{15}$$

After orthonormalizing $\boldsymbol{Q}$, a second projection is formed:

$$\boldsymbol{Q}^\top := \boldsymbol{P}^\top \boldsymbol{G} \in \mathbb{R}^{r \times n}. \tag{16}$$

The reconstructed update is then given by

$$\widetilde{\boldsymbol{G}} := \boldsymbol{P}\boldsymbol{Q}^\top, \tag{17}$$

which serves as a rank-$r$ approximation of $\boldsymbol{G}$.

Regarding communication efficiency, instead of transmitting the full $p \times n$ matrix (which requires $32pn$ bits), the sender only transmits the two Low-Rank factors $\boldsymbol{P} \in \mathbb{R}^{p \times r}$ and $\boldsymbol{Q}^\top \in \mathbb{R}^{r \times n}$, requiring a total of $32r(p + n)$ bits. Thus, when $r \ll \min\{p, n\}$, the communication cost is drastically reduced.

We summarize here how the communication cost, computational complexityand storage complexity of Top-$\alpha$, Sign, Sketch, Low-Rankand HarMo are obtained. For Top-$\alpha$, each iteration transmits the

largest $\alpha d$ coordinates of a $d$-dimensional vector together with their indices, leading to a communication cost of $64\lceil \alpha d\rceil$ bits. Identifying these entries requires partial sorting, giving a computational complexity of $\mathcal{O}(d\log d)$. The storage complexity is dominated by the local model and the residual maintained by the compressor, resulting in $\mathcal{O}(d)$ storage. For the Sign operator, communication consists of $d$ one-bit signs and a 32-bit scaling factor. Since only coordinate-wise sign extraction and a norm computation are required, the computational complexity is $\mathcal{O}(d)$and storing the local model and residual again yields $\mathcal{O}(d)$ storage.

Table 4: Per-iteration communication cost (bit), computational complexity and storage complexity (per node) for different methods.

| Method | Communication Cost | Computational Complexity | Storage Complexity |
|---|---|---|---|
| Top-$\alpha$ | $64\lceil \alpha d\rceil$ | $\mathcal{O}(d\log d)$ | $\mathcal{O}(d)$ |
| Sign | $d+32$ | $\mathcal{O}(d)$ | $\mathcal{O}(d)$ |
| Sketch | $32b$ | $\mathcal{O}(bd)$ | $\mathcal{O}(bd)$ |
| Low-Rank | $32r(p+n)$ | $\mathcal{O}(rd)$ | $\mathcal{O}(r(p+n))$ |
| HarMo | $m$ | $\mathcal{O}(d)$ | $\mathcal{O}(d)$ |

Sketching compresses a $d$-dimensional vector into a $b$-dimensional sketch through a linear mapping implemented by a sketching matrix or equivalent hashing structure, resulting in a communication cost of $32b$ bits. Computing the sketch requires a matrix–vector multiplication of size $b\times d$, producing a computational complexity of $\mathcal{O}(bd)$. Storing the sketching matrix (or its hash parameters) requires $\mathcal{O}(bd)$ memory, which determines the overall storage complexity.

Low-Rank compression reshapes the $d$-dimensional vector into a matrix $\boldsymbol{G}\in\mathbb{R}^{p\times n}$ (with $pn=d$) and transmits two factor matrices of sizes $p\times r$ and $r\times n$, yielding a communication cost of $32r(p+n)$ bits per iteration. The dominant computation comes from forming the products $\boldsymbol{GQ}$ and $\boldsymbol{P}^\top\boldsymbol{G}$, each costing $\mathcal{O}(rd)$, so the overall computational complexity is $\mathcal{O}(rd)$. Storing both Low-Rank factors requires $\mathcal{O}(r(p+n))$ memory in addition to the model parameters, giving a total storage complexity of $\mathcal{O}(r(p+n))$.

In contrast, HarMo communicates only a single $m$-bit scalar obtained by projecting the vector onto a harmonic direction and quantizing the result, yielding an $m$-bit communication cost per iteration. Both the projection and reconstruction require linear time in $d$, resulting in $\mathcal{O}(d)$ computational complexity. Since the harmonic direction is generated on the fly and no residual is maintained, HarMo stores only the local model parameters, achieving the minimal storage complexity of $\mathcal{O}(d)$ while providing substantially lower communication cost than all existing compression methods.

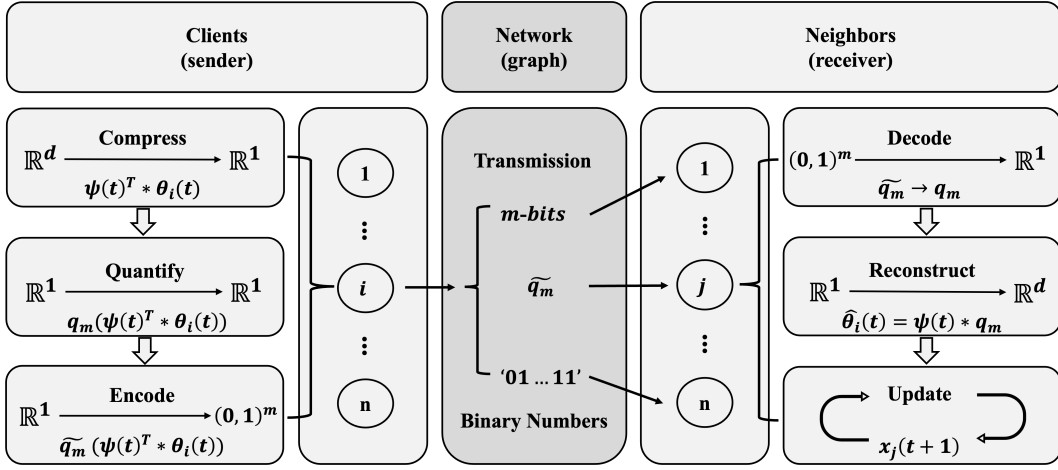

Figure 6: Illustration of LBGD-HarMo algorithm, where high-dimensional updates are compressed, quantizedand transmitted as binary codes over the communication graph, then decoded, reconstructedand used for variable updates by neighboring clients.

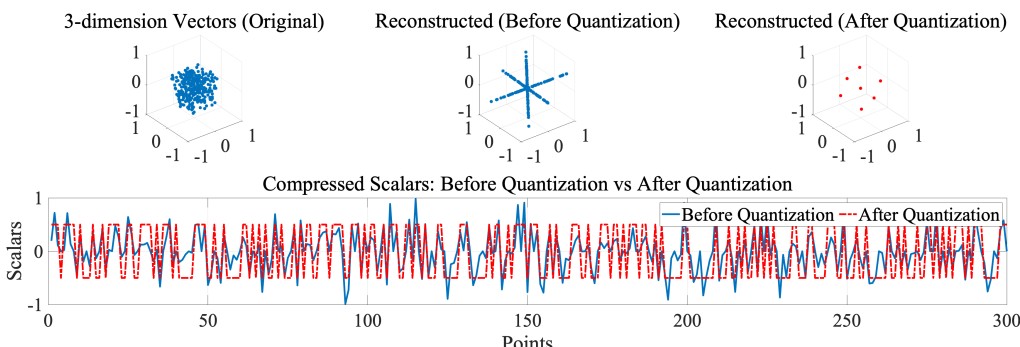

Figure 7: Illustration of HarMo applied to 300 3-dimension vectors (before and after quantization).

### D.3 HYPERPARAMETERS FOR SYNTHETIC QUADRATIC OPTIMIZATION PROBLEM

The parameters of the synthetic quadratic optimization problem are set as follows:

For the DSGD algorithm, we adopt a diminishing learning rate $\gamma_t = \frac{0.0036}{\sqrt{t}}$. For the CHOCO algorithm, we employ the Top-$\alpha$ compressor with $\alpha = 0.125$, a diminishing learning rate $\eta_t = \frac{0.2}{t+1}$ and a fixed consensus step size $\gamma = 0.08$. For the MoTEF algorithm, we adopt the Top-$\alpha$ compressor with $\alpha = 0.125$, a fixed learning rate $\gamma = 0.2$, a consensus step size $\eta = 0.0005$ and a momentum coefficient $\lambda = 0.005$. For the LBGD-Sign algorithm, we adopt the Sign quantizer (1 bit), the scaling factor $g_0 = 10$ with decay $\gamma = 0.9999$, the gradient step size $\kappa = 0.05$, the consensus step size $\kappa_0 = 0.005$, and the parameter $\eta = 5$. For the LBGD-HarMo algorithm, we set the quantizer parameters as $m \in \{3, 4, 8, 16\}$, the scaling factor $g_0 = 10$ with decay $\gamma = 0.9999$, the gradient step size $\kappa = 0.05$, the consensus step size $\kappa_0 = 0.005$ and the parameter $\eta \in \{0.022, 0.032, 0.035, 0.04, 0.05, 0.065\}$.

### D.4 HYPERPARAMETERS FOR LOGISTIC REGRESSION WITH STRONGLY CONVEX REGULARIZER

The parameters of the logistic regression with strongly convex regularizer are set as follows:

For the DSGD algorithm, we adopt a learning rate $\gamma = 0.1$. For the CHOCO algorithm, we employ the Top-$\alpha$ compressor with $\alpha = 0.1$, a learning rate $\eta = 0.1$ and a fixed consensus step size $\gamma = 0.01$. For the MoTEF algorithm, we adopt the Top-$\alpha$ compressor with $\alpha = 0.1$, a fixed learning rate $\gamma = 0.5$, a consensus step size $\eta = 0.005$ and a momentum coefficient $\lambda = 0.05$. For the LBGD-Sign algorithm, we adopt the Sign quantizer (1 bit), the scaling factor $g_0 = 5$ with decay $\gamma = 0.99999$, the gradient step size $\kappa = 0.1$, the consensus step size $\kappa_0 = 0.01$, and the parameter $\eta = 5$. For the LBGD-HarMo algorithm, we set the quantizer parameters as $m \in \{12, 16\}$, the scaling factor $g_0 = 5$ with decay $\gamma = 0.99999$, the gradient step size $\kappa = 0.1$, the consensus step size $\kappa_0 = 0.01$, and the parameter $\eta \in \{0.022, 0.065, 0.085, 0.1\}$.

### D.5 HYPERPARAMETERS FOR NEURAL NETWORK TRAINING

The parameters of the neural network training are set as follows:

The experimental parameters for neural network training are configured as follows. For FedAvg and DSGD, we use a learning rate of $\gamma = 0.1$. For CHOCO, we employ the Top-$\alpha$ compressor with $\alpha = 0.1$, a learning rate of $\eta = 1.60$, and a fixed consensus step size of $\gamma = 0.15$. For LBGD-Sign, we adopt the Sign quantizer (1 bit) with a scaling factor $g_0 = 5$ and decay $\gamma = 0.99999$, a gradient step size $\kappa = 0.25$, a consensus step size $\kappa_0 = 0.15$ and parameter $\eta = 5$. For LBGD-HarMo, we set the quantizer precision $m = 24$, the scaling factor $g_0 = 5$ with decay $\gamma = 0.99999$, the gradient step size $\kappa \in \{0.1, 0.25, 0.5\}$, the consensus step size $\kappa_0 \in \{0.01, 0.025, 0.05\}$ and $\eta = 5$.

# E  ADDTIONAL EXPERIMENTS

## E.1  SYNTHETIC QUADRATIC OPTIMIZATION PROBLEM

### E.1.1  PARAMETER SENSITIVITY ANALYSIS

We analyzed the sensitivity of the following parameters in the LBGD-HarMo algorithm: $\beta$, $\eta$, $g_0$, $\gamma$, $\kappa$ and $\kappa_0$. The experimental results, shown in Figure 8, indicate that higher values of $\beta$ enhance convergence by improving the alignment between local updates and the consensus model, while simultaneously reducing the communication burden. For $\eta$, increasing its value accelerates convergence, but it may lead to instability in later stages. In contrast, smaller values of $\eta$ ensure smoother convergence, albeit at the cost of slower progress. The initial step size $g_0$ directly influences the convergence rate, with larger values facilitating faster convergence but risking overshooting, while smaller values ensure more gradual updates. The decay factor $\gamma$ dictates how rapidly the step size diminishes during iterations. Larger values of $\gamma$ expedite convergence initially but may cause the step size to decrease too quickly, ultimately slowing progress, whereas smaller values lead to smoother updates without causing abrupt changes. The consensus step size $\kappa_0$ governs the speed of synchronization with neighboring nodes. Larger values of $\kappa_0$ accelerate synchronization, but overly large values can introduce instability. Finally, the local step size $\kappa$, determines the magnitude of local updates. Larger values of $\kappa$ result in faster convergence, but they may lead to overshooting, whereas smaller values provide more controlled updates, improving stability.

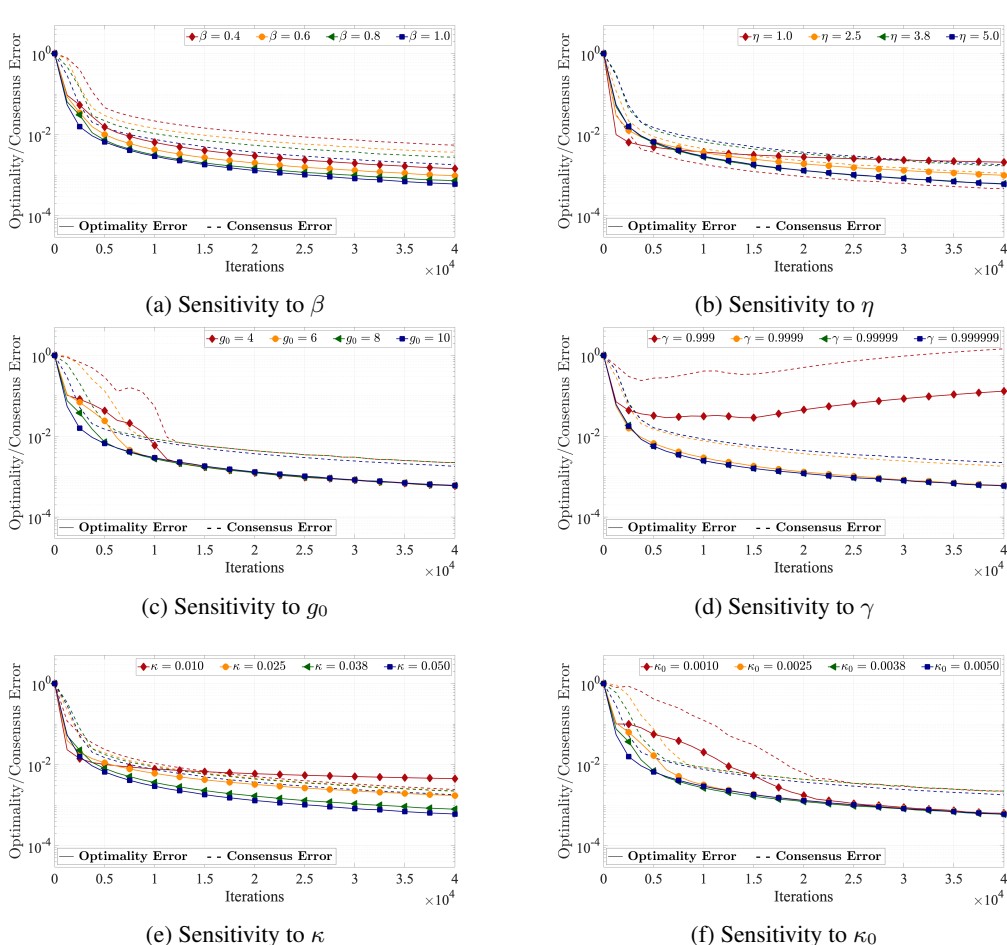

(a) Sensitivity to $\beta$

(b) Sensitivity to $\eta$

(c) Sensitivity to $g_0$

(d) Sensitivity to $\gamma$

(e) Sensitivity to $\kappa$

(f) Sensitivity to $\kappa_0$

Figure 8: Parameter sensitivity analysis showing the effect of different parameter values on the convergence performance of algorithm 1.

### E.1.2 DIFFERENT VARIABLE DIMENSIONS

We further examine the effect of dimensionality by evaluating LBGD-HarMo under $d \in \{4, 8, 16, 100\}$. As shown in Fig. 9a, increasing the dimension leads to a moderate rise in both optimality and consensus errors, which is consistent with the fact that higher-dimensional models involve more parameters, so the aggregated updates tend to exhibit larger variation before reaching consensus. Nonetheless, the convergence trend remains stable across all configurations, indicating that LBGD-HarMo remains effective in higher-dimensional problems.

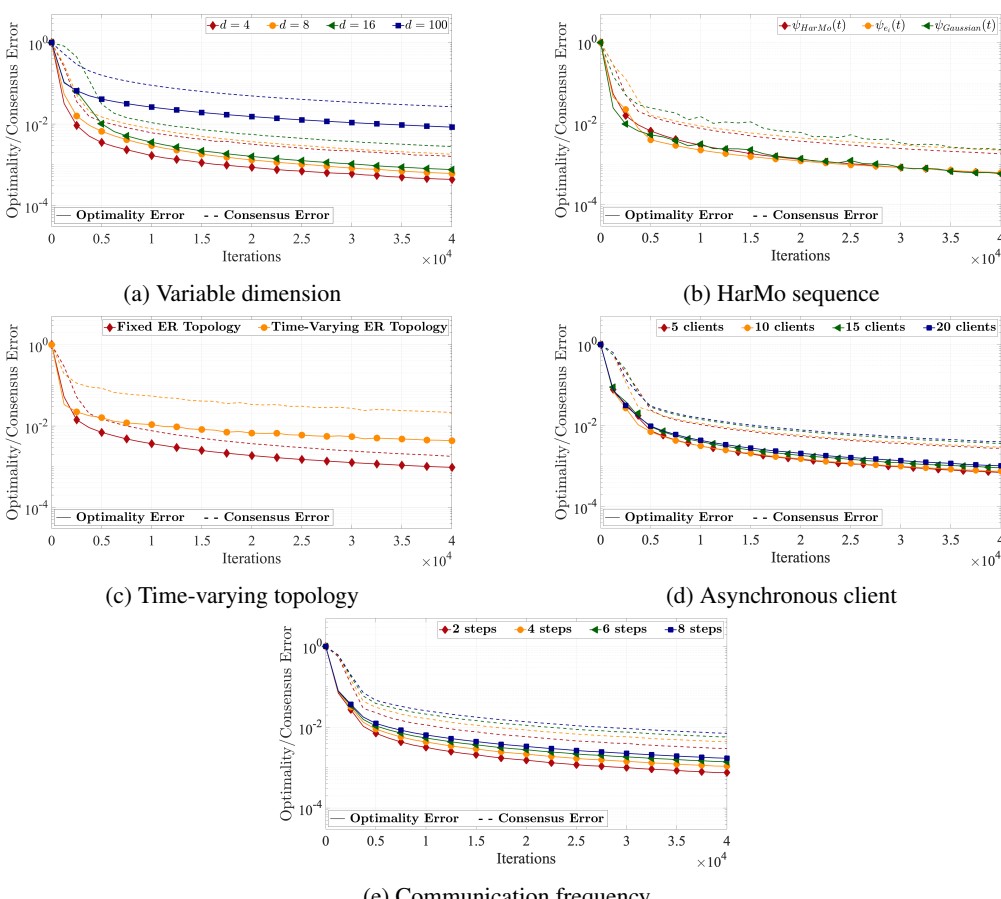

(a) Variable dimension

(b) HarMo sequence

(c) Time-varying topology

(d) Asynchronous client

(e) Communication frequency

Figure 9: Convergence performance of synthetic quadratic optimization problem under different settings: (a) varying the variable dimension; (b) varying the HarMo sequence; (c) using the time-varying topology; (d) varying the number of asynchronous client; (e) varying the communication frequency, set the number of clients to 25 and quantization precisions to 8 bits.

### E.1.3 DIFFERENT HARMO SEQUENCES

We evaluate how different choices of projection vectors influence the performance of our algorithm. In addition to the harmonic sequence $\psi_{\text{HarMo}}(t)$, we also test the cyclic coordinate vectors $\psi_{\boldsymbol{e}_i}(t) = \boldsymbol{e}_i$ (where $i = 1 + (t \bmod d)$) and pseudo-random Gaussian vectors $\psi_{\text{Gaussian}}(t)$ drawn from an isotropic distribution with a fixed seed. As shown in Figure 9b, all three choices yield very similar convergence trajectories, indicating that the algorithm is largely insensitive to the specific form of the projection direction as long as it provides sufficient directional variation over time. The Gaussian vectors perform slightly more irregularly due to their stochastic nature, while the cyclic vectors produce smooth but highly axis-aligned updates. The harmonic vector $\psi_{\text{HarMo}}(t)$ achieves a desirable middle ground: it introduces rich directional variability through its frequency structure while remaining fully deterministic and reproducible. This makes $\psi_{\text{HarMo}}(t)$ a particularly suitable

choice for large-scale decentralized optimization, combining stable empirical performance with theoretical tractability and ease of implementation.

### E.1.4 TIME-VARYING TOPOLOGY

In the time-varying topology experiments, we used an Erdos-Renyi (ER) graph with a connection probability of $p_{er} = 0.1$ for each time step. For comparison, we also tested a fixed ER graph. As shown in Figure 9c, while the time-varying topology still converges, its performance is slightly degraded compared to the fixed topology. This performance drop can be attributed to the dynamic nature of the time-varying topology, which causes occasional disruptions in the network structure. These changes lead to temporary loss of connectivity or inconsistent communication between clients, which may slow down the consensus process and lead to a longer convergence time.

### E.1.5 ASYNCHRONOUS UPDATE

In the asynchronous update experiments, we tested the effect of partial clients exchanging information with the rest of the clients in a non-fixed manner. We varied the number of these asynchronous clients and the communication frequency between them. As shown in Figure 9d and 9e, the algorithm is still able to converge, but as the number of asynchronous clients increases and the communication frequency decreases, performance is slightly affected. This degradation in performance can be attributed to the reduced synchronization between clients, which leads to delayed updates and a slower consensus process. However, the algorithm still maintains convergence, demonstrating robustness to changes in the update scheme.

### E.1.6 DIFFERENT COMPRESSION RATIOS FOR CHOCO AND MoTEF ALGORITHMS

In this experiment, we compare the performance of the CHOCO (Koloskova et al., 2020a) and MoTEF (Islamov et al., 2025) algorithms under different compression ratios, specifically varying the parameter $\alpha$. We selected $\alpha = 0.125$ (corresponding to retaining only one element) for comparison with our proposed algorithm. This choice of $\alpha$ strikes a balance between communication efficiency and algorithm performance.

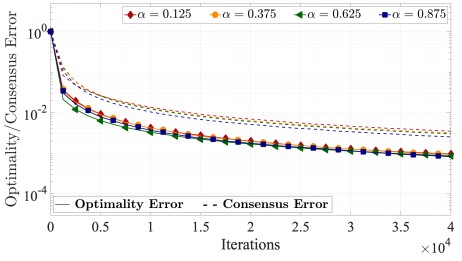
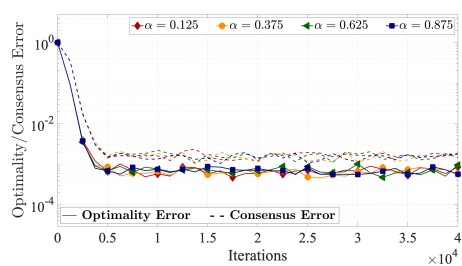

| (a) CHOCO with different compression-ratios | (b) MoTEF with different compression-ratios |

Figure 10: Convergence performance of synthetic quadratic optimization problem under (a) varying the compression ratio $\alpha$ for CHOCO and (b) varying the compression ratio $\alpha$ for MoTEF.

### E.1.7 CONVERGENCE ITERATIONS

We recorded the total number of iterations and communication cost for different experimental settings and algorithms to reach an Optimality Error of $10^{-3}$. For quantization precisions of 3 and 4 bits, the optimality errors were observed to be 0.1 and 0.06, respectively, due to lower quantization precision. Additionally, for a variable dimension of 100, the optimality error reaches $10^{-2}$. The table compares various experimental settings, with default values for the number of clients (25), communication topology (ring), quantization precision (8 bits) and communication ratio $\alpha = 0.125$ for CHOCO and MoTEF. Parameters were selected as outlined in Appendix D, with synchronization used for the updates. The settings are adjusted according to specific experimental conditions. The table 5 and table 6 provide the corresponding iteration counts and communication costs.

Table 5: Total number of iterations and communication cost under **different experimental settings** on the synthetic quadratic optimization problem.

| Number of clients | Iteration | Communication cost (MB) |
|---|---|---|
| 9 | 39,472 | 1.36 |
| 25 | 25,810 | 2.46 |
| 64 | 28,020 | 6.84 |
| 100 | 36,713 | 14.00 |

| Topology | Iteration | Communication cost (MB) |
|---|---|---|
| Fully-connected | 9,550 | 10.93 |
| Torus | 28,523 | 5.44 |
| Fixed ER | 38,233 | 4.38 |
| Ring | 25,810 | 2.46 |
| Time-varying ER | 79,554 | 9.10 |

| Dimension | Iteration | Communication cost (MB) |
|---|---|---|
| 4 | 17,056 | 1.67 |
| 8 | 25,810 | 2.46 |
| 16 | 31,354 | 2.99 |
| 100 | 33,105 | 3.16 |

| Sequence | Iteration | Communication cost (MB) |
|---|---|---|
| $\psi_{\text{HarMo}}$ | 25,810 | 2.46 |
| $\psi_{\boldsymbol{e}_i}(t)$ | 24,224 | 2.31 |
| $\psi_{\text{Gaussian}}(t)$ | 25,711 | 2.45 |

| Number of asynchronous clients | Iteration | Communication cost (MB) |
|---|---|---|
| 5 | 28,879 | 2.48 |
| 10 | 29,985 | 2.29 |
| 15 | 37,063 | 2.47 |
| 20 | 41,027 | 2.35 |

| Comm. frequency | Iteration | Communication cost (MB) |
|---|---|---|
| 2 | 29,985 | 2.29 |
| 4 | 42,545 | 2.84 |
| 6 | 56,287 | 3.58 |
| 8 | 69,185 | 4.29 |

| Quantization precision (bit) | Iteration | Communication cost (MB) |
|---|---|---|
| 3 | 47,653 | 4.54 |
| 4 | 38,717 | 3.69 |
| 8 | 25,810 | 2.46 |
| 16 | 24,789 | 4.73 |

| Compression ratio $\alpha$ (CHOCO) | Iteration | Communication cost (MB) |
|---|---|---|
| 0.125 | 38,074 | 29.05 |
| 0.375 | 37,118 | 84.96 |
| 0.625 | 33,939 | 129.47 |
| 0.875 | 33,546 | 179.15 |

| Compression ratio $\alpha$ (MoTEF) | Iteration | Communication cost (MB) |
|---|---|---|
| 0.125 | 4,180 | 3.19 |
| 0.375 | 3,812 | 8.72 |
| 0.625 | 3,775 | 14.40 |
| 0.875 | 3,687 | 19.69 |

Table 6: Total number of iterations and communication cost under **different algorithms** on the synthetic quadratic optimization problem.

| Algorithm | Method | Iteration | Communication cost (MB) |
|---|---|---|---|
| DSGD | None | 9,100 | 27.77 |
| CHOCO | TOP-$\alpha$ | 38,074 | 14.52 |
| MoTEF | TOP-$\alpha$ | 4,180 | 3.19 |
| LBGD | Sign | 46,707 | 22.27 |
| **LBGD** | **HarMo** | 25,810 | 2.46 |

The table 7 presents the total number of iterations and communication cost under various hyperparameter values on the synthetic quadratic optimization problem to reach an optimality error of $10^{-3}$. The results are consistent with the parameter sensitivity analysis presented in Appendix E.1.1, showing that our algorithm exhibits robustness across a range of hyperparameter values.

Table 7: Total number of iterations and communication cost under **different hyperparameter values** on the synthetic quadratic optimization problem.

| Value of hyperparameter $\beta$ | Iteration | Communication cost (MB) |
|---|---|---|
| 0.4 | 56,568 | 5.39 |
| 0.6 | 38,639 | 3.68 |
| 0.8 | 29,732 | 2.84 |
| 1.0 | 25,810 | 2.46 |
| **Value of hyperparameter $\eta$** | **Iteration** | **Communication cost (MB)** |
| 1.0 | 56,686 | 5.41 |
| 2.5 | 40,103 | 3.82 |
| 3.8 | 26,543 | 2.53 |
| 5.0 | 25,810 | 2.46 |
| **Value of hyperparameter $g_0$** | **Iteration** | **Communication cost (MB)** |
| 4 | 26,230 | 2.50 |
| 6 | 26,121 | 2.49 |
| 8 | 25,880 | 2.47 |
| 10 | 25,810 | 2.46 |
| **Value of hyperparameter $\gamma$** | **Iteration** | **Communication cost (MB)** |
| 0.999 | divergence | / |
| 0.9999 | 25,810 | 2.46 |
| 0.99999 | 25,907 | 2.47 |
| 0.999999 | 25,908 | 2.47 |
| **Value of hyperparameter $\kappa$** | **Iteration** | **Communication cost (MB)** |
| 0.010 | 84,927 | 8.10 |
| 0.025 | 70,111 | 6.69 |
| 0.038 | 32,599 | 3.11 |
| 0.050 | 25,810 | 2.46 |
| **Value of hyperparameter $\kappa_0$** | **Iteration** | **Communication cost (MB)** |
| 0.0010 | 27,100 | 2.58 |
| 0.0025 | 26,583 | 2.54 |
| 0.0038 | 26,486 | 2.53 |
| 0.0050 | 25,810 | 2.46 |

## E.2 LOGISTIC REGRESSION WITH STRONGLY CONVEX REGULARIZER

We evaluate the performance of LBGD-HarMo on the logistic regression under varying system configurations, including different numbers of clients (4, 9, 16and 25) and communication topologies (Fully-connected, Torus, ERand Ring), using both IID and Non-IID data distributions. As shown in Table 8, LBGD-HarMo consistently maintains stable test accuracy across all settings. These results demonstrate the robustness and scalability of LBGD-HarMo in decentralized optimization with strongly convex objectives across diverse network structures and data heterogeneity conditions.

Table 8: Test accuracy (%) and Runtime (s) after the entire training process of logistic regression with strongly convex regularizer under different numbers of clients (4, 9, 16, 25), data distributions (IID vs Non-IID)and communication topologies (Fully-connected, Torus, Fixed / Time-varying ERand Ring). We use **LBGD-HarMo** with $m = 16$ bits for all cases.

| Number of clients | Data distribution | Test accuracy (%) | Runtime (s) |
|---|---|---|---|
| 4 | IID | 87.98 | 639.99 |
| | Non-IID | 87.64 | 640.04 |
| 9 | IID | 87.84 | 733.49 |
| | Non-IID | 87.63 | 734.12 |
| 16 | IID | 87.68 | 852.07 |
| | Non-IID | 87.49 | 852.65 |
| 25 | IID | 87.33 | 921.19 |
| | Non-IID | 87.17 | 922.05 |
| **Topology** | **Data distribution** | **Test accuracy (%)** | **Runtime (s)** |
| Fully-connected | IID | 88.10 | 740.16 |
| | Non-IID | 87.92 | 741.25 |
| Torus | IID | 87.99 | 736.95 |
| | Non-IID | 87.83 | 736.13 |
| Fixed ER | IID | 87.91 | 735.46 |
| | Non-IID | 87.72 | 735.92 |
| Ring | IID | 87.84 | 733.49 |
| | Non-IID | 87.63 | 734.12 |
| Time-varying ER | IID | 86.81 | 965.06 |
| | Non-IID | 86.69 | 966.48 |

## E.3 NEURAL NETWORK TRAINING PROBLEM

**Parameters of ResNet-18.** ResNet-18 is a widely used convolutional neural network architecture composed of 18 layers, including convolutional, normalizationand residual blocks. It contains approximately 11.2 million parameters, corresponding to a total size of about 44.7 MB.

Table 9: Parameter count and size of each layer in ResNet-18.

| Layer | Number of Parameters | Size (MB) |
|---|---|---|
| Conv1 (7×7, 64) | 9,408 | 0.036 |
| BatchNorm1 | 128 | 0.0005 |
| Layer1 (2×BasicBlock, 64) | 147,456 | 0.59 |
| Layer2 (2×BasicBlock, 128) | 524,288 | 2.10 |
| Layer3 (2×BasicBlock, 256) | 2,097,152 | 8.39 |
| Layer4 (2×BasicBlock, 512) | 8,388,608 | 33.55 |
| Fully Connected (512→10) | 5,130 | 0.020 |
| **Total** | **11,172,170** | **44.7 MB** |

**Data Distributions**. To simulate both IID and Non-IID data distributions across clients, we set $p = 100000$ for the IID case and $p = 0.3$ for the Non-IID caseand the resulting data s are illustrated in figure 11.

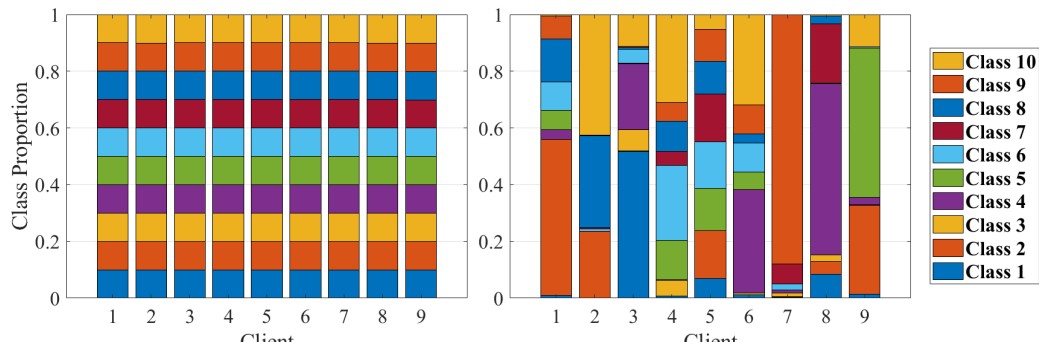

Figure 11: Visualization of data distributions across clients with different Dirichlet parameters: the left subfigure ($p = 100000$) corresponds to the IID caseand the right subfigure ($p = 0.3$) corresponds to the Non-IID case.

We evaluate the performance of LBGD-HarMo under various configurations, following the same experimental settings as in Appendix E.2 for the logistic regression task. As shown in Table 10, LBGD-HarMo maintains stable test accuracy across all configurations, demonstrating robustness to network variations. These results confirm the effectiveness of LBGD-HarMo in decentralized non-convex optimization.

Table 10: Test accuracy (%) and Runtime (min) after the entire training process of neural network training under different numbers of clients (4, 9, 16, 25), data distributions (IID vs Non-IID)and communication topologies (Fully-connected, Torus, Fixed / Time-varying ERand Ring). We use **LBGD-HarMo** with $m = 24$ bits for all cases.

| Number of clients | Data distribution | Test accuracy (%) | Runtime (min) |
|---|---|---|---|
| 4 | IID | 88.16 | 158 |
| | Non-IID | 86.61 | 158 |
| 9 | IID | 86.69 | 203 |
| | Non-IID | 85.17 | 203 |
| 16 | IID | 85.45 | 270 |
| | Non-IID | 84.10 | 270 |
| 25 | IID | 84.82 | 354 |
| | Non-IID | 83.04 | 354 |
| **Topology** | **Data distribution** | **Test accuracy (%)** | **Runtime (min)** |
| Fully-connected | IID | 86.96 | 203 |
| | Non-IID | 85.85 | 203 |
| Torus | IID | 86.88 | 203 |
| | Non-IID | 85.26 | 203 |
| Fixed ER | IID | 86.70 | 204 |
| | Non-IID | 85.23 | 204 |
| Ring | IID | 86.69 | 203 |
| | Non-IID | 85.17 | 203 |
| Time-varying ER | IID | 83.51 | 258 |
| | Non-IID | 82.36 | 258 |

## F    LARGE LANGUAGE MODELS USAGE STATEMENT

In compliance with the ICLR 2026 policy on the use of Large Language Models (LLMs), we hereby disclose their role in the preparation of this paper. We employed LLMs (ChatGPT, GPT-5 by OpenAI) for (1) polishing the English writing style to improve readability and conciseness, (2) generating alternative phrasings and suggestions for smoother transitions, and (3) reformatting LaTeX code (tables, figures, equationsand cross-references). All technical ideas, algorithmic designs, theoretical analysesand experimental implementations were developed entirely by the authors without LLM assistance. The LLMs were not used to generate new scientific content, results, or proofs, but served purely as a writing aid.

