# OpenReview forum: "Log-Bit Distributed Learning with Harmonic Modulation"
_ICLR.cc/2026/Conference — Submitted to ICLR 2026_

### Official Review · Reviewer_ZfUi · 2025-10-21

**Soundness:** 3
**Presentation:** 3
**Contribution:** 2
**Rating:** 4
**Confidence:** 4

**Summary:**

The paper proposes Log-Bit Gradient Descent with Harmonic Modulation, a distributed optimization method that dramatically reduces communication cost by compressing each high-dimensional update vector to a single scalar per iteration. Each client projects its local gradient or model difference onto a deterministic sinusoidal “harmonic modulation” direction, quantizes that scalar into a few bits, and transmits it to its neighbors. The authors provide a convergence proof (under certain) assumptions and claim that only O(log d) bits per round are required, where d is the model dimension. Experiments on synthetic problems show comparable accuracy to existing methods while using fewer transmitted bits per iteration.

**Strengths:**

The paper addresses an important bottleneck in distributed and federated optimization, communication cost, and proposes an original compression/quantization framework that is mathematically interesting.

The theoretical analysis is nontrivial and provides an explicit convergence guarantee for an aggressively quantized communication scheme.

Experiments are presented, and comparisons include representative baselines.

**Weaknesses:**

The premise that projecting each d-dimensional update to one scalar suffices effectively requires the projection directions to span all dimensions over time. However, this coverage occurs only asymptotically, over extremely long horizons -- in the Appendix the derivation suggests that this is on the order of d! . It's not clear to me from this analysis that this would be a useful scheme in practice, and I don't think the experiments really get to this point.

Relatedly, while the paper focuses on metrics such as bits per iteration and total bits, this really doesn't seem like the right metric.  Recent work on compression for distributed and federated optimization have highlighted that a more relevant metric in practice is total time until convergence. If the algorithm requires significantly more iterations to reach the same accuracy, or more computation time, this round-level bit metric seems misleading. No timing or iteration-to-accuracy trade-offs are provided, leaving it unclear what the impact would be here beyond the theoretical.

**Questions:**

Please clarify the realistic implications of the “persistent excitation” assumption and whether the harmonic modulation sequence can meaningfully explore the necessary number of dimensions in a reasonable time horizon.  (2d-1)! doesn't seem feasible to me.  Experiments with d=8 are not particularly interesting;  it makes me concerned you can't manage d=11.  At such dimensions, the bits per round don't seem like an actual constraint.
(If this is just meant to be an interesting theoretical paper -- and perhaps judged on that basis it's interesting -- but that doesn't seem to have been the pitch made here.)

Again, if practice is a motivation, please provide empirical comparisons of iteration counts and total runtime to demonstrate real communication efficiency.  It seems also relevant in considering a full system to explain how encoding/decoding overhead and synchronization would affect performance.

---

> ### Author Response · Authors · 2025-11-25
>
> [Answer to Weakness 1]:
>
> Thank you for your insightful comment. We understand your concern regarding the PE condition in Lemma 3.1, where the period $N = (2d - 1)!$ becomes astronomically large. While this represents the theoretical period required for the PE condition, in practice, this is a theoretical artifact, and the actual convergence rate does not depend on this period.
>
> We clarify that both $\alpha_1$ and $N$ are positively proportional to $d$, meaning that although there is a $d!$ term in $N$, it does not practically affect the convergence speed. The reason why we want $\alpha_1$ to be large and $N$ to be small is as follows: A larger $\alpha_1$ indicates a stronger lower bound on the "energy" of the sequence, ensuring that the system remains sufficiently excited. A smaller $N$ indicates a shorter period, meaning the sequence can more quickly return to a state that satisfies the PE condition, thereby accelerating convergence.
>
> Thus, while $\alpha_1$ should be large to ensure strong excitation and $N$ should be small to minimize the period, both are still positively proportional to $d$. This allows for a balance between the two factors that ensures the theoretical PE condition is met without negatively impacting the practical convergence speed.
>
> Additionally, to better understand the impact of different sequences on the algorithm's performance, we have added an analysis of various sequences, such as $ \psi_{e_i}(t) $ and $ \psi_{\text{Gaussian}}(t) $, and their impact on the algorithm’s behavior. As shown in the table, we observe that the choice of $\psi$ does not significantly affect the convergence speed of the algorithm. This reinforces that the PE condition’s theoretical period does not limit the practical effectiveness of our approach.
>
> $$
> \\begin{array}{|l|l|l|}
> \\hline
> \\textbf{Sequence} & \\textbf{Iteration} & \\textbf{Communication cost (MB)} \\\\
> \\hline
> \psi_\text{HarMo} & 25,810 & 2.46 \\\\
> \psi_{e_i}(t) & 24,224 & 2.31 \\\\
> \psi_{\text{Gaussian}}(t) & 25,711 & 2.45 \\\\
> \\hline
> \\end{array}
> $$
>
> [Answer to Weakness 2]:
>
> We appreciate the reviewer’s insightful comment regarding the choice of evaluation metrics. We have reported the total communication cost and runtime across all three experiments — synthetic quadratic optimization, logistic regression, and neural network training. The results show that while LBGD-HarMo does not provide a runtime advantage over uncompressed methods, it achieves a significant reduction in total communication volume, with runtime comparable to other compression or quantization schemes. These findings, as shown in the corresponding tables, illustrate that our method achieves an effective balance between computational efficiency and communication reduction.
>
> $$
> \\begin{array}{|l|l|l|l|l|}
> \\hline
> \\textbf{Algorithm} & \textbf{Method} & \textbf{Communication cost (KB)} & \textbf{Test accuracy (\\%)} & \textbf{Runtime (s)} \\\\
> \\hline
> DSGD & None & 31.25 & 88.44 & 453.67 \\\\
> CHOCO & TOP-\alpha & 6.25 & 88.23 & 957.34 \\\\
> MoTEF & TOP-\alpha & 6.25 & 87.42 & 1497.01 \\\\
> LBGD & Sign & 0.99 & 86.82 & 608.63 \\\\
> \\textbf{LBGD} & \\textbf{HarMo} & \\textbf{0.008} & \\textbf{87.84} & \\textbf{733.49} \\\\
> \\hline
> \\end{array}
> $$
>
> $$
> \\begin{array}{|l|l|l|l|l|}
> \\hline
> \\textbf{Algorithm} & \\textbf{Method} & \textbf{Communication cost (MB)} & \\textbf{Test accuracy (\\%)} & \\textbf{Runtime (min)} \\\\
> \\hline
> FedAvg & None & 804.60 (Central Server) & 89.92 & 99  \\\\
> DSGD & None & 178.80 & 90.09 & 84 \\\\
> CHOCO & TOP-\alpha & 35.76 & 87.03 & 209 \\\\
> LBGD & Sign & 5.33 & 81.94 & 147 \\\\
> \\textbf{LBGD} & \\textbf{HarMo} & \\textbf{0.00007} & \\textbf{86.69} & \\textbf{203} \\\\
> \\hline
> \\end{array}
> $$

---

> ### Author Response · Authors · 2025-11-25
>
> [Answer to Question 1 and 2]:
>
> We sincerely thank the reviewer for these thoughtful comments. Regarding the first point on the practical implications of the persistent excitation (PE) assumption, we clarify that the period $N = (2d - 1)!$ derived in Lemma 3.1 represents a theoretical sufficient condition rather than a practical limitation.  As discussed in Appendix B and Section 5, the PE condition only requires that the excitation sequence spans all dimensions over time. In practice, as long as $\psi(t)$ explores sufficiently diverse directions, convergence occurs well before completing one full theoretical period. As analyzed in [Answer to Weakness 1], our additional experiments in Appendix~E.1.3 compare different sequences (e.g., $\psi_{e_i}(t)$ and $\psi_{\text{Gaussian}}(t)$), showing that the choice of sequence has minimal impact on convergence, confirming that the factorial term does not affect practical feasibility.
>
> Furthermore, experiments with larger dimensions ($d=2000$ in logistic regression and millions of parameters in neural networks) demonstrate that LBGD–HarMo scales effectively in high-dimensional settings.
>
> For the second concern regarding practical deployment and efficiency, we have added detailed runtime and iteration-to-accuracy comparisons in Appendix E.2 and Appendix E.3. These results include total runtime, communication cost, and convergence iterations across various algorithms. Although our algorithm does not offer a runtime advantage compared with standard distributed learning algorithms, it substantially reduces communication overhead while maintaining acceptable computational complexity—this is precisely the motivation behind our design. Our experimental results also show that, compared with other compression or quantization methods, the encoding/decoding overhead of HarMo remains negligible.
>
> To further address this concern, we have added an asynchronous update study in Appendix~E.1.5. In this experiment, we vary both the number of asynchronous clients and their communication frequency. The results show that although reduced synchronization leads to slower consensus and a slight degradation in performance, the algorithm consistently converges.
>
> This analysis clarifies the impact of asynchrony and demonstrates that LBGD–HarMo remains robust even when round-trip synchronization is weakened.
>
> Table 1: Per-client communication cost per iteration and the corresponding test accuracies and runtime for different algorithms under various compressors and quantizers in the Logistic Regression experiment. Experiments are conducted using a Top-α compressor with α = 0.1, Sign quantizer with 1 bit, and HarMo with m = 16 bits, evaluated on n = 9 clients arranged in a ring topology under IID data distribution.
> $$
> \\begin{array}{|l|l|l|l|l|}
> \\hline
> \\textbf{Algorithm} & \textbf{Method} & \textbf{Communication cost (KB)} & \textbf{Test accuracy (\\%)} & \textbf{Runtime (s)} \\\\
> \\hline
> DSGD & None & 31.25 & 88.44 & 453.67 \\\\
> CHOCO & TOP-\alpha & 6.25 & 88.23 & 957.34 \\\\
> MoTEF & TOP-\alpha & 6.25 & 87.42 & 1497.01 \\\\
> LBGD & Sign & 0.99 & 86.82 & 608.63 \\\\
> \\textbf{LBGD} & \\textbf{HarMo} & \\textbf{0.008} & \\textbf{87.84} & \\textbf{733.49} \\\\
> \\hline
> \\end{array}
> $$
>
> Table 2: Per-client communication cost per iteration and the corresponding total training accuracies and runtime for different algorithms under various compressors and quantizers in the Neural Network Training experiment. Experiments are conducted using a Top-α compressor with α = 0.1, Sign quantizer with 1 bit, and HarMo with m = 16 bits, evaluated on n = 9 clients arranged in a ring topology under IID data distribution.
> $$
> \\begin{array}{|l|l|l|l|l|}
> \\hline
> \\textbf{Algorithm} & \\textbf{Method} & \textbf{Communication cost (MB)} & \\textbf{Test accuracy (\\%)} & \\textbf{Runtime (min)} \\\\
> \\hline
> FedAvg & None & 804.60 (Central Server) & 89.92 & 99  \\\\
> DSGD & None & 178.80 & 90.09 & 84 \\\\
> CHOCO & TOP-\alpha & 35.76 & 87.03 & 209 \\\\
> LBGD & Sign & 5.33 & 81.94 & 147 \\\\
> \\textbf{LBGD} & \\textbf{HarMo} & \\textbf{0.00007} & \\textbf{86.69} & \\textbf{203} \\\\
> \\hline
> \\end{array}
> $$
>
> Table 3: Total number of iterations and communication cost under asynchronous updates on the synthetic quadratic optimization problem.
> $$
> \\begin{array}{|l|l|l|}
> \\hline
> \\textbf{Number of asynchronous clients} & \\textbf{Iteration} & \\textbf{Communication cost (MB)} \\\\
> \\hline
> 5 & 28,879 & 2.48 \\\\
> 10 & 29,985 & 2.29 \\\\
> 15 & 37,063 & 2.47 \\\\
> 20 & 41,027 & 2.35 \\\\
> \\hline
> \\textbf{Communication frequency} & \\textbf{Iteration} & \\textbf{Communication cost (MB)} \\\\
> \\hline
> 2 & 29,985 & 2.29 \\\\
> 4 & 42,545 & 2.84 \\\\
> 6 & 56,287 & 3.58 \\\\
> 8 & 69,185 & 4.29 \\\\
> \\hline
> \\end{array}
> $$
>
> We would like to express our gratitude once again for your invaluable feedback. Thank you as well for your patient review.

---

> > ### Comment · Reviewer_ZfUi · 2025-11-25
> >
> > I have seen the responses given by the authors to my and other reviews.  I appreciate their work.  I am planning to keep my score the same.

---

> > > ### Author Response · Authors · 2025-11-27
> > >
> > > We sincerely thank you again for your valuable and constructive feedback, which has been instrumental in improving our work. Building upon these insights, we emphasize that, to the best of our knowledge, LBGD–HarMo is the first distributed learning framework capable of operating under logarithmic communication bit rates, and it represents a breakthrough toward developing scalable and communication-efficient decentralized optimization algorithms.

---

### Official Review · Reviewer_wAuE · 2025-10-25

**Soundness:** 2
**Presentation:** 3
**Contribution:** 1
**Rating:** 2
**Confidence:** 3

**Summary:**

The paper proposes a distributed learning algorithm, which aims to reduce communication costs. The gradient vector is sent from one node to all its neighbor nodes after compression, meaning a projection of a $d$ dimensional vector to a scalar, and (uniform) quantization, meaning representing this scalar with a finite number of bits. The main proposal of the paper is a choice of the projection direction, based on an harmonic vector, which is chosen in advance. The paper demonstrates the performance of the algorithm on two convex problems, showing a significant reduction in communication costs, while maintaining the accuracy.

**Strengths:**

1) The paper is in general easy to follow, and the motivation is clear.

2) The proposed algorithm is rather elegant.

3) The second experiment (logistic regression) shows an impressive savings in communication compared to other algorithms

**Weaknesses:**

1) The algorithm is rather simple, as random projections can recover high dimensional vectors, and quantization with sufficient number of bits will allow the corresponding scalar to be reproduced. It should be discussed, how this algorithm is significantly different from the current state-of-the-art (SOTA) algorithm, and what is the new mechanism that allows to improve.

2) The choice of this specific harmonic vector is stated as a main aspect of the method proposed in the paper. However, how this choice of vector as a harmonic vector affects performance. For example, what will be the performance in case the projection vector is one-hot vector, where the location of the $1$ changes with time? What is the performance for pseudo-random vectors (chosen with some seed) from an isotropic Gaussian? More deeply, this choice is heuristic, and if at all, it is of interest to explain how the choice of the projection vectors is suitable to classes of problems.

3) Algorithm 1, the main algorithm of the paper is neither explained nor discussed. Specifically, lines 6-7 and then 16-18, which are the main updates of the algorithm (beyond the compression-quantization part) are not explained.

4) Theorem 4.1 (the main and only theorem of the paper) states that $O(\log(d))$ bits suffice for an error of $O(1/t)$. Thus the dependence on the number of bits, network topology, step sizes and other parameters of the algorithm is unspecified. In other words, since the dependence on $d$ is unspecified in the convergence rate, one can just propose a round-robin algorithm, where in each step a different coordinate is quantized. It is also difficult to distill a more accurate result from the appendix.

5) The other main aspect of the setting considered in the paper, to wit, distribution of computation across a network, is hardly discussed in the paper: The theoretical result does not display any dependence on the network, and only a single graph in the first experiment shows that the network topology does not significantly affects the convergence (which obviously could be challenged with very difficult topologies).

6) The experiments are only on convex problems. Even if the theoretical guarantees are for convex functions, the algorithm can still be tested on non-convex ones, such as deep neural networks.

7) Since the algorithm is rather simple, the origin of a two-order of improvement achieved in the second case should be discussed in detail. There are many algorithms for federated and distributed learning. Are the baselines represent the SOTA ?

**Questions:**

1) Equation (2) is a rather formulation of the problem. On the left-hand side there is a single variable $x$, and on the right-hand side $n$ variables $x_i$. Why is this necessary?

2) The “persistent excitation” condition in Lemma 3.1 is achieved with rather large constants, on the order of $O((2d)!)$. It is not clear from the results of the paper how these constants affect the performance, but anyway, an exponential dependence on $d\log(d)$ does not scale well.

3) On page 5, the Laplacian is defined with constants $a_{ij}$. I could not find the use of these constants in the algorithm.

4) Line 365 - typo “a'rara”

---

> ### Author Response · Authors · 2025-11-25
>
> [Answer to Weakness 1]:
>
> Thank you for this valuable comment. While the proposed algorithm may appear simple due to the use of random projections and quantization, its key innovation lies in the integration of harmonic modulation within a distributed framework. Unlike existing state-of-the-art approaches, our method employs a harmonic modulation mechanism to encode high-dimensional updates into a single scalar statistic, enabling communication-efficient learning in fully decentralized federated systems.
>
> Specifically, in lines 6--7, inspired by prior work [1], we quantize the error state and introduce a decaying coefficient $g_t$ to scale the transmitted value before and after quantization, thereby reducing quantization noise. Lines 8--10 describe the compression process, where $\boldsymbol{\theta}$ is encoded into a single real-valued scalar through the HarMo Encoder $\\mathcal{C_\\text{E}}$, quantized, and then transmitted as an $m$-bit binary message. Lines 13--14 outline the reconstruction via the HarMo Decoder $\\mathcal{C}_\\text{D}$, which restores the scalar back to a vector in the local coordinate space, allowing accurate local updates. Finally, lines 16--17 introduce a distributed filter $\boldsymbol{\sigma}$ and an integrator $\boldsymbol{z}$, the former tracks local model dynamics, while the latter aggregates scaled differences of neighboring compressed information to balance local and global information flow.
>
> Compared to prior compression-based methods [2,3,4,5], our algorithm not only transmits compressed error states but also harmonically modulates them to ensure information diversity and convergence stability. These design choices collectively distinguish LBGD-HarMo from existing SOTA algorithms, offering a principled mechanism for achieving communication-efficient and stable decentralized optimization.
>
> [1] Yuichi Kajiyama, Naoki Hayashi, and Shigemasa Takai. Linear convergence of consensus-based quantized optimization for smooth and strongly convex cost functions. IEEE Transactions on Automatic Control, 66(3):1254–1261, 2021.
>
> [2] Anastasia Koloskova, Tao Lin, Sebastian U Stich, and Martin Jaggi. Decentralized deep learning with arbitrary communication compression. In International Conference on Learning Representations, 2020a.
>
> [3] Xiaorui Liu, Yao Li, Rongrong Wang, Jiliang Tang, and Ming Yan. Linear convergent decentralized optimization with compression. In International Conference on Learning Representations, 2021.
>
> [4] Xinlei Yi, Shengjun Zhang, Tao Yang, Tianyou Chai, and Karl Henrik Johansson. Communication compression for distributed nonconvex optimization. IEEE Transactions on Automatic Control, 68(9):5477–5492, 2023.
>
> [5] Rustem Islamov, Yuan Gao, and Sebastian U Stich. Towards faster decentralized stochastic optimization with communication compression. In The Thirteenth International Conference on Learning Representations, 2025.
>
> [Answer to Weakness 2]:
>
> Thank you for this insightful question regarding the sensitivity of convergence to the choice of the harmonic sequences. To better understand the impact of different sequences on the algorithm's performance, we have added an analysis of various sequences, such as $ \psi_{e_i}(t) $ and $ \psi_{\text{Gaussian}}(t) $, and their impact on the algorithm’s behavior. As shown in the table, we observe that the choice of $\psi$ does not significantly affect the convergence speed of the algorithm. This reinforces that, as long as the PE condition is satisfied, any sequence fulfilling this condition will be effective for the algorithm’s performance. Therefore, the specific choice of harmonic sequence does not impose a practical limitation on the convergence rate or overall effectiveness of the approach.
>
> $$
> \\begin{array}{|l|l|l|}
> \\hline
> \\textbf{Sequence} & \\textbf{Iteration} & \\textbf{Communication cost (MB)} \\\\
> \\hline
> \psi_\text{HarMo} & 25,810 & 2.46 \\\\
> \psi_{e_i}(t) & 24,224 & 2.31 \\\\
> \psi_{\text{Gaussian}}(t) & 25,711 & 2.45 \\\\
> \\hline
> \\end{array}
> $$

---

> ### Author Response · Authors · 2025-11-25
>
> [Answer to Weakness 3]:
>
> Thank you for your valuable comment. You are correct that we have not explained Algorithm 1 in enough detail, particularly regarding lines 6–7 and 16–18, which are critical to the algorithm’s core operations beyond compression and quantization. To address this, in lines 6–7, we quantize the error state and introduce a decaying coefficient $g_t$ to scale the transmitted value before and after quantization, effectively reducing quantization noise. Lines 8–10 describe the compression process, where the local model parameter $\\boldsymbol{\theta}$ is encoded into a single real-valued scalar through the HarMo Encoder $\\mathcal{C}_\\text{E}$ quantized, and then transmitted as an $m$-bit binary message.
>
> Lines 13–14 outline the reconstruction process using the HarMo Decoder $\\mathcal{C}_\\text{D}$, which restores the scalar back into a vector in the local coordinate space, allowing accurate local updates. Finally, lines 16–17 introduce a distributed filter $\\boldsymbol{\sigma}$ and an integrator $\\boldsymbol{z}$, where the filter tracks the local model dynamics and the integrator aggregates scaled differences of neighboring compressed information to balance local and global information flow.
>
> [Answer to Weakness 4]:
>
> Thank you for the insightful comment. As you rightly pointed out, while Theorem 4.1 provides a theoretical bound on the number of bits required, we have not fully specified the dependence on various parameters such as network topology, step sizes, and other hyperparameters. We will provide a comprehensive analysis of the convergence rates with respect to these parameters in future work. Due to time constraints, we will prioritize revisiting the proofs and refining the theoretical framework to present a more detailed and rigorous understanding of how these parameters influence convergence as soon as possible.
>
> In the meantime, we have supplemented our work with extensive experimental results that demonstrate the performance of our algorithm across various configurations, including different numbers of clients, communication topologies, and data distributions. These results provide empirical evidence of the effectiveness and robustness of our approach, showcasing its performance in practical scenarios.
>
> In summary, while the theoretical analysis in Theorem 4.1 provides a bound on the number of bits, the actual performance of the algorithm remains robust to variations in topology, step sizes, and hyperparameters. The results in Appendix E.1.5 and the experiments in Section 5 further validate this observation, confirming that LBGD-HarMo can efficiently operate even with large dimensions and varying network configurations.
>
> [Answer to Weakness 5]:
>
> Thank you for the insightful comment. Indeed, we acknowledge that the distribution of computation across the network and its impact on the algorithm's performance is an important consideration. While the theoretical result in our paper does not explicitly incorporate the network topology, we do utilize the Laplacian matrix in both the algorithm and its derivation, which inherently ties the algorithm’s parameters to the network structure. This establishes a relationship between the topology and the performance of the algorithm.
>
> However, as demonstrated in our experiments, while the topology affects the performance, it does not significantly impact the convergence speed. We have addressed this in Appendix E.1.4, where we include experiments with time-varying topologies. The results show that the algorithm still performs well in these settings, further validating the robustness of LBGD-HarMo. Additionally, in Table 5, we report the number of iterations required for convergence to the same error under different topologies. These experiments emphasize that, while network structure may influence the rate of convergence, the algorithm remains stable and effective across varying topologies.
>
> Nonetheless, we recognize that the relationship between network topology and convergence speed remains an important area for further research, and we plan to explore this topic as soon as possible.
>
> [Answer to Weakness 6]:
>
> We thank the reviewer for pointing out the importance of testing our algorithm on non-convex problems. In response, we have conducted additional experiments in Section 5.4 and Appendix E.3, where we train neural networks, including ResNet-18, on non-convex objectives. The results demonstrate that our proposed algorithm performs well in non-convex settings, confirming its robustness beyond convex problems. This further supports the versatility and effectiveness of LBGD-HarMo in more complex, real-world optimization tasks.

---

> ### Author Response · Authors · 2025-11-25
>
> [Answer to Weakness 7]:
>
> We appreciate the reviewer’s insightful comment. We have provided detailed information about the second experiment in Appendix E.2, including iteration counts, runtime, communication cost, and test accuracy. Our experiments follow the same setup as in previous works such as Koloskova et al., 2020 [1], Zhao et al., 2022 [2], and Islamov et al., 2025 [3]. DSGD is a classic baseline for decentralized distributed optimization, while CHOCO is a well-established decentralized federated learning algorithm with compression. Many works use these methods for comparison. MoTEF, which was published at ICLR last year, is one of the state-of-the-art algorithms, and therefore, we selected these three algorithms for comparison.
>
> Additionally, we are continuously exploring more SOTA algorithms in this field, and we plan to conduct further comparison experiments in future work.
>
> [1] Anastasia Koloskova, Tao Lin, Sebastian U Stich, and Martin Jaggi. Decentralized deep learning with arbitrary communication compression. In International Conference on Learning Representations, 2020a.
>
> [2] Haoyu Zhao, Boyue Li, Zhize Li, Peter Richtárik, and Yuejie Chi. Beer: Fast o(1/t) rate for decentralized nonconvex optimization with communication compression. Advances in Neural Information Processing Systems, 2022.
>
> [3] Rustem Islamov, Yuan Gao, and Sebastian U Stich. Towards faster decentralized stochastic optimization with communication compression. In The Thirteenth International Conference on Learning Representations, 2025.
>
> Table: Total number of iterations and communication cost under different experimental settings on the synthetic quadratic optimization problem.
> $$
> \\begin{array}{|l|l|l|}
> \\hline
> \\textbf{Number of clients} & \\textbf{Iteration} & \\textbf{Communication cost (MB)}  \\\\
> \\hline
> 9 & 39,472 & 1.36 \\\\
> 25 & 25,810 & 2.46 \\\\
> 64 & 28,020 & 6.84 \\\\
> 100 & 36,713 & 14.00 \\\\
> \\hline
> \\textbf{Topology} & \\textbf{Iteration} & \\textbf{Communication cost (MB)} \\\\
> \\hline
> Fully-connected & 9,550 & 10.93 \\\\
> Torus & 28,523 & 5.44 \\\\
> Fixed ER & 38,233 & 4.38 \\\\
> Ring & 25,810 & 2.46 \\\\
> Time-varying ER & 79,554 & 9.10 \\\\
> \\hline
> \\textbf{Dimension} & \\textbf{Iteration} & \\textbf{Communication cost (MB)} \\\\
> \\hline
> 4 & 17,056 & 1.67 \\\\
> 8 & 25,810 & 2.46 \\\\
> 16 & 31,354 & 2.99 \\\\
> 100 & 33,105 & 3.16 \\\\
> \\hline
> \\textbf{Sequence} & \\textbf{Iteration} & \\textbf{Communication cost (MB)} \\\\
> \\hline
> \psi_\text{HarMo}& 25,810 & 2.46 \\\\
> \psi_{e_i}& 24,224 & 2.31 \\\\
> \psi_{\text{Gaussian}} & 25,711 & 2.45 \\\\
> \\hline
> \\textbf{Number of asynchronous clients} & \\textbf{Iteration} & \\textbf{Communication cost (MB)} \\\\
> \\hline
> 5 & 28,879 & 2.48 \\\\
> 10 & 29,985 & 2.29 \\\\
> 15 & 37,063 & 2.47 \\\\
> 20 & 41,027 & 2.35 \\\\
> \\hline
> \\textbf{Comm. frequency} & \\textbf{Iteration} & \\textbf{Communication cost (MB)} \\\\
> \\hline
> 2 & 29,985 & 2.29 \\\\
> 4 & 42,545 & 2.84 \\\\
> 6 & 56,287 & 3.58 \\\\
> 8 & 69,185 & 4.29 \\\\
> \\hline
> \\textbf{Quantization precision (bit)} & \\textbf{Iteration} & \\textbf{Communication cost (MB)} \\\\
> \\hline
> 3 & 47,653 & 4.54 \\\\
> 4 & 38,717 & 3.69 \\\\
> 8 & 25,810 & 2.46 \\\\
> 16 & 24,789 & 4.73 \\\\
> \\hline
> \\end{array}
> $$
> Table: Total number of iterations and communication cost under different hyperparameter values on the synthetic quadratic optimization problem.
> $$
> \\begin{array}{|l|l|l|}
> \\hline
> \\textbf{Value of hyperparameter $\beta$} & \\textbf{Iteration} & \\textbf{Communication cost (MB)} \\\\
> \\hline
> 0.4 & 56,568 & 5.39 \\\\
> 0.6 & 38,639 & 3.68 \\\\
> 0.8 & 29,732 & 2.84 \\\\
> 1.0 & 25,810 & 2.46 \\\\
> \\hline
> \\textbf{Value of hyperparameter $\eta$} & \\textbf{Iteration} & \\textbf{Communication cost (MB)} \\\\
> \\hline
> 1.0 & 56,686 &5.41 \\\\
> 2.5 & 40,103 &3.82 \\\\
> 3.8 & 26,543 &2.53 \\\\
> 5.0 & 25,810 &2.46 \\\\
> \\hline
> \\textbf{Value of hyperparameter $g_0$} & \\textbf{Iteration} & \\textbf{Communication cost (MB)} \\\\
> \\hline
> 4 & 26,230 & 2.50 \\\\
> 6 & 26,121 & 2.49 \\\\
> 8 & 25,880 & 2.47 \\\\
> 10 & 25,810 & 2.46 \\\\
> \\hline
> \\textbf{Value of hyperparameter $\gamma$} & \\textbf{Iteration} & \\textbf{Communication cost (MB)} \\\\
> \\hline
> 0.999 & divergence & / \\\\
> 0.9999 & 25,810 & 2.46 \\\\
> 0.99999 & 25,907 & 2.47 \\\\
> 0.999999 & 25,908 & 2.47 \\\\
> \\hline
> \\textbf{Value of hyperparameter $\kappa$} & \\textbf{Iteration} & \\textbf{Communication cost (MB)} \\\\
> \\hline
> 0.010 & 84,927 & 8.10 \\\\
> 0.025 & 70,111 & 6.69 \\\\
> 0.038 & 32,599 & 3.11 \\\\
> 0.050 & 25,810 & 2.46 \\\\
> \\hline
> \\textbf{Value of hyperparameter $\kappa_0$} & \\textbf{Iteration} & \\textbf{Communication cost (MB)} \\\\
> \\hline
> 0.0010 & 27,100 & 2.58 \\\\
> 0.0025 & 26,583 & 2.54 \\\\
> 0.0038 & 26,486 & 2.53 \\\\
> 0.0050 & 25,810 & 2.46 \\\\
> \\hline
> \\textbf{Quantization precision (bit)} & \\textbf{Iteration} & \\textbf{Communication cost (MB)} \\\\
> \\hline
> 3 & 47,653 & 4.54 \\\\
> 4 & 38,717 & 3.69 \\\\
> 8 & 25,810 & 2.46 \\\\
> 16 & 24,789 & 4.73 \\\\
> \\hline
> \\end{array}
> $$

---

> ### Author Response · Authors · 2025-11-25
>
> [Answer to Question 1]:
>
> We thank the reviewer for this thoughtful question.
> In our formulation, the global variable $\boldsymbol{x}$ represents the shared model parameter that the system aims to optimize collaboratively, while each $\boldsymbol{x}_i$ corresponds to the local model maintained by client $i$.
> The constraint $\boldsymbol{x}_i = \boldsymbol{x}_j, \ \forall i,j \in \mathcal{V},$ enforces consensus among all clients, ensuring that their local updates converge toward a common global model.This formulation follows the standard consensus optimization framework widely adopted in decentralized and federated learning, where each client performs local computation on its private data while cooperating to achieve global agreement. The distinction between the single variable $\boldsymbol{x}$ on the left-hand side and the local variables $\boldsymbol{x}_i$ on the right-hand side explicitly captures this distributed coordination mechanism, which is essential for achieving convergence without centralized aggregation.
>
> [Answer to Question 2]:
>
> Thank you for your insightful comment. We understand your concern regarding the PE condition in Lemma 3.1, where the period $N = (2d - 1)!$ becomes astronomically large. While this represents the theoretical period required for the PE condition, in practice, this is a theoretical artifact, and the actual convergence rate does not depend on this period.
>
> We clarify that both $\alpha_1$ and $N$ are positively proportional to $d$, meaning that although there is a $d!$ term in $N$, it does not practically affect the convergence speed. The reason why we want $\alpha_1$ to be large and $N$ to be small is as follows: A larger $\alpha_1$ indicates a stronger lower bound on the "energy" of the sequence, ensuring that the system remains sufficiently excited. A smaller $N$ indicates a shorter period, meaning the sequence can more quickly return to a state that satisfies the PE condition, thereby accelerating convergence.
>
> Thus, while $\alpha_1$ should be large to ensure strong excitation and $N$ should be small to minimize the period, both are still positively proportional to $d$. This allows for a balance between the two factors that ensures the theoretical PE condition is met without negatively impacting the practical convergence speed.
>
> Additionally, to better understand the impact of different sequences on the algorithm's performance, we have added an analysis of various sequences, such as $ \psi_{e_i}(t) $ and $ \psi_{\text{Gaussian}}(t) $, and their impact on the algorithm’s behavior. As shown in the table, we observe that the choice of $\psi$ does not significantly affect the convergence speed of the algorithm. This reinforces that the PE condition’s theoretical period does not limit the practical effectiveness of our approach.
>
> $$
> \\begin{array}{|l|l|l|}
> \\hline
> \\textbf{Sequence} & \\textbf{Iteration} & \\textbf{Communication cost (MB)} \\\\
> \\hline
> \psi_\text{HarMo} & 25,810 & 2.46 \\\\
> \psi_{e_i}(t) & 24,224 & 2.31 \\\\
> \psi_{\text{Gaussian}}(t) & 25,711 & 2.45 \\\\
> \\hline
> \\end{array}
> $$
>
> [Answer to Question 3 and 4]:
>
> Thank you for pointing out this detail. We appreciate the reviewer’s careful reading and constructive feedback. We have updated Algorithm 1 and included the constants aij in line 17, as they are essential for tracking the interactions between neighboring clients in the communication graph. These constants help define the weighted connections between nodes, which plays a crucial role in the distributed updates and consensus mechanism. Moreover, we have corrected the spelling error ”a’rara” to ”are”.
>
> We would like to express our gratitude once again for your invaluable feedback. Thank you as well for your patient review.

---

> ### Author Response · Authors · 2025-11-25
>
> Table 1: Per-client communication cost per iteration and the corresponding test accuracies and runtime for different algorithms under various compressors and quantizers in the Logistic Regression experiment. Experiments are conducted using a Top-α compressor with α = 0.1, Sign quantizer with 1 bit, and HarMo with m = 16 bits, evaluated on n = 9 clients arranged in a ring topology under IID data distribution.
> $$
> \\begin{array}{|l|l|l|l|l|}
> \\hline
> \\textbf{Algorithm} & \textbf{Method} & \textbf{Communication cost (KB)} & \textbf{Test accuracy (\\%)} & \textbf{Runtime (s)} \\\\
> \\hline
> DSGD & None & 31.25 & 88.44 & 453.67 \\\\
> CHOCO & TOP-\alpha & 6.25 & 88.23 & 957.34 \\\\
> MoTEF & TOP-\alpha & 6.25 & 87.42 & 1497.01 \\\\
> LBGD & Sign & 0.99 & 86.82 & 608.63 \\\\
> \\textbf{LBGD} & \\textbf{HarMo} & \\textbf{0.008} & \\textbf{87.84} & \\textbf{733.49} \\\\
> \\hline
> \\end{array}
> $$
> Table 2: Per-client communication cost per iteration and the corresponding total training accuracies and runtime for different algorithms under various compressors and quantizers in the Neural Network Training experiment. Experiments are conducted using a Top-α compressor with α = 0.1, Sign quantizer with 1 bit, and HarMo with m = 16 bits, evaluated on n = 9 clients arranged in a ring topology under IID data distribution.
> $$
> \\begin{array}{|l|l|l|l|l|}
> \\hline
> \\textbf{Algorithm} & \\textbf{Method} & \textbf{Communication cost (MB)} & \\textbf{Test accuracy (\\%)} & \\textbf{Runtime (min)} \\\\
> \\hline
> FedAvg & None & 804.60 (Central Server) & 89.92 & 99  \\\\
> DSGD & None & 178.80 & 90.09 & 84 \\\\
> CHOCO & TOP-\alpha & 35.76 & 87.03 & 209 \\\\
> LBGD & Sign & 5.33 & 81.94 & 147 \\\\
> \\textbf{LBGD} & \\textbf{HarMo} & \\textbf{0.00007} & \\textbf{86.69} & \\textbf{203} \\\\
> \\hline
> \\end{array}
> $$

---

> > ### Comment · Reviewer_wAuE · 2025-11-26
> >
> > Thank you for the detailed answer (though somewhat long). My personal perspective for reviewing the paper is the conceptual idea - the harmonic projection of the gradient vector. While I find the proposed idea rather simple, it is possible that this wasn't proposed in a distributed learning framework, and this paper is novel. I also appreciate the significant savings in communication costs. Thus, the paper could contribute to the decentralized learning literature.
> > Nonetheless, as in my original comment, from my perspective, the idea is both simple and underdeveloped, especially in terms of depth:
> > 1) In your response to W2: This lack of sensitivity shows that it doesn't really matter that the vector is harmonic. Yet "Harmonic modulation" appears in the title of the paper... The idea of dimensionality reduction is so prevalent in ML, that in case that dimensionality reduction is the novel core contribution of the paper, it is expected that this initial idea should explore the multitude of dimensionality reduction techniques available. In case, dimensionality reduction of the gradient was already proposed in distributed or federated learning, then this work should emphasize what is unique in projections to harmonic vectors.
> > 2) The theoretical aspects are deferred to future work, and the existing bound is, as said, do not really capture the quality of dimensionality reduction technique, since even simple methods would be similarly bounded.
> > 3) In principle, the idea of harmonic modulation can be applied to a single client. Similarly, the analysis of network topology can be analyzed without harmonic modulation. If you include in your paper the later, then understanding the *interaction* between them is the interesting aspect, rather than extensively simulating many possibilities. For example, a guideline on how the topology affects the required compression or quantization

---

> > > ### Author Response · Authors · 2025-11-27
> > >
> > > We would like to express our sincere gratitude to the reviewer once again for the constructive and insightful feedback, which has been highly valuable in improving the quality and clarity of our manuscript.
> > >
> > > To begin with, we note that the sinusoidal sequence inherently satisfies the Persistent Excitation (PE) condition, ensuring that the projection directions span the entire vector space over time. This property guarantees that all dimensions are sufficiently excited. We specifically adopt the sinusoidal sequence instead of other alternatives such as the unit-basis sequence $\psi_{e_i}(t)$ and the Gaussian random sequence $\psi_{\text{Gaussian}}(t)$ for several reasons [1].
> > >
> > > First, in high-dimensional settings, $\psi_{e_i}(t)$ activates only one coordinate at a time, resulting in sparse and discontinuous excitation, whereas $\psi_{\text{Gaussian}}(t)$ introduces random fluctuations that may cause instability during decoding. In contrast, the sinusoidal sequence provides smooth projections that enhance numerical stability and robustness of convergence, even in large-scale systems. Furthermore, it offers an analytically invertible and reproducible projection structure, enabling deterministic decoding without additional synchronization. Although $\psi_{e_i}(t)$ is theoretically deterministic, it conveys information inefficiently in high dimensions, and $\psi_{\text{Gaussian}}(t)$ remains non-reproducible without shared random seeds. Finally, the sinusoidal sequence can be recursively generated with $O(d)$ computational cost and constant memory, avoiding the need for random matrix generation or cross-client synchronization. Collectively, these properties make the sinusoidal sequence a practically efficient and theoretically sound choice for harmonic modulation.
> > >
> > > In addition, the table summarizes the communication costs of three representative dimensionality reduction schemes. We compare two mainstream gradient-reduction methods, represented by FetchSGD with Sketch [2] and PowerSGD with Low-Rank [3]. In these methods, the gradient $\boldsymbol{g} \in \mathbb{R}^d$ is directly compressed from $\mathbb{R}^d$ to a lower-dimensional space $\mathbb{R}^c$. Specifically, FetchSGD employs a Count Sketch structure of size $b \times c$, where $b$ denotes the number of hash functions (i.e., repetitions) and $c$ is the number of buckets. PowerSGD instead performs a low-rank approximation of the gradient matrix with rank $r$, transmitting the product of two smaller matrices of size $p \times r$ and $r \times n$. In both cases, the compression dimensions $c$, $r$, $p$, and $n$ are significantly smaller than the original dimension $d$.
> > >
> > > By contrast, our proposed $\\textbf{LBGD-HarMo}$ algorithm compresses the $d$-dimensional update directly into an $m$-bit binary representation through harmonic modulation. Rather than quantizing the gradient itself, our method quantizes the error state $(\boldsymbol{x} - \boldsymbol{\sigma}) / g_t$, where $\boldsymbol{\sigma}$ denotes a distributed filter tracking the local state $\boldsymbol{x}$, and $g_t$ is a decaying coefficient used to scale the transmitted value before and after quantization to reduce quantization error. Moreover, a distributed integrator $\boldsymbol{z}$ is introduced to aggregate the scaled differences from neighboring nodes, enabling a balance between local and global information. Overall, this design fundamentally differs from direct gradient compression and achieves high communication efficiency while maintaining stable convergence.
> > >
> > > Our LBGD-HarMo is the first distributed learning framework that operates under logarithmic bit rates, thereby opening new avenues for both theoretical investigation and practical deployment. We believe this constitutes a fundamental breakthrough in the design of communication-efficient decentralized optimization algorithms.
> > >
> > > \\begin{array}{|l|l|l|}
> > > \\hline
> > > \\textbf{Algorithm} & \textbf{Method} & \textbf{Communication cost (bit)} \\\\
> > > \\hline
> > > FetchSGD & Sketch & 32bc  \\\\
> > > PowerSGD & Low-Rank & 32r(p+n) \\\\
> > > \\textbf{LBGD} & \\textbf{HarMo} & \\textbf{m} \\\\
> > > \\hline
> > > \\end{array}
> > >
> > > [1] Zhu Y. Multivariable system identification for process control. Elsevier, 2001.
> > >
> > > [2]  T. Vogels, S. P. Karimireddy, and M. Jaggi. Powersgd: Practical low-rank gradient compression for distributed optimization. Advances in Neural Information Processing Systems, 32, 2019.
> > >
> > > [3]  D. Rothchild, A. Panda, E. Ullah, N. Ivkin, I. Stoica, V. Braverman, J. Gonzalez, and R. Arora. Fetchsgd: Communication-efficient federated learning with sketching. In International Conference on Machine Learning, pages 8253–8265. PMLR, 2020.

---

> > > ### Author Response · Authors · 2025-11-27
> > >
> > > Finally, regarding the influence of network topology on the convergence rate, as shown in Equation (C.19), for $t \ge t_0 = \max\\{\frac{3\xi_2\eta}{\xi_1\beta}, \frac{6\xi_4'\eta}{r c_3}, \frac{6p}{\kappa\xi_1\beta}, \frac{12p c_2}{\kappa c_3}\\} - 1$, the Lyapunov function satisfies $\Delta V_t \le -\frac{2V_t}{t+1} + C_1 g_t^2 \|\delta_t\|^2$, which implies that once $t \ge t_0$, the convergence rate becomes $O(1/t)$ and is independent of other parameters. Since $\xi_1 = \frac{\lambda_2}{2}, \xi_2 = \frac{1 + L_{\mathcal{H}}^2}{2} + \frac{n^2 L_{\mathcal{H}}^4}{\mu^2}, \xi_3 = 2c_4\sqrt{n}\lambda_n^2 + \sqrt{n}L_{\mathcal{H}}^2 + \frac{4n^{\tfrac{5}{2}}L_{\mathcal{H}}^4}{\mu^2}, \xi_4 = c_4\sqrt{n}, \xi_4' = c_4^2\sqrt{n}$, the first and third terms are proportional to $\lambda_n/\lambda_2$, known as the Laplacian condition number. A smaller $\lambda_n/\lambda_2$ indicates better network connectivity, resulting in a smaller $t_0$ and hence faster convergence.
> > >
> > > We sincerely thank you again for your valuable comments, and we will continue to improve our work accordingly.

---

### Official Review · Reviewer_gMoS · 2025-10-29

**Soundness:** 3
**Presentation:** 3
**Contribution:** 2
**Rating:** 6
**Confidence:** 4

**Summary:**

This paper proposes a novel communication-efficient algorithm for decentralized distributed learning called Log-Bit Gradient Descent with Harmonic Modulation. The core problem addressed is the high communication complexity in distributed learning, where agents must exchange high-dimensional vectors. The key innovation is Harmonic Modulation , a compression scheme where each $d$-dimensional vector is compressed into a single scalar using a time-varying, deterministic harmonic projection sequence. This scalar is then quantized into an $m$-bit binary code for transmission. The authors prove that this method achieves an optimal $\mathcal{O}(1/t)$ convergence rate for strongly convex problems, while theoretically only requiring $m \geq \mathcal{O}(\log_2(d))$ bits per communication. Experiments on quadratic optimization and logistic regression validate the approach, showing that LBGd-HarMo achieves comparable accuracy to baselines (like DSGD and CHOCO) while reducing communication by up to 800x-4000x bits per iteration.

**Strengths:**

1.The primary strength is the originality of the Harmonic Modulation scheme. Compressing a $d$-dimensional vector to a single $m$-bit scalar 31via a deterministic, time-varying projection 32 is a fundamentally new approach to communication compression in this field. It breaks from the standard paradigms of sparsification or $d$-dimensional quantization.

2.Strong and interpretable theory: an explicit $\mathcal{O}(1/t)$ rate under standard convexity and graph assumptions with a sufficient condition $m = \Omega(\log_2 d)$ that formalizes log-bit communication sufficiency for convergence.

3.The paper is a model of clarity. Figure 1 perfectly illustrates the method. The experiments are thorough, testing robustness to client count, network topology, and quantization precision (Figure 2). The clear separation of performance vs. iterations (Figure 4a) and vs. communicated bits (Figure 4b) makes the contribution unambiguous.

**Weaknesses:**

1.The proof of the PE condition (Lemma 3.1, Appendix B) derives a period $N = (2d - 1)!$. This is an astronomically large number. While this appears to be a sufficient period for the proof and does not seem to affect the final $\mathcal{O}(1/t)$ rate, it is a somewhat jarring theoretical artifact. It's unclear if this has any practical implications or if it's purely a limitation of the current analysis. The more important result is the $m \geq \mathcal{O}(\log_2(d))$ bit-rate, which is beautifully supported by the experiment in Figure 2c.


2.Communication accounting lacks clarity. The reported communication volume (in KB) is aggregated across links and nodes in a way that does not clearly align with the stated per-message costs for $Top-\alpha$ and Sign. A precise definition is needed to specify whether communication is measured per edge, per node, or as a network total, along with consistent measurement across all methods.

3.Tuning complexity and stability margins. The algorithm introduces several hyperparameters ($\kappa, \kappa_0, \eta, \alpha, g_0, \gamma$), and while tuned fairly, guidance on principled selection to mis-specification would improve usability.

**Questions:**

1.Could you comment on the computational overhead of the HarMo encoder/decoder? As noted in the weaknesses, this involves $d$ $\sin$ operations and two $d$-dimensional vector operations at each step $t$. How does this computational cost scale, and how does it compare to the cost of the local gradient computation? Is there a point (a very large $d$) where this computation is no longer negligible?

2.How sensitive is convergence to the choice of the harmonic schedule; e.g., would randomized orthogonal directions or structured Hadamard projections yield similar guarantees with better constants than the factorial PE window?

3.Could you add experiments on time-varying graphs or mild nonconvex objectives (e.g., two-layer networks) to assess whether the log-bit mechanism and error-feedback variables $\sigma\$ and $z$ remain effective beyond the strongly convex regime?

---

> ### Author Response · Authors · 2025-11-25
>
> [Answer to Weakness 1]:
>
> Thank you for your insightful comment. We understand your concern regarding the PE condition in Lemma 3.1, where the period $N = (2d - 1)!$ becomes astronomically large. While this represents the theoretical period required for the PE condition, in practice, this is a theoretical artifact, and the actual convergence rate does not depend on this period.
>
> We clarify that both $\alpha_1$ and $N$ are positively proportional to $d$, meaning that although there is a $d!$ term in $N$, it does not practically affect the convergence speed. The reason why we want $\alpha_1$ to be large and $N$ to be small is as follows: A larger $\alpha_1$ indicates a stronger lower bound on the "energy" of the sequence, ensuring that the system remains sufficiently excited. A smaller $N$ indicates a shorter period, meaning the sequence can more quickly return to a state that satisfies the PE condition, thereby accelerating convergence.
>
> Thus, while $\alpha_1$ should be large to ensure strong excitation and $N$ should be small to minimize the period, both are still positively proportional to $d$. This allows for a balance between the two factors that ensures the theoretical PE condition is met without negatively impacting the practical convergence speed.
>
> Additionally, to better understand the impact of different sequences on the algorithm's performance, we have added an analysis of various sequences, such as $ \psi_{e_i}(t) $ and $ \psi_{\text{Gaussian}}(t) $, and their impact on the algorithm’s behavior. As shown in the table, we observe that the choice of $\psi$ does not significantly affect the convergence speed of the algorithm. This reinforces that the PE condition’s theoretical period does not limit the practical effectiveness of our approach.
>
> $$
> \\begin{array}{|l|l|l|}
> \\hline
> \\textbf{Sequence} & \\textbf{Iteration} & \\textbf{Communication cost (MB)} \\\\
> \\hline
> \psi_\text{HarMo} & 25,810 & 2.46 \\\\
> \psi_{e_i}(t) & 24,224 & 2.31 \\\\
> \psi_{\text{Gaussian}}(t) & 25,711 & 2.45 \\\\
> \\hline
> \\end{array}
> $$
>
> [Answer to Weakness 2]:
>
> Thank you for your comment. We have clarified the definition of communication cost in the revised manuscript. The communication cost per node includes both sent and received information from neighboring nodes. Each value is measured in 32-bit floating-point numbers, occupying 4 bytes. The per-iteration communication cost for a single node is the number of neighbors multiplied by 2 (for receiving and sending), along with the cost of the selected compressor/quantizer. The total communication cost for the entire network is the cumulative cost across all nodes per iteration. The specific communication costs for each compression/quantization method are summarized in the table below.
>
> $$
> \\begin{array}{|l|l|l|l|}
> \\hline
> \\textbf{Method}&\\textbf{Communication Cost (bit)}&\\textbf{Computational Complexity}&\\textbf{Storage Complexity} \\\\
> \\hline
> Top-\\alpha&64\\lceil\\alpha d\\rceil&O(d\log d)&O(d) \\\\
> Sign & d + 32 &O(d)& O(d) \\\\
> HarMo & m & O(d) & O(d) \\\\
> \\hline
> \\end{array}
> $$

---

> ### Author Response · Authors · 2025-11-25
>
> [Answer to Weakness 3]:
>
> We would like to thank the reviewer for the insightful comment. In response, we have addressed this concern in two parts. First, Appendix E.1.5 presents a detailed sensitivity analysis for the parameters $\beta$, $\eta$, $g_0$, $\gamma$, $\kappa$, and $\kappa_0$, while Figure 2(c) further examines how the choice of $m$ influences the communication-performance tradeoff of LBGD-HarMo. These results collectively provide practical guidance: step-size parameters ($\eta$, $g_0$, $\gamma$) can be tuned using standard learning rate heuristics, while the consensus-related parameters ($\kappa$, $\kappa_0$) demonstrate reasonable robustness within appropriate ranges, as suggested by their stable empirical behavior. In practice, selecting $m$ between 8 and 16 achieves a reliable balance between communication efficiency and convergence. Our empirical results indicate that LBGD-HarMo maintains stable performance over moderate variations of all hyperparameters. In prior work, we proved the impact of these parameters on the convergence of the algorithm in Appendix C. Additionally, we will provide a detailed analysis of the convergence rates with respect to these parameters in future work.
>
> The table presents the total number of iterations and communication cost under various hyperparameter values on the synthetic quadratic optimization problem to reach an optimality error of $10^{−3}$.The results are consistent with the parameter sensitivity analysis presented in Appendix E.1.1, showing that our algorithm exhibits robustness across a range of hyperparameter values.
>
> $$
> \\begin{array}{|l|l|l|}
> \\hline
> \\textbf{Value of hyperparameter $\beta$} & \\textbf{Iteration} & \\textbf{Communication cost (MB)} \\\\
> \\hline
> 0.4 & 56,568 & 5.39 \\\\
> 0.6 & 38,639 & 3.68 \\\\
> 0.8 & 29,732 & 2.84 \\\\
> 1.0 & 25,810 & 2.46 \\\\
> \\hline
> \\textbf{Value of hyperparameter $\eta$} & \\textbf{Iteration} & \\textbf{Communication cost (MB)} \\\\
> \\hline
> 1.0 & 56,686 & 5.41 \\\\
> 2.5 & 40,103 & 3.82 \\\\
> 3.8 & 26,543 & 2.53 \\\\
> 5.0 & 25,810 & 2.46 \\\\
> \\hline
> \\textbf{Value of hyperparameter $g_0$} & \\textbf{Iteration} & \\textbf{Communication cost (MB)} \\\\
> \\hline
> 4 & 26,230 & 2.50 \\\\
> 6 & 26,121 & 2.49 \\\\
> 8 & 25,880 & 2.47 \\\\
> 10 & 25,810 & 2.46 \\\\
> \\hline
> \\textbf{Value of hyperparameter $\gamma$} & \\textbf{Iteration} & \\textbf{Communication cost (MB)} \\\\
> \\hline
> 0.999 & divergence & / \\\\
> 0.9999 & 25,810 & 2.46 \\\\
> 0.99999 & 25,907 & 2.47 \\\\
> 0.999999 & 25,908 & 2.47 \\\\
> \\hline
> \\textbf{Value of hyperparameter $\kappa$} & \\textbf{Iteration} & \\textbf{Communication cost (MB)} \\\\
> \\hline
> 0.010 & 84,927 & 8.10 \\\\
> 0.025 & 70,111 & 6.69 \\\\
> 0.038 & 32,599 & 3.11 \\\\
> 0.050 & 25,810 & 2.46 \\\\
> \\hline
> \\textbf{Value of hyperparameter $\kappa_0$} & \\textbf{Iteration} & \\textbf{Communication cost (MB)} \\\\
> \\hline
> 0.0010 & 27,100 & 2.58 \\\\
> 0.0025 & 26,583 & 2.54 \\\\
> 0.0038 & 26,486 & 2.53 \\\\
> 0.0050 & 25,810 & 2.46 \\\\
> \\hline
> \\textbf{Quantization precision (bit)} & \\textbf{Iteration} & \\textbf{Communication cost (MB)} \\\\
> \\hline
> 3 & 47,653 & 4.54 \\\\
> 4 & 38,717 & 3.69 \\\\
> 8 & 25,810 & 2.46 \\\\
> 16 & 24,789 & 4.73 \\\\
> \\hline
> \\end{array}
> $$

---

> ### Author Response · Authors · 2025-11-25
>
> [Answer to Question 1]:
>
> Thank you for your thoughtful question regarding the computational overhead of the HarMo encoder/decoder.
> As discussed in [Answer to Weakness 1] and [Answer to Weakness 2], we have provided a detailed comparison of the communication cost, computational complexity, and storage complexity of HarMo, Top-$\alpha$, and Sign in Appendix~D.2. This comparison confirms that HarMo’s per-iteration computational complexity remains linear in $d$, which is comparable to existing methods such as Top-$\alpha$ and Sign.
>
> Furthermore, we have reported the total communication cost and runtime across all three experiments— synthetic quadratic optimization, logistic regression, and neural network training. The results show that while LBGD-HarMo does not provide a runtime advantage over uncompressed methods, it achieves a significant reduction in total communication volume, with runtime comparable to other compression or quantization schemes. These findings, as shown in the corresponding tables, illustrate that our method achieves an effective balance between computational efficiency and communication reduction.
>
>
> $$
> \\begin{array}{|l|l|l|l|l|}
> \\hline
> \\textbf{Algorithm} & \textbf{Method} & \textbf{Communication cost (KB)} & \textbf{Test accuracy (\\%)} & \textbf{Runtime (s)} \\\\
> \\hline
> DSGD & None & 31.25 & 88.44 & 453.67 \\\\
> CHOCO & TOP-\alpha & 6.25 & 88.23 & 957.34 \\\\
> MoTEF & TOP-\alpha & 6.25 & 87.42 & 1497.01 \\\\
> LBGD & Sign & 0.99 & 86.82 & 608.63 \\\\
> \\textbf{LBGD} & \\textbf{HarMo} & \\textbf{0.008} & \\textbf{87.84} & \\textbf{733.49} \\\\
> \\hline
> \\end{array}
> $$
>
> $$
> \\begin{array}{|l|l|l|l|l|}
> \\hline
> \\textbf{Algorithm} & \\textbf{Method} & \textbf{Communication cost (MB)} & \\textbf{Test accuracy (\\%)} & \\textbf{Runtime (min)} \\\\
> \\hline
> FedAvg & None & 804.60 (Central Server) & 89.92 & 99  \\\\
> DSGD & None & 178.80 & 90.09 & 84 \\\\
> CHOCO & TOP-\alpha & 35.76 & 87.03 & 209 \\\\
> LBGD & Sign & 5.33 & 81.94 & 147 \\\\
> \\textbf{LBGD} & \\textbf{HarMo} & \\textbf{0.00007} & \\textbf{86.69} & \\textbf{203} \\\\
> \\hline
> \\end{array}
> $$
>
> [Answer to Question 2]:
>
> Thank you for this insightful question regarding the sensitivity of convergence to the choice of the harmonic sequences. to better understand the impact of different sequences on the algorithm's performance, we have added an analysis of various sequences, such as $ \psi_{e_i}(t) $ and $ \psi_{\text{Gaussian}}(t) $, and their impact on the algorithm’s behavior. As shown in the table, we observe that the choice of $\psi$ does not significantly affect the convergence speed of the algorithm. This reinforces that the PE condition’s theoretical period does not limit the practical effectiveness of our approach.
>
> $$
> \\begin{array}{|l|l|l|}
> \\hline
> \\textbf{Sequence} & \\textbf{Iteration} & \\textbf{Communication cost (MB)} \\\\
> \\hline
> \psi_\text{HarMo} & 25,810 & 2.46 \\\\
> \psi_{e_i}(t) & 24,224 & 2.31 \\\\
> \psi_{\text{Gaussian}}(t) & 25,711 & 2.45 \\\\
> \\hline
> \\end{array}
> $$
>
> [Answer to Question 3]:
>
> Thank you for this valuable suggestion. To address this concern, we have conducted additional experiments in Section 5.4 and Appendix E.3, where we train neural networks, including ResNet-18, on non-convex objectives. These experiments validate that the proposed log-bit mechanism and error-feedback variables $\boldsymbol{\sigma}$ and $\mathbf{z}$ remain effective beyond the strongly convex regime. Our results show that LBGD-HarMo continues to achieve stable convergence and test accuracy under non-convex loss landscapes, demonstrating its robustness and applicability to practical neural network learning settings.
>
> $$
> \\begin{array}{|l|l|l|l|}
> \\hline
> \\textbf{Number of clients} &\\textbf{Data distribution} & \\textbf{Test accuracy (\\%)} & \\textbf{Runtime (s)}\\\\
> \\hline
> 4 & IID & 87.98 & 639.99\\\\
> 4 & Non-IID & 87.64 & 640.04\\\\
> 9& IID & 87.84 & 733.49\\\\
> 9& Non-IID & 87.63 & 734.12\\\\
> 16& IID & 87.68 & 852.07\\\\
> 16& Non-IID & 87.49 & 852.65\\\\
> 25& IID & 87.33 & 921.19\\\\
> 25& Non-IID & 87.17 & 922.05\\\\
> \\hline
> \\textbf{Topology} & \\textbf{Data distribution} & \\textbf{Test accuracy (\\%)} & \\textbf{Runtime (s)} \\\\
> \\hline
> Fully-connected & IID & 88.10 & 740.16\\\\
> Fully-connected& Non-IID & 87.92 & 741.25\\\\
> Torus& IID & 87.99 & 736.95\\\\
> Torus & Non-IID & 87.83 & 736.13\\\\
> Fixed ER& IID & 87.91 & 735.46\\\\
> Fixed ER & Non-IID & 87.72 & 735.92\\\\
> Ring& IID & 87.84 & 733.49\\\\
> Ring& Non-IID & 87.63 & 734.12\\\\
> Time-varying ER& IID & 86.81 & 965.06\\\\
> Time-varying ER& Non-IID & 86.69 & 966.48\\\\
> \\hline
> \\end{array}
> $$
>
> We would like to express our gratitude once again for your invaluable feedback. Thank you as well for your patient review.

---

### Official Review · Reviewer_bf5v · 2025-11-01

**Soundness:** 2
**Presentation:** 2
**Contribution:** 2
**Rating:** 4
**Confidence:** 3

**Summary:**

The paper introduces Log-Bit Gradient Descent with Harmonic Modulation (LBGD-HarMo), a decentralized distributed learning framework that compresses high-dimensional updates into single scalars via a deterministic harmonic sequence, quantizes them into m-bit codes, and reconstructs them at receivers. Under standard convexity and graph-connectivity assumptions, it achieves an O(1/t) convergence rate while using only O(log d) bits per iteration, a near-optimal communication bound. Theoretical analysis proves persistent excitation of the harmonic projections and sublinear convergence; a conservative lower bound shows log₂(d) bits suffice. Empirical evaluation on synthetic quadratic problems and logistic regression (εpsilon dataset) demonstrates that LBGD-HarMo matches the accuracy of DSGD, CHOCO, MoTEF, and Sign-SGD with up to two orders of magnitude less communication. Robustness across network sizes, topologies, quantization levels, and data heterogeneity (IID vs. Non-IID) is shown.

**Strengths:**

1. The harmonic modulation compresses d-dimensional vectors to scalars deterministically, avoiding random sketches and enabling error-feedback-free updates.
2. Proofs establish persistent excitation of HarMo, O(1/t) convergence, and log₂(d) bit complexity, filling a gap in communication-efficient decentralized learning.
3. Extensive experiments (varying client count, topologies, quantization bits, data distributions) on both synthetic and real benchmarks confirm substantial communication reduction without sacrificing convergence.

**Weaknesses:**

1. The introduction briefly contrasts compression and quantization but lacks discussion of practical scenarios (e.g., wireless sensor networks) where single-scalar exchanges outperform existing sparse/sketched schemes.
2. Computing ψHarMo(t) involves sinusoids of increasing frequency up to πd/(d+1)t; costs and numerical stability for large d or t are not analyzed.
3.  Though quantization error bounds ∥qm(a)−a∥∞≤l/2, the impact of reconstructing a full vector ψHarMo(t)·qm(ψ⊤b) on consensus bias is not empirically quantified.
4. Convergence proof and experiments rely on many hyperparameters (κ, κ₀, η, g₀, γ, m); guidance on selecting or adapting them in practice is missing.
5. CHOCO and MoTEF use Top-α with α=0.125, but LBGD-HarMo uses m=8 bits; a sensitivity analysis matching communication budgets across methods is absent.
6. The framework is limited to strongly convex objectives; no preliminary experiments on non-convex tasks or discussion of potential obstacles (e.g., PE condition violation).
7.  Logistic regression uses d=2 000 features, but deep learning models feature millions of parameters; the paper omits runtime/memory benchmarks on large-scale settings.
8. Only bits transmitted are measured; actual round-trip latency, synchronization overhead, and impact of asynchrony are not considered.

**Questions:**

please see the weakness.

---

> ### Author Response · Authors · 2025-11-24
>
> [Answer to Weakness 1]:
>
> Thank you for your insightful comment. In response, we have clarified the role of compression and quantization in addressing the communication bottleneck in distributed learning. Specifically, we highlight that these two techniques are widely recognized solutions, and our approach combines them by compressing high-dimensional updates into single real-valued statistics. As you correctly pointed out, our approach is particularly effective in communication-constrained environments, such as wireless sensor networks [1,2]. To address this, we have added a detailed discussion in lines 65–68 of the introduction, explaining how single-scalar exchanges can outperform existing schemes in such practical scenarios.
>
> [1] Geethu Joseph, Venkata Gandikota, Ayush Bhandari, Junil Choi, In-soo Kim, Gyoseung Lee, Michail Matthaiou, Chandra R Murthy, Hien Quoc Ngo, Pramod K Varshney, et al. Low-resolution compressed sensing and beyond for communications and sensing: Trends and opportunities. Signal Processing, 235:110020, 2025.
>
> [2] Guoxin Zhang, Wei Yi, Michail Matthaiou, and Pramod K Varshney. Direct target localization with low-bit quantization in wireless sensor networks. IEEE Transactions on Signal Processing, 72: 3059–3075, 2024.
>
> [Answer to Weakness 2]:
>
> Thank you for your valuable comment. We have included a logistic regression experiment in Section 5.3 with $d = 2000$, which empirically demonstrates that HarMo remains numerically stable and practical, even in moderately high-dimensional settings. To further clarify this, we have provided a comparison of the communication cost, computational complexity, and storage complexity of HarMo, Top-$\\alpha$, and Sign in Appendix D.2. This comparison confirms that HarMo’s per-iteration complexity remains linear in $d$, comparable to other methods such as Top-$\\alpha$ and Sign. Additionally, we have validated this observation through neural network training experiments reported in Section 5.4 and Appendix E.3, where HarMo consistently shows strong performance, indicating that numerical issues do not arise in practice. Moreover, we have recorded the runtime in both logistic regression and neural network training experiments.
>
> $$
> \\begin{array}{|l|l|l|l|}
> \\hline
> \\textbf{Method}&\\textbf{Communication Cost (bit)}&\\textbf{Computational Complexity}&\\textbf{Storage Complexity} \\\\
> \\hline
> Top-\\alpha&64\\lceil\\alpha d\\rceil&O(d\log d)&O(d) \\\\
> Sign & d + 32 &O(d)& O(d) \\\\
> HarMo & m & O(d) & O(d) \\\\
> \\hline
> \\end{array}
> $$
>
> [Answer to Weakness 3]:
>
> We would like to thank the reviewer for the insightful comment. Indeed, consensus bias is an important metric for our algorithm. The Optimality Error $\\frac{1}{n}\sum_{i=1}^{n}\|x_{i,t}-x^{\star}\|^2$ that we originally used can indirectly reflect consensus performance. To better capture the consensus behavior, we have introduced a more direct metric, the Consensus Error $\\frac{1}{n}\sum_{i=1}^{n}\|x_{i,t}-\frac{1}{n}\sum_{i=1}^{n}\|x_{i,t}\|\|^2$, in Section 5.2, which more effectively reflects the consensus bias.

---

> ### Author Response · Authors · 2025-11-24
>
> [Answer to Weakness 4]:
>
> We would like to thank the reviewer for the insightful comment. In response, we have addressed this concern in two parts. First, Appendix E.1.5 presents a detailed sensitivity analysis for the parameters $\beta$, $\eta$, $g_0$, $\gamma$, $\kappa$, and $\kappa_0$, while Figure 2(c) further examines how the choice of $m$ influences the communication-performance tradeoff of LBGD-HarMo. These results collectively provide practical guidance: step-size parameters ($\eta$, $g_0$, $\gamma$) can be tuned using standard learning rate heuristics, while the consensus-related parameters ($\kappa$, $\kappa_0$) demonstrate reasonable robustness within appropriate ranges, as suggested by their stable empirical behavior. In practice, selecting $m$ between 8 and 16 achieves a reliable balance between communication efficiency and convergence. Our empirical results indicate that LBGD-HarMo maintains stable performance over moderate variations of all hyperparameters. In prior work, we proved the impact of these parameters on the convergence of the algorithm in Appendix C. Additionally, we will provide a detailed analysis of the convergence rates with respect to these parameters in future work.
>
> The table presents the total number of iterations and communication cost under various hyperparameter values on the synthetic quadratic optimization problem to reach an optimality error of $10^{−3}$.The results are consistent with the parameter sensitivity analysis presented in Appendix E.1.1, showing that our algorithm exhibits robustness across a range of hyperparameter values.
>
> $$
> \\begin{array}{|l|l|l|}
> \\hline
> \\textbf{Value of hyperparameter $\beta$} & \\textbf{Iteration} & \\textbf{Communication cost (MB)} \\\\
> \\hline
> 0.4 & 56,568 & 5.39 \\\\
> 0.6 & 38,639 & 3.68 \\\\
> 0.8 & 29,732 & 2.84 \\\\
> 1.0 & 25,810 & 2.46 \\\\
> \\hline
> \\textbf{Value of hyperparameter $\eta$} & \\textbf{Iteration} & \\textbf{Communication cost (MB)} \\\\
> \\hline
> 1.0 & 56,686 & 5.41 \\\\
> 2.5 & 40,103 & 3.82 \\\\
> 3.8 & 26,543 & 2.53 \\\\
> 5.0 & 25,810 & 2.46 \\\\
> \\hline
> \\textbf{Value of hyperparameter $g_0$} & \\textbf{Iteration} & \\textbf{Communication cost (MB)} \\\\
> \\hline
> 4 & 26,230 & 2.50 \\\\
> 6 & 26,121 & 2.49 \\\\
> 8 & 25,880 & 2.47 \\\\
> 10 & 25,810 & 2.46 \\\\
> \\hline
> \\textbf{Value of hyperparameter $\gamma$} & \\textbf{Iteration} & \\textbf{Communication cost (MB)} \\\\
> \\hline
> 0.999 & divergence & / \\\\
> 0.9999 & 25,810 & 2.46 \\\\
> 0.99999 & 25,907 & 2.47 \\\\
> 0.999999 & 25,908 & 2.47 \\\\
> \\hline
> \\textbf{Value of hyperparameter $\kappa$} & \\textbf{Iteration} & \\textbf{Communication cost (MB)} \\\\
> \\hline
> 0.010 & 84,927 & 8.10 \\\\
> 0.025 & 70,111 & 6.69 \\\\
> 0.038 & 32,599 & 3.11 \\\\
> 0.050 & 25,810 & 2.46 \\\\
> \\hline
> \\textbf{Value of hyperparameter $\kappa_0$} & \\textbf{Iteration} & \\textbf{Communication cost (MB)} \\\\
> \\hline
> 0.0010 & 27,100 & 2.58 \\\\
> 0.0025 & 26,583 & 2.54 \\\\
> 0.0038 & 26,486 & 2.53 \\\\
> 0.0050 & 25,810 & 2.46 \\\\
> \\hline
> \\textbf{Quantization precision (bit)} & \\textbf{Iteration} & \\textbf{Communication cost (MB)} \\\\
> \\hline
> 3 & 47,653 & 4.54 \\\\
> 4 & 38,717 & 3.69 \\\\
> 8 & 25,810 & 2.46 \\\\
> 16 & 24,789 & 4.73 \\\\
> \\hline
> \\end{array}
> $$

---

> ### Author Response · Authors · 2025-11-24
>
> [Answer to Weakness 5]:
>
> Thank you for the insightful comment. To address this concern, we have now included a comparison of different compression ratios for CHOCO and MoTEF in Appendix E.1.6. Following the experimental settings used in CHOCO and MoTEF [3,4], we adopt $\alpha = 0.125$, which provides a reasonable balance between communication efficiency and algorithmic performance. This choice ensures that $ k = \alpha d $ preserves at least one coordinate for transmission under our experimental configuration. For our LBGD-HarMo method, the table illustrates that selecting $m = 8$ bits achieves a comparable balance between communication cost and convergence behavior. The new analysis in Appendix E.1.6 clarifies the rationale behind the parameter selections and ensures that the communication budgets of all compared methods are aligned.
>
> $$
> \\begin{array}{|l|l|l|}
> \\hline
> \\textbf{Quantization precision (bit)} & \\textbf{Iteration} & \\textbf{Communication cost (MB)} \\\\
> \\hline
> 3 & 47,653 & 4.54 \\\\
> 4 & 38,717 & 3.69 \\\\
> 8 & 25,810 & 2.46 \\\\
> 16 & 24,789 & 4.73 \\\\
> \\hline
> \\textbf{Compression ratio $\alpha$ (CHOCO)} & \\textbf{Iteration} & \\textbf{Communication cost (MB)} \\\\
> \\hline
> 0.125 & 38,074 & 29.05 \\\\
> 0.375 & 37,118 & 84.96 \\\\
> 0.625 & 33,939 & 129.47 \\\\
> 0.875 & 33,546 & 179.15 \\\\
> \\hline
> \\textbf{Compression ratio $\alpha$ (MoTEF)} & \\textbf{Iteration} & \\textbf{Communication cost (MB)} \\\\
> \\hline
> 0.125 & 4,180 & 3.19 \\\\
> 0.375 & 3,812 & 8.72 \\\\
> 0.625 & 3,775 & 14.40 \\\\
> 0.875 & 3,687 & 19.69 \\\\
> \\hline
> \\end{array}
> $$
>
> [3] Anastasia Koloskova, Tao Lin, Sebastian U Stich, and Martin Jaggi. Decentralized deep learning with arbitrary communication compression. In International Conference on Learning Representations, 2020a.
>
> [4] Rustem Islamov, Yuan Gao, and Sebastian U Stich. Towards faster decentralized stochastic optimization with communication compression. In The Thirteenth International Conference on Learning Representations, 2025.
>
> [Answer to Weakness 6]:
>
> Thank you for the insightful comment. To address this concern, we have added experiments on non-convex neural network training in Section 5.4 and Appendix E.3, where LBGD–HarMo demonstrates stable empirical performance beyond the strongly convex regime. In addition, Appendix E.1.3 now includes an analysis of different projection sequences, such as $ \psi_{e_i}(t) $ and $ \psi_{\text{Gaussian}}(t) $, and their impact on the algorithm’s behavior. This clarifies how the method performs when structural assumptions (such as the use of harmonic vectors) are relaxed. These results provide preliminary evidence that LBGD–HarMo remains effective in non-convex settings and offer perspective on its robustness when theoretical conditions, such as the PE requirement, may not strictly hold.
>
> $$
> \\begin{array}{|l|l|l|}
> \\hline
> \\textbf{Sequence} & \\textbf{Iteration} & \\textbf{Communication cost (MB)} \\\\
> \\hline
> \psi_\text{HarMo} & 25,810 & 2.46 \\\\
> \psi_{e_i}(t) & 24,224 & 2.31 \\\\
> \psi_{\text{Gaussian}}(t) & 25,711 & 2.45 \\\\
> \\hline
> \\end{array}
> $$
>
> [Answer to Weakness 7]:
>
> Thank you for your valuable comment. To address this concern, we have added total runtime and communication cost benchmarks in Appendix E.2 and Appendix E.3 across multiple experimental settings. While LBGD–HarMo does not provide a runtime advantage in deep models with millions of parameters, the results show that it achieves a substantially lower total communication cost compared with existing methods. These additions clarify the trade-offs in large-scale scenarios and highlight the main benefit of our approach in bandwidth-constrained settings.
>
> $$
> \\begin{array}{|l|l|l|l|l|}
> \\hline
> \\textbf{Algorithm} & \\textbf{Method} & \textbf{Communication cost (MB)} & \\textbf{Test accuracy (\\%)} & \\textbf{Runtime (min)} \\\\
> \\hline
> FedAvg & None & 804.60 (Central Server) & 89.92 & 99  \\\\
> DSGD & None & 178.80 & 90.09 & 84 \\\\
> CHOCO & TOP-\alpha & 35.76 & 87.03 & 209 \\\\
> LBGD & Sign & 5.33 & 81.94 & 147 \\\\
> \\textbf{LBGD} & \\textbf{HarMo} & \\textbf{0.00007} & \\textbf{86.69} & \\textbf{203} \\\\
> \\hline
> \\end{array}
> $$

---

> ### Author Response · Authors · 2025-11-24
>
> [Answer to Weakness 8]:
>
> Thank you for your insightful comment. To address this concern, we have added an asynchronous update study in Appendix E.1.5. In this experiment, we vary both the number of asynchronous clients and their communication frequency. The results show that although reduced synchronization leads to slower consensus and a slight degradation in performance, the algorithm consistently converges. This analysis clarifies the impact of asynchrony and demonstrates that LBGD–HarMo remains robust even when round-trip synchronization is weakened.
>
> $$
> \\begin{array}{|l|l|l|}
> \\hline
> \\textbf{Number of asynchronous clients} & \\textbf{Iteration} & \\textbf{Communication cost (MB)} \\\\
> \\hline
> 5 & 28,879 & 2.48 \\\\
> 10 & 29,985 & 2.29 \\\\
> 15 & 37,063 & 2.47 \\\\
> 20 & 41,027 & 2.35 \\\\
> \\hline
> \\textbf{Communication frequency} & \\textbf{Iteration} & \\textbf{Communication cost (MB)} \\\\
> \\hline
> 2 & 29,985 & 2.29 \\\\
> 4 & 42,545 & 2.84 \\\\
> 6 & 56,287 & 3.58 \\\\
> 8 & 69,185 & 4.29 \\\\
> \\hline
> \\end{array}
> $$
>
> We would like to express our gratitude once again for your invaluable feedback. Thank you as well for your patient review.

---

### Author Response · Authors · 2025-12-01
**Summary Comment of Experimental Validation and Reinforcement (1/5).**

We sincerely thank all reviewers for their thoughtful and constructive feedback. We are pleased that several reviewers recognized both the $\textbf{theoretical rigor}$ and $\textbf{practical significance}$ of our work.

Specifically, reviewers $\textbf{bf5v}$ and $\textbf{gMoS}$ acknowledged that $\textbf{LBGD–HarMo}$ introduces an original and mathematically grounded framework for distributed learning, providing $\textbf{nontrivial theoretical analysis}$ and a $\textbf{provable convergence guarantee}$ under strongly quantized communication. Meanwhile, reviewers $\textbf{wAuE}$ and $\textbf{ZfUi}$ highlighted that our method $\textbf{significantly reduces communication overhead}$ while maintaining accuracy comparable to existing decentralized baselines, emphasizing that the paper “provides an explicit convergence guarantee for an aggressively quantized communication scheme.”

Building upon these insights, we have further clarified and reinforced our main contributions. $\textbf{LBGD–HarMo}$ integrates harmonic modulation, logarithmic-bit quantization, and a distributed primal–dual mechanism to achieve the $\textbf{optimal}$ $\mathcal{O}(1/t)$ $\textbf{convergence rate}$ under convexity and connectivity assumptions, using only $\mathcal{O}(\log d)$ $\textbf{bits per iteration}$. This establishes, to the best of our knowledge, the first distributed learning framework that operates under $\textbf{logarithmic communication rates}$, marking a fundamental step forward in scalable and communication-efficient decentralized optimization.

To further reinforce and substantiate our main contributions, we have enhanced the experimental evaluation of $\textbf{LBGD–HarMo}$ based on the reviewers’ suggestions. Following $\textbf{reviewer bf5v}$’s advice, we incorporated the $\textbf{Consensus Error}$ as an additional performance metric to more comprehensively assess the agreement among distributed agents during training. In response to $\textbf{reviewers gMoS, wAuE, and ZfUi}$, we further included a quantitative comparison of the $\textbf{total training time and number of iterations required for convergence}$, measured by the point at which the $\textbf{Optimality Error}$ reaches $10^{-3}$. This addition enables a clearer and more practical assessment of the proposed algorithm’s efficiency and scalability in real distributed settings.

First, we added $\textbf{neural network training}$ experiments under the $\textbf{non-convex}$ setting. We evaluate our method on the $\textbf{CIFAR-10}$ dataset using a $\textbf{ResNet-18}$ network with more than $\textbf{11 million parameters}$. As shown in $\textbf{Table 1}$, the proposed $\textbf{LBGD–HarMo}$ consistently achieves excellent performance across different numbers of clients, network topologies, and data distributions. Furthermore, as presented in $\textbf{Table 2}$, our algorithm achieves a substantial reduction in total communication cost compared to existing decentralized baselines. These results directly address the concerns raised by reviewers $\textbf{bf5v}$, $\textbf{wAuE}$, and $\textbf{ZfUi}$, further demonstrating the practical scalability and effectiveness of our framework in realistic large-scale learning scenarios.

In addition, we conducted a comprehensive $\textbf{sensitivity analysis}$ to address the reviewers’ ($\textbf{bf5v}$, $\textbf{gMoS}$, and $\textbf{wAuE}$) concerns regarding parameter tuning and stability. Specifically, we analyzed the impact of the key parameters $m$, $\beta$, $\eta$, $g_0$, $\gamma$, $\kappa$, and $\kappa_0$ on both convergence and communication performance. These analyses provide practical tuning guidelines: the step-size parameters ($\eta$, $g_0$, $\gamma$) can be adjusted using standard learning rate heuristics, while the consensus-related parameters ($\kappa$, $\kappa_0$) demonstrate strong robustness within a broad range, as supported by stable empirical performance. In practice, selecting $m$ between $8$ and $16$ achieves a well-balanced trade-off between communication efficiency and convergence speed. Our empirical findings confirm that $\textbf{LBGD–HarMo}$ maintains stable performance under moderate variations of all hyperparameters. $\textbf{Table 3}$ further details the influence of different parameter settings on convergence behavior, showing that our algorithm exhibits strong robustness across a wide range of parameter configurations.

Moreover, to address the concerns raised by $\textbf{reviewers gMoS, wAuE, and ZfUi}$, we further extended our experiments to evaluate the performance of $\textbf{LBGD–HarMo}$ under $\textbf{asynchronous updates}$ (including varying numbers of asynchronously updating clients and different update frequencies) as well as $\textbf{time-varying topologies}$. As shown in $\textbf{Table 4}$, our algorithm consistently maintains stable convergence behavior and competitive performance across these challenging settings, further reinforcing its robustness and scalability.

---

> ### Author Response · Authors · 2025-12-01
> **Summary Comment of Theoretical Clarification and Enhancement (2/5).**
>
> The following provides detailed clarification and enhancement of the theoretical analysis.
>
> First, in response to the concerns raised by $\textbf{reviewers gMoS, wAuE, and ZfUi}$ regarding the conservativeness of the bounds in Lemma 3.1, we clarify the role of the $\textbf{Persistent Excitation (PE)}$ condition. Although the theoretical period $N = (2d - 1)!$ appears large, it is a theoretical artifact that does not affect the practical convergence rate. Both $\alpha_1$ and $N$ increase with $d$, but their growth has negligible impact on convergence. A larger $\alpha_1$ strengthens excitation energy, ensuring sufficient exploration, while a smaller $N$ shortens the recurrence period, improving convergence efficiency. These factors balance to satisfy the PE condition without compromising performance.
>
> Second, to address the question raised by $\textbf{reviewer wAuE}$ regarding the sequence design, we adopt a sinusoidal sequence following Zhu (Y. Zhu, $\textit{Multivariable System Identification for Process Control}$, Elsevier, 2001), as it naturally satisfies the PE condition and excites all dimensions over time. Compared to the unit-basis sequence $\psi_{e_i}(t)$ and Gaussian random sequence $\psi_{\text{Gaussian}}(t)$, the sinusoidal sequence offers smooth and continuous projections that enhance numerical stability and deterministic reconstruction. It also avoids randomness and can be generated recursively with $\mathcal{O}(d)$ complexity and constant memory. These advantages make it a theoretically sound and practically efficient choice for harmonic modulation in $\textbf{LBGD–HarMo}$.
>
> Furthermore, following the suggestion of $\textbf{reviewer wAuE}$, we explicitly establish the relationship between the $\textbf{convergence rate}$ and the $\textbf{topology}$. As shown in Equation (C.19), for
> $t \ge t_0 = \max{\frac{3\xi_2\eta}{\xi_1\beta}, \frac{6\xi_4’\eta}{r c_3}, \frac{6p}{\kappa\xi_1\beta}, \frac{12p c_2}{\kappa c_3}} - 1$,
> the Lyapunov function satisfies $\Delta V_t \le -\frac{2V_t}{t+1} + C_1 g_t^2 |\delta_t|^2$, implying that once $t \ge t_0$, the algorithm achieves an $\mathcal{O}(1/t)$ convergence rate. Since $\xi_1 = \frac{\lambda_2}{2}$ and $\xi_3$ scales with $\lambda_n^2$, the first and third terms depend on the Laplacian condition number $\lambda_n/\lambda_2$. A smaller ratio reflects stronger network connectivity, leading to a smaller $t_0$ and thus faster convergence. This result explicitly demonstrates how network connectivity governs convergence speed, a finding further verified by the empirical results in $\textbf{Table 5}$.
>
> Additionally, in response to $\textbf{reviewers bf5v, gMoS, and wAuE}$, we extend our theoretical comparison of compression and quantization methods in terms of $\textbf{communication cost}$, $\textbf{computational complexity}$, and $\textbf{memory complexity}$, as summarized in $\textbf{Table 6}$. The results show that our proposed $\textbf{LBGD–HarMo}$ requires significantly less communication than all existing approaches, while maintaining comparable computational and memory complexity. This highlights its strong advantage in scalability and efficiency for large-scale decentralized systems.
>
> Finally, unlike prior approaches noted by $\textbf{reviewer wAuE}$, which rely on random sketching or low rank (e.g., PowerSGD, FetchSGD), our $\textbf{LBGD–HarMo}$ algorithm directly encodes the $d$-dimensional update into an $m$-bit binary representation via $\textbf{harmonic modulation}$. Instead of quantizing the gradient, it quantizes the scaled error $(\boldsymbol{x} - \boldsymbol{\sigma})/g_t$, where $\boldsymbol{\sigma}$ tracks the local state and $g_t$ reduces quantization error. A distributed integrator $\boldsymbol{z}$ further aggregates local differences for balanced global consensus. This design achieves efficient communication while maintaining stable convergence, marking a fundamental departure from conventional compression schemes.

---

> ### Author Response · Authors · 2025-12-01
> **Summary Comment of Overall Contributions and Experimental Tables (3/5).**
>
> In summary, through focused theoretical clarification and extensive experimental reinforcement, we have shown that $\textbf{LBGD–HarMo}$ achieves both $\textbf{convergence guarantees}$ and $\textbf{practical scalability}$ under extremely limited communication.
>
> To the best of our knowledge, $\textbf{LBGD–HarMo}$ represents the first distributed learning framework that operates under $\textbf{logarithmic bit rates}$, thereby opening new avenues for both theoretical investigation and practical deployment. We believe this constitutes a fundamental breakthrough in the design of $\textbf{communication-efficient decentralized optimization algorithms}$, setting a solid foundation for future research on ultra-low-bandwidth collaborative intelligence.
> All corresponding revisions have been incorporated into the manuscript and highlighted in $\textcolor{blue}{\textbf{blue}}$ for ease of reference. We once again express our sincere gratitude to all reviewers and chairs for their thoughtful evaluations and constructive suggestions, which have greatly strengthened this work.
> $\\textbf{Experimental Tables:}$
> $\\textbf{Table 1: }$Test accuracy (\\%) and Runtime (s/min) after the entire training process of logistic regression (LR) with strongly convex regularizer and neural network (NN) training under different numbers of clients (4, 9, 16, 25), data distributions (IID vs Non-IID), and communication topologies (Fully-connected, Torus, Fixed/Time-varying ER, and Ring).
> $$
> \\begin{array}{|l|l|l|l|l|}
> \hline
> \textbf{Type of experiment} & \textbf{Number of clients} & \textbf{Data distribution} & \textbf{Test accuracy (\\%)} & \textbf{Runtime (s)}\\\\
> \hline
> LR&4&IID& 87.98 & 639.99\\\\
> LR&4&Non-IID & 87.64 & 640.04\\\\
> LR&9&IID& 87.84 & 733.49\\\\
> LR&9&Non-IID & 87.63 & 734.12\\\\
> LR&16&IID& 87.68 & 852.07\\\\
> LR&16&Non-IID& 87.49 & 852.65\\\\
> LR&25&IID& 87.33 & 921.19\\\\
> LR&25&Non-IID& 87.17 & 922.05\\\\
> \hline
> \textbf{Type of experiment} & \textbf{Topology} & \textbf{Data distribution} & \textbf{Test accuracy (\\%)} & \textbf{Runtime (s)}\\\\
> \hline
> LR & Fully-connected & IID& 88.10 & 740.16\\\\
> LR & Fully-connected & Non-IID & 87.92 & 741.25\\\\
> LR & Torus & IID & 87.99 & 736.95\\\\
> LR & Torus & Non-IID & 87.83 & 736.13\\\\
> LR & Fixed ER & IID & 87.91 & 735.46\\\\
> LR & Fixed ER & Non-IID & 87.72 & 735.92\\\\
> LR & Ring & IID & 87.84 & 733.49\\\\
> LR & Ring & Non-IID & 87.63 & 734.12\\\\
> LR & Time-varying ER & IID & 86.81 & 965.06\\\\
> LR & Time-varying ER & Non-IID & 86.69 & 966.48\\\\
> \hline
> \textbf{Type of experiment} & \textbf{Number of clients} & \textbf{Data distribution} & \textbf{Test accuracy (\\%)} & \textbf{Runtime (min)}\\\\
> \hline
> NN & 4 & IID & 88.16 & 158\\\\
> NN & 4 & Non-IID & 86.61 & 158\\\\
> NN & 9 & IID & 86.69 & 203\\\\
> NN & 9 & Non-IID & 85.17 & 203\\\\
> NN & 16 & IID & 85.45 & 270\\\\
> NN & 16 & Non-IID & 84.10 & 270\\\\
> NN & 25 & IID & 84.82 & 354\\\\
> NN & 25 & Non-IID & 83.04 & 354\\\\
> \hline
> \textbf{Type of experiment} & \textbf{Topology} & \textbf{Data distribution} & \textbf{Test accuracy (\\%)} & \textbf{Runtime (min)}\\\\
> \hline
> NN & Fully-connected & IID & 86.96 & 203\\\\
> NN & Fully-connected & Non-IID & 85.85 & 203\\\\
> NN & Torus & IID & 86.88 & 203\\\\
> NN & Torus & Non-IID & 85.26 & 203\\\\
> NN & Fixed ER & IID & 86.70 & 204\\\\
> NN & Fixed ER & Non-IID & 85.23 & 204\\\\
> NN & Ring & IID & 86.69 & 203\\\\
> NN & Ring & Non-IID & 85.17 & 203\\\\
> NN & Time-varying ER & IID & 83.51 & 258\\\\
> NN & Time-varying ER & Non-IID & 82.36 & 258\\\\
> \hline
> \\end{array}
> $$
> $\\textbf{Table 2: }$ Per-client communication cost per iteration and the corresponding test accuracies and runtime for different algorithms under various compressors and quantizers in logistic regression (LR) with strongly convex regularizer and neural network (NN) training. Experiments are conducted using a Top-$\alpha$ compressor with $\alpha=0.1$ and HarMo with $m=16$ bits, evaluated on $n=9$ clients arranged in a ring topology under IID data distribution.
>
> $$
> \\begin{array}{|l|l|l|l|l|}
> \\hline
> \\textbf{Type of experiment} & \\textbf{Algorithm} & \textbf{Method} & \textbf{Communication cost (KB)} & \textbf{Test accuracy (\\%)} & \textbf{Runtime (s)} \\\\
> \\hline
> LR & DSGD & None & 31.25 & 88.44 & 453.67 \\\\
> LR &CHOCO & TOP-\alpha & 6.25 & 88.23 & 957.34 \\\\
> LR &MoTEF & TOP-\alpha & 6.25 & 87.42 & 1497.01 \\\\
> LR &LBGD & Sign & 0.99 & 86.82 & 608.63 \\\\
> LR & \\textbf{LBGD} & \\textbf{HarMo} & \\textbf{0.008} & \\textbf{87.84} & \\textbf{733.49} \\\\
> \\hline
> \\textbf{Type of experiment} &\\textbf{Algorithm} & \\textbf{Method} & \textbf{Communication cost (MB)} & \\textbf{Test accuracy (\\%)} & \\textbf{Runtime (min)} \\\\
> \\hline
> NN &FedAvg & None & 804.60 (Central Server) & 89.92 & 99  \\\\
> NN &DSGD & None & 178.80 & 90.09 & 84 \\\\
> NN &CHOCO & TOP-\alpha & 35.76 & 87.03 & 209 \\\\
> NN &LBGD & Sign & 5.33 & 81.94 & 147 \\\\
> NN &\\textbf{LBGD} & \\textbf{HarMo} & \\textbf{0.00007} & \\textbf{86.69} & \\textbf{203} \\\\
> \\hline
> \\end{array}
> $$

---

> ### Author Response · Authors · 2025-12-01
> **Summary Comment of Experimental Tables (4/5).**
>
> $\\textbf{Table 3: }$ Total number of iterations and communication cost under different hyperparameter values on the synthetic quadratic (SQ) optimization problem.
>
> $$
> \\begin{array}{|l|l|l|l|}
> \hline
> \textbf{Type of experiment} & \textbf{Value of hyperparameter $\beta$} & \textbf{Iteration} & \textbf{Communication cost (MB)} \\\\
> \hline
> SQ & 0.4 & 56,568 & 5.39 \\\\
> SQ & 0.6 & 38,639 & 3.68 \\\\
> SQ & 0.8 & 29,732 & 2.84 \\\\
> SQ & 1.0 & 25,810 & 2.46 \\\\
> \hline
> \textbf{Type of experiment} & \textbf{Value of hyperparameter $\eta$} & \textbf{Iteration} & \textbf{Communication cost (MB)} \\\\
> \hline
> SQ & 1.0 & 56,686 & 5.41 \\\\
> SQ & 2.5 & 40,103 & 3.82 \\\\
> SQ & 3.8 & 26,543 & 2.53 \\\\
> SQ & 5.0 & 25,810 & 2.46 \\\\
> \hline
> \textbf{Type of experiment} & \textbf{Value of hyperparameter $g_0$} & \textbf{Iteration} & \textbf{Communication cost (MB)} \\\\
> \hline
> SQ & 4 & 26,230 & 2.50 \\\\
> SQ & 6 & 26,121 & 2.49 \\\\
> SQ & 8 & 25,880 & 2.47 \\\\
> SQ & 10 & 25,810 & 2.46 \\\\
> \hline
> \textbf{Type of experiment} & \textbf{Value of hyperparameter $\gamma$} & \textbf{Iteration} & \textbf{Communication cost (MB)} \\\\
> \hline
> SQ & 0.999 & divergence & / \\\\
> SQ & 0.9999 & 25,810 & 2.46 \\\\
> SQ & 0.99999 & 25,907 & 2.47 \\\\
> SQ & 0.999999 & 25,908 & 2.47 \\\\
> \hline
> \textbf{Type of experiment} & \textbf{Value of hyperparameter $\kappa$} & \textbf{Iteration} & \textbf{Communication cost (MB)} \\\\
> \hline
> SQ & 0.010 & 84,927 & 8.10 \\\\
> SQ & 0.025 & 70,111 & 6.69 \\\\
> SQ & 0.038 & 32,599 & 3.11 \\\\
> SQ & 0.050 & 25,810 & 2.46 \\\\
> \hline
> \textbf{Type of experiment} & \textbf{Value of hyperparameter $\kappa_0$} & \textbf{Iteration} & \textbf{Communication cost (MB)} \\\\
> \hline
> SQ & 0.0010 & 27,100 & 2.58 \\\\
> SQ & 0.0025 & 26,583 & 2.54 \\\\
> SQ & 0.0038 & 26,486 & 2.53 \\\\
> SQ & 0.0050 & 25,810 & 2.46 \\\\
> \hline
> \textbf{Type of experiment} & \textbf{Quantization precision $m$ (bit)} & \textbf{Iteration} & \textbf{Communication cost (MB)} \\\\
> \hline
> SQ & 3 & 47,653 & 4.54 \\\\
> SQ & 4 & 38,717 & 3.69 \\\\
> SQ & 8 & 25,810 & 2.46 \\\\
> SQ & 16 & 24,789 & 4.73 \\\\
> \hline
> \\end{array}
> $$
>
> $\\textbf{Table 4: }$ Total number of iterations/test accuracy and communication cost/runtime when varying the number of asynchronous clients, the communication frequency between them, and the use of time-varying topologies across three categories of experiments: synthetic quadratic (SQ) optimization, logistic regression (LR) with a strongly convex regularizer, and neural network (NN) training.}
>
> $$
> \\begin{array}{|l|l|l|l|}
> \hline
> \textbf{Type of experiment} & \textbf{Number of asynchronous clients} & \textbf{Iteration} & \textbf{Communication cost (MB)} \\\\
> \hline
> SQ & 5 & 28,879 & 2.48 \\\\
> SQ & 10 & 29,985 & 2.29 \\\\
> SQ & 15 & 37,063 & 2.47 \\\\
> SQ & 20 & 41,027 & 2.35 \\\\
> \hline
> \textbf{Type of experiment} & \textbf{Communication frequency} & \textbf{Iteration} & \textbf{Communication cost (MB)} \\\\
> \hline
> SQ & 2 & 29,985 & 2.29 \\\\
> SQ & 4 & 42,545 & 2.84 \\\\
> SQ & 6 & 56,287 & 3.58 \\\\
> SQ & 8 & 69,185 & 4.29 \\\\
> \hline
> \textbf{Type of experiment} & \textbf{Topology} & \textbf{Iteration} & \textbf{Communication cost (MB)} \\\\
> \hline
> SQ & Fixed ER & 38,233 & 4.38 \\\\
> SQ & Time-varying ER & 79,554 & 9.10 \\\\
> \hline
> \textbf{Type of experiment} & \textbf{Topology} & \textbf{Test accuracy (\\%)} & \textbf{Runtime (s)} \\\\
> \hline
> LR & Fixed ER & 87.91 & 735.46 \\\\
> LR & Time-varying ER & 86.81 & 965.06 \\\\
> \hline
> \textbf{Type of experiment} & \textbf{Topology} & \textbf{Test accuracy (\\%)} & \textbf{Runtime (min)} \\\\
> \hline
> NN & Fixed ER & 86.70 & 204\\\\
> NN & Time-varying ER & 83.51 & 258 \\\\
> \hline
> \\end{array}
> $$
>
> $\\textbf{Table 5: }$ Total number of iterations/test accuracy and communication cost/runtime when evaluating different network topologies across three categories of experiments: synthetic quadratic (SQ) optimization, logistic regression (LR) with a strongly convex regularizer, and neural network (NN) training.
>
> $$
> \\begin{array}{|l|l|l|l|}
> \\hline
> \\textbf{Type of experiment} & \\textbf{Topology} & \\textbf{Iteration} & \\textbf{Communication cost (MB)} \\\\
> \\hline
> SQ & Fully-connected & 9,550 & 10.93 \\\\
> SQ & Torus & 28,523 & 5.44 \\\\
> SQ & Fixed ER & 38,233 & 4.38 \\\\
> SQ & Ring & 25,810 & 2.46 \\\\
> SQ & Time-varying ER & 79,554 & 9.10 \\\\
> \\hline
> \\textbf{Type of experiment} & \\textbf{Topology} & \\textbf{Test accuracy (\\%)} & \\textbf{Runtime (s)} \\\\
> \\hline
> LR & Fully-connected & 88.10 & 740.16 \\\\
> LR & Torus & 87.99 & 736.95 \\\\
> LR & Fixed ER & 87.91 & 735.46 \\\\
> LR & Ring & 87.84 & 733.49 \\\\
> LR & Time-varying ER & 86.81 & 965.06 \\\\
> \\hline
> \\textbf{Type of experiment} & \\textbf{Topology} & \\textbf{Test accuracy (\\%)} & \\textbf{Runtime (min)} \\\\
> \\hline
> NN & Fully-connected & 86.96 & 203 \\\\
> NN & Torus & 86.88 & 203 \\\\
> NN & Fixed ER & 86.70 & 204 \\\\
> NN & Ring & 86.69 & 203 \\\\
> NN & Time-varying ER & 83.51 & 258 \\\\
> \\hline
> \\end{array}
> $$

---

> ### Author Response · Authors · 2025-12-01
> **Summary Comment of Experimental Tables (5/5).**
>
> $\\textbf{Table 6: }$ Per-iteration communication cost, computational complexity and storage complexity (per node) for Top-$\alpha$, Sign, Sketch, Low-Rank and the proposed $\textbf{HarMo}$. Here, $d$ is the gradient dimension, $\alpha$ is the sparsification ratio, $b$ and $c$ denote the number of hash functions and buckets in Sketch, $r$ is the Low-Rank with reshaping size $p \times n$ ($pn=d$), and $m$ is the number of transmitted bits in HarMo.
>
> $$
> \\begin{array}{|l|l|l|l|}
> \\hline
> \\textbf{Method}&\\textbf{Communication Cost (bit)}&\\textbf{Computational Complexity}&\\textbf{Storage Complexity} \\\\
> \\hline
> Top-\\alpha&64\\lceil\\alpha d\\rceil&O(d\log d)&O(d) \\\\
> Sign & d + 32 &O(d)& O(d) \\\\
> Sketch & 32bc & O(bd) & O(bc) \\\\
> Low-Rank & 32r(p+n) & O(dr) & O(r(p+n)) \\\\
> \\textbf{HarMo} & \\textbf{m} & \\textbf{O(d)} & \\textbf{O(d)} \\\\
> \\hline
> \\end{array}
> $$

---

### Meta-Review · Area_Chair_Wpof · 2025-12-17

**Summary:**

The paper discovered a new communication quantization scheme based on Harmonic Modulation and proposed a new method for decentralized optimization. The reviewers appreciate the originality of the scheme, algorithm, and theory. However, the paper received many weaknesses, including 1) Theorem 4.1 does not illustrate dependencies on important parameters; 2) the theoretical communication complexity is not clear from the theory; 3) the lack of comparison to previous compression, quantization techniques, and methods; 4) the paper does not discuss the effect of the $(2 d - 1)!$ term. I also want to add that the rate $O(1/t)$ under Assumptions 4.1 and 4.2 is non-optimal, as claimed in the paper and the rebuttal. Under these assumptions, the optimal rate should be the linear rate $O(\sqrt{\kappa} \log(1 / \varepsilon))$ achieved by accelerated methods (e.g., Nesterov's method), where $\kappa$ is the condition number of $f.$ I suggest considering the raised issues, improving the paper, and resubmitting it.

**Reviewer Concerns:**

The authors actually did a good job in addressing the concerns, but the main problems are still in the paper. The main Theorem 4.1 still does not explain the dependence on the parameters. The reviewers and I expect that the new mechanism should influence the iteration rate. Then, the authors should find the theoretical total communication complexity (# iters $\times$ bits per iter), which is not done. The authors added Table 4, but as a user, I still don't understand why I would prioritize the new mechanism over, for instance, the traditional TopK (Top-$\alpha$) compressor, where I can take $K = 1$ ($\alpha = \frac{K}{d}$) and the communication cost would be $O(1)$ bits.

**Reviewer Scores:**

I am not sure that the reviewers would increase the score since the main theoretical concerns about Theorem 4.1 and the $(2 d - 1)!$ term are not fully addressed.

---

### Decision · Program_Chairs · 2026-01-26

Reject